# Revolutionizing Training-Free NAS: Towards Efficient Automatic Proxy Discovery via Large Language Models

**Haidong Kang**[1][*]   **Lihong Lin**[1]   **Hanling Wang**[2]
[1]Northeastern University, China       [2]Pengcheng Laboratory, China
hdkang@stumail.neu.edu.cn, linlh@mails.neu.edu.cn, wanghl03@pcl.ac.cn

## Abstract

The success of computer vision tasks is mainly attributed to the architectural design of neural networks. This highlights the need to automatically design high-performance architectures via Neural Architecture Search (NAS). To accelerate the search process, training-free NAS is proposed, which aims to search high-performance architectures at initialization via zero-cost proxies (ZCPs). However, existing zero-cost proxies heavily rely on manual design, which is often labor-intensive and requires extensive expert knowledge. In addition, these crafted proxies often suffer from poor correlation with final model performance and high computational complexity, severely limiting NAS efficiency in real-world applications. To address those issues, this paper proposes a novel Large Language Models (LLMs)-driven Automatic Proxy Discovery (**APD**) framework, which revolutionizes the design paradigm of ZCPs by leveraging LLMs to automatically discover optimal ZCPs for Training-Free NAS. Moreover, we utilize actor-critic based reinforcement learning to optimize prompts, enabling to generate better ZCPs in the next generation. We conduct extensive experiments on mainstream NAS benchmarks, demonstrating APD excels in both performance and efficiency. Besides, we firmly believe that our APD will dramatically benefit the deep learning community through providing novel paradigm of design algorithms via LLMs.

## 1   Introduction

Neural networks plays an indispensable role in computer vision tasks due to its superior performance, which raises a trend to deploy neural networks (i.e., ResNet He et al. [2016]) on resource-intensive scenarios. However, conventional neural networks designed by human experts suffer from Out-Of-Memory (OOM) problems due to dramatically limited resources Xie et al. [2023], Kang et al. [2025]. Therefore, this highlights the need to design lightweight architectures, liberating the handicraft neural networks from the OOM bottleneck.

Recently, Neural Architecture Search (NAS) Liu et al. [2018], Ye et al. [2022], Kang has emerged as a promising paradigm for its searching high-performance architectures in an automatic manner, while disrupting the conventional paradigm of manually designed architecture. Despite its potential, NAS still suffers from a key bottleneck of huge computational budgets Ma et al. [2024]. To tackle this, training-free NAS Mellor et al. [2021], Abdelfattah et al. [2021], Wang et al. [2020], Chen et al. [2022], Lee and Ham [2024], Peng et al. [2024] is proposed to liberate NAS from the computational bottleneck via the lens of without gradient descent. Essentially, the training-free NAS leverages ZCPs, predicting the accuracy ranking of architectures in a training-free manner. Specifically, the ZCPs rely

---

[*]Corresponding author

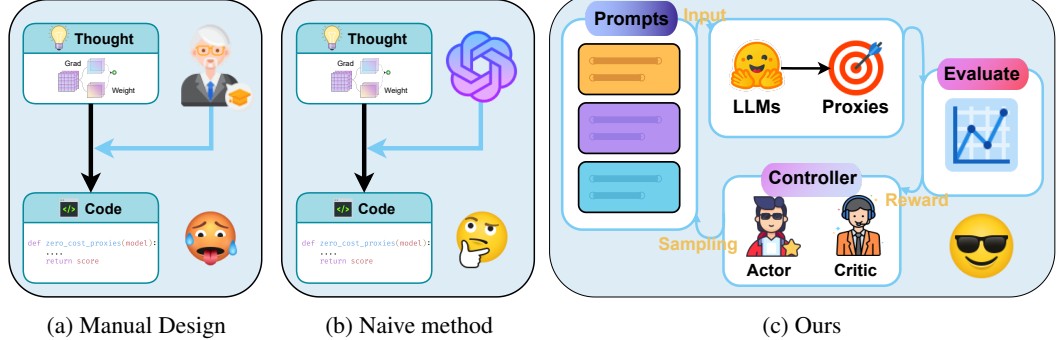

| (a) Manual Design | (b) Naive method | (c) Ours |

Figure 1: A comparison of the designed way of ZCPs. (a) Manual design relies on expert knowledge. (b) A naive method via LLMs. (c) Our method proposes an LLM-driven APD framework to automatically discover optimal ZCPs for Training-Free NAS.

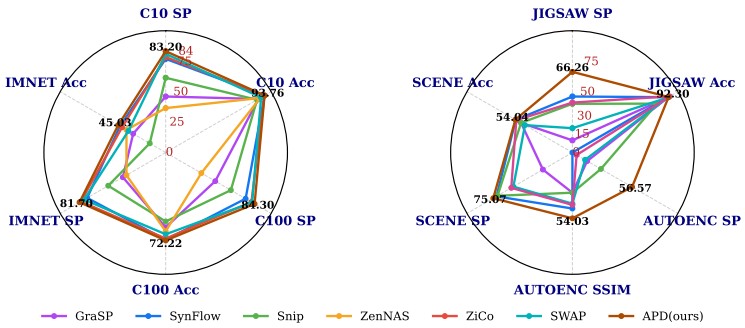

Figure 2: Spearman (SP) ranking correlations & accuracies (Acc) of zero-cost proxies on NAS-Bench-201 (Left) and Trans-Bench-101 (right).

on some statistical or theoretical properties (i.e., the number of parameters, Neural Tangent Kernel) of neural architectures to assess their expressivity.

## 1.1 Challenges

Although these approaches have taken the first step towards ZCPs tailored for training-free NAS Haidong et al. [2025], current ZCPs still suffer from several fatal drawbacks: (1) *they require extensive expert knowledge through hundreds of time-consuming trial-and-error processes (as shown in Fig. 1a and Table 1), increasing the labor costs and making it hard to design new ZCPs; (2) these crafted ZCPs incur poor correlation with the final accuracy of searched architecture (as shown in Fig. 2 and 3), making the users do not know why those ZCPs failed and hard to deploy in real-world applications.* This is identified with ZiCO Li et al. [2023], which points out that the performance of simple proxies (i.e., #Params, FLOPs) excels most of the crafted ZCPs. Those limitations highlight the need to rethink the design paradigm of AZPs.

**Our New Observation.** Different from existing methods for training-free NAS, as depicted in Fig. 1b, and Fig. 1c, we observe a new way to automatically design ZCPs via Large Language Models (LLMs) in this work, which can effectively address the aforementioned drawbacks by revolutionizing the traditional manual design manner. More details are shown in Section 3.

## 1.2 Contributions

In this work, we attempt to analyze and address the above drawbacks. To fulfill our goal, we first conduct an in-depth analysis by rethinking the design of ZCPs, and experimentally confirm their limitations (as shown in Section 2). To tackle those drawbacks, motivated by powerful large language models (LLMs) Radford et al. [2018], Brown et al. [2020], Achiam et al. [2023], Chang et al. [2024], Liu et al. [2024] for generating new ideas and knowledge, we provide an affirmative answer by proposing a novel way to automatically design ZCPs via LLMs, dubbed APD. Specifically, we first explore a naive method (as depicted in Fig. 1b) using a simple prompt as input to LLMs, however,

Spearman (SP) ranking correlation of ZCPs searched by the naive method is very poor. This raises the challenge and need of how to seek new strategy, improving the effectiveness of LLM-driven ZCPs. To reveal the root cause, we conduct an in-depth analysis and observe that the naive method lacks thought between the prompt and tasks, which could be a potential reason of poor correlation. Inspired by GPT-4o and Deepseek R1 Guo et al. [2025], we leverage actor-critic based reinforcement learning (as shown in Fig. 1c) as the reasoning engine, enabling it to generate intermediate reasoning steps before the final answer. This plays a critical Chain-of-Thought (CoT) role in enhancing APD performance, especially in NAS tasks requiring multi-step reasoning. We summarize the main contributions of this work as follows:

- **New ZCPs paradigm.** To the best of our knowledge, we are the first to propose a novel ZCPs paradigm by leveraging LLMs, providing a new perspective for understanding and designing training-free NAS.

- **CoT driven strategy**. Beyond the limit of poor correlation of simple prompts for designing ZCPs, we first reveal that the root cause is the absence of positive reward signals for these proxies during search. To address this, we propose an actor-critic based reinforcement learning to build reasoning engine, achieving improvements akin to Chain-of-Thought reasoning in GPT-4o and DeepSeek-R1 and yielding much stronger proxy-to-performance correlation.

- **Numerical Verification.** Extensive experiments validate the superiority of our APD, outperforming previous methods on mainstream search spaces and datasets.

## 2 Rethinking the design of Zero-cost Proxies

The primary goal of AZPs is to accurately predict the ranking of architectures without training on a given search space. As shown in Table 1, the representative AZPs (i.e., SNIP, SWAP) leverage heuristics, statistical, or gradient properties to measure expressibility of architectures. However, those AZPs heavily rely on human expertise, which may be suboptimal for new search spaces or datasets. In addition, designing new ZCPs is time-consuming.

| Method | Corresponding formula | Human expert |
|--------|----------------------|:---:|
| SNIP | $\left\|\left(\frac{\partial \mathcal{L}}{\partial \theta}\right) \odot \theta\right\|$ | ✓ |
| Fisher | $\sum_{z \in A}\left(\frac{\partial \mathcal{L}}{\partial z} z\right)^2$ | ✓ |
| SynFlow | $\left(\frac{\partial \mathcal{R}}{\partial \theta}\right) \odot \theta, \mathcal{R} = \mathbb{1}^T\left(\prod_{\theta_i \in W}\|\theta_i\|\right)\mathbb{1}$ | ✓ |
| AZ-NAS | $s^{AZ}(i) = \sum_{\mathcal{M} \in \{\mathcal{E}, \mathcal{P}, \mathcal{T}, \mathcal{C}\}} \log \frac{\text{Rank}(s^{\mathcal{M}}(i))}{m}$ | ✓ |
| SWAP | $\Psi_{\mathcal{N},\theta} = \left\|\hat{\mathbb{A}}_{\mathcal{N},\theta}\right\|$ | ✓ |
| **APD** | $s^{APD} = \left(\sum_{l \in \mathcal{B}} \frac{\|M_l(\theta)\|_F^2}{\|M_l(\theta)\|_2^2}\right) \times \left(\sum_{l \in \mathcal{C}} \frac{\|W_l(\theta)\|_1}{\|W_l(\theta)\|_2}\right)$ | ✗ |

Table 1: Comparison of existing ZCPs. Here, $s^{APD}$ is the best-performing ZCP discovered by APD on NAS-Bench-201. In APD, $\mathcal{B}$ and $\mathcal{C}$ represent batchnorm and convolutional layers, while $M$ and $W$ denote their outputs and weights.

Notably, ZiCO Li et al. [2023] highlights that simple proxies such as the number of parameters and FLOPs often outperform many hand-crafted ZCPs in predicting neural architecture performance. As shown in Fig. 3, hand-crafted ZCPs (i.e., Grasp, AZ-NAS, SWAP) suffer from a significant poor correlation issue. Those experimentation with hand-crafted ZCPs motivate us to seek a new ZCPs paradigm for designing training-free NAS.

Inspired by the knowledge generation capabilities of LLMs, we raise a new question: can LLMs automatically generate zero-cost proxies (ZCPs) for NAS tasks? To test this, we designed a simple prompt (**App. A**) that asks the LLM to output a ZCP, based on the assumption that LLMs have internalized rich knowledge of neural networks during pretraining. On NAS-Bench-201, the generated ZCP achieved a 60.51% spearman correlation (as shown in Table 6) on NAS-Bench-201 search space in CIFAR-10 dataset, lower than hand-crafted methods, yet encouraging for LLM-driven ZCP discovery. Further analysis reveals that the lack of reasoning and feedback is the key limitation, as prompt-based generation is a black-box process. To address this, we propose an actor–critic reinforcement learning framework, inspired by Chain-of-Thought (CoT) prompting in GPT-4o. This

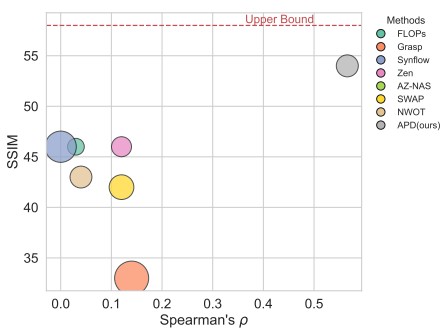

Figure 3: A comparison of ZCPs on AutoEncoding of Trans-Bench-101.

design introduces a feedback loop between the LLM
and the NAS task, enabling the LLM to refine ZCPs iteratively and significantly improving both the
performance and efficiency of proxy generation (as shown in Fig. 3).

## 3 APD: A Resourceful Adviser for Training-Free NAS

### 3.1 Automatic Proxy Discovery

To fulfill the goal of designing ZCPs for the training-free NAS, APD utilizes LLMs to automatically
generate proxies by evolving both natural language descriptions and corresponding code. In addition,
we propose an actor-critic RL controller sampling appropriate prompt strategies to guide the evolution,
aiming to optimize the correlation between proxies and final model performance. The overall
framework of APD is depicted in Fig. 4, which consisting of three main components as follows:

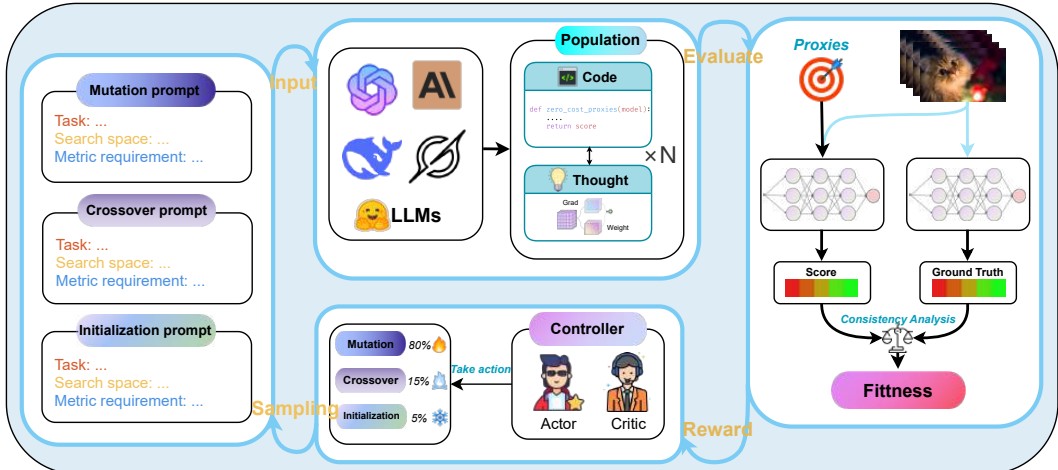

Figure 4: Overview of APD. APD utilizes large-scale pre-trained models to automatically search for
the optimal zero-cost proxies tailored for training-free NAS.

**Proxy Candidate Generator.** LLM in APD serves as proxy candidate generator. Carefully structured
prompts enable it to synthesize new ZCPs or refine existing ZCPs. Let $\mathcal{P}$ denote the set of valid
proxies that satisfy a fixed I/O contract. During time step $t$, with the current proxy population
$P_t(P_t \subseteq \mathcal{P})$, the LLM $\mathcal{L}$ receives both a structured prompt $\pi \in \Pi$ that specifies the requested
operation $\Pi(\text{initialization}, \text{mutation}, \text{crossover})$ and a bounded context window $\mathcal{C}_t \subseteq P_t$ containing
existing proxies together with their natural language rationales and codes. This pair $(\pi, \mathcal{C}_t)$ induces a
context-conditioned distribution:

$$\mu_{\pi,\mathcal{C}_t} = \mathrm{P}_{\mathcal{L}}(f|\pi, \mathcal{C}_t), \tag{2}$$

where $\mu_{\pi,\mathcal{C}_t}$ represents the probability distribution over candidate proxies induced by prompt $\pi$ and
context $\mathcal{C}_t$ with LLM $\mathcal{L}$. Composing over all prompts and admissible contexts yields an implicit,
context-aware search space:

$$\mathcal{F} = \bigcup_{t \in N} \bigcup_{\pi \in \Pi} \bigcup_{\mathcal{C} \subseteq P_t} \mathrm{supp}(\mu_{\pi,\mathcal{C}}), \tag{3}$$

where $\mathrm{supp}(\mu_{\pi,\mathcal{C}})$ is the support set of distribution $\mu_{\pi,\mathcal{C}}$. Obviously, we have $\mathcal{F} \subseteq \mathcal{P}$. Relative to
context-free pairs, this construction enlarges support while preserving ergodic reachability and allows
the generator to reuse salient patterns already discovered in $\mathcal{P}$, and thereby drives the realized search
space $\mathcal{A}$ to lie as close as possible to $\mathcal{P}$.

In APD, three categories of context-conditioned pairs are employed, each of which gives rise to
distinct distribution:

- **Initialization** $\mu_{\text{init},\emptyset}$ provides only the task description, I/O contract and ample prior knowl-
  edge that depicts input architectures and data.

- **Mutation** $\mu_{\mathrm{mut},f}$ conditions on a single proxy $f$ to perform local perturbations, enabling fine-grained exploitation.
- **Crossover** $\mu_{\mathrm{cross},\mathbf{f}}$ furnishes at least two parent proxies and asks the LLM to identify their common principles and fuse complementary components, encouraging recombination of useful motifs.

The context-aware Proxy Candidate Generator endows APD with flexible exploration granularity: during a single evolutionary step, the algorithm can elect either to densely sample previously unvisited regions of the proxy space or to perform fine-grained, local refinement around promising candidates, thus supporting both broad coverage and deep exploitation as needed.

**Fitness Evaluator.** The fitness evaluator swiftly quantifies each candidates ZCP by computing its Spearman correlation with ground-truth accuracies on the given NAS benchmark $\mathcal{A}$ (e.g., NAS-Bench-201). For a benchmark set $\mathcal{B} = \{(a_i, p_i)\}_{i=1}^{m}$ of architectures and ground-truth accuracies under dataset $\mathcal{D}$. The fitness assigned to $f$ is

$$\phi(f) = \rho(f(\mathbf{a}), \mathbf{p}) - \beta \mathrm{cost}(f). \tag{4}$$

Where $\rho$ is Spearman or Kendall correlation, computed on randomly sampled subset of $\mathcal{B}$ to provide an unbiased estimate of ranking fidelity. $\mathrm{cost}(f)$ is the average runtime required to evaluate one architecture of $\mathcal{B}$. $\beta$ is a hyperparameter that controls the trade-off between predictive quality and computational efficiency.

**RL Evolution Scheduler.** To render the proxy evolution strategy learnable and capable of converging efficiently toward optimal ZCPs, APD introduces an actor–critic module that serves as the evolutionary decision-maker. The actor–critic treats the fitness score returned by the evaluator as its reward and learns a policy that maximizes this signal, thereby jointly optimizing both the evolving set of ZCPs and the evolution strategy itself.

In a given search space $\mathcal{A}$ of candidate architectures and their ground-truth performance $p(a)$, the objective of APD is to learn a proxy $f : \mathcal{A} \to \mathbb{R}$ whose scores preserve the ranking induced by $p$. Therefore, we aim to maximize the expected correlation:

$$\max_{f \in \mathcal{F}} \mathbb{E}_{\mathbf{a} \subseteq \mathcal{A}}[\rho(f(\mathbf{a}), p(\mathbf{a}))], \tag{1}$$

where $\rho$ is Spearman $\rho_s$. Each proxy $f$ is represented as a tuple $(\mathcal{T}, \mathcal{C})$ of $\mathcal{T}$ a natural language thought describing the proxy's principle, and $\mathcal{C}$ an executable code that returns a scalar score, ensuring interpretability and deterministic replay throughout the evolutionary process.

To accelerate convergence toward high correlation proxies while maintaining effective exploration in search space $\mathcal{F}$, APD introduces a light-weight Actor-Critic as evolution scheduler that governs every evolutionary step. At generation $t$, the controller observes a compact state vector with fixed-width histogram of fitness values and strategies, sampling an action $a_t \in \Pi$ from a categorical policy $\pi_\theta(a|s_t)$. Executing $a_t$ yields candidate proxies $P_t'$, whose fitness $\phi(P_t')$ is evaluated by the Fitness Evaluator. The scheduler then receives a clipped reward $r_t = \mathbb{E}(\phi(P_t'))$ that promotes substantive improvements and updates the Actor-Critic by standard advantage-actor-critic gradient steps:

$$\begin{aligned}
\theta &\leftarrow \theta + \eta \, \nabla_\theta \log \pi_\theta(a_t \mid s_t) \left[ r_t + \gamma V_\psi(s_{t+1}) - V_\psi(s_t) \right], \\
\psi &\leftarrow \psi - \eta_v \, \nabla_\psi \big( r_t + \gamma V_\psi(s_{t+1}) - V_\psi(s_t) \big)^2.
\end{aligned} \tag{5}$$

Where $\theta$ and $\psi$ denote the actor and critic parameters, $\eta$ and $\eta_v$ are the actor and critic learning rates, $V_\psi$ is the value baseline parameterized by $\psi$, and $\gamma$ is the discount factor. The critic minimizes $(r + \gamma V(s_{t+1}) - V(s_t))^2$, and the actor maximizes the advantage-weighted log likelihood.

## 3.2 Evolution Framework

Based on the components outlined in Section 3.1, we integrate the Proxy Candidate Generator, Fitness Evaluator, and the RL Evolution Scheduler into evolutionary loop. As shown in **Algorithm** 1, APD unfolds through the following mutually dependent steps:

**Step 0 Initialization** In time step 0, an $(\mathrm{initialization}, \emptyset)$ is issued to the LLM, producing the seed $P_0 = \{f_1, \cdots, f_N\}$.

**Step 1 Generation** At generation t the actor samples an action $a_t = (op, C_t) \sim \pi_\theta(\cdot|s_t)$ where $op \in \{init, mut, cross\}$. The corresponding prompt and context window $\mathcal{C}_t \subseteq P_t$ are fed to the LLM, yielding candidate proxies $P_t^{'}$.

**Step 2 Evaluation** The candidates are scored in $\mathcal{B}$ by the fitness function $\phi(\cdot)$. If the proxy fails the contract check, it is assigned $\phi = -\infty$.

**Step 3 Policy update** The scheduler receives reward $r_t = \mathbb{E}(\phi(f))$ by calculating the mean $\phi$ and updates $(\theta, \psi)$ by the rule in (5).

**Step 4 Population replacement** The union $P_t \cup \{P_t^{'}\}$ is sorted by fitness. The worst proxies are dropped to keep $|P_{t+1}| = N$. A tie-breaking rule that favors younger proxies prevents age-related stagnation. The loop then returns to **Step 1**.

The loop terminates after $T_{max}$ generations. Empirically, the framework produced proxies whose Spearman correlation on NASBench201 exceeds 0.80 within 30 generations roughly one GPU-hour on a single RTX4090.

---

**Algorithm 1** Evolution Framework

---

**Require:** LLM $\mathcal{L}$, benchmark $\mathcal{B}$, population $N$, budget $T_{max}$, actor-critic $(\pi_\theta, V_\psi)$
   $P_0 \leftarrow$ first $N$ proxies from $\mu_{init,\emptyset}$, $s_0 \leftarrow \{\emptyset, \emptyset\}$
   **for** $t = 1$ to $T_{max}$ **do**
      $op \sim \pi(\cdot|s_t), \mathcal{C} \leftarrow P$
      $P^{'} \leftarrow \mathcal{L}(\mu_{op,\mathcal{C}}), \varphi \leftarrow \phi(P^{'})$
      $r_t \leftarrow \mathbb{E}(\varphi),$ update $(\theta, \psi, s_{t+1})$
      $P \leftarrow (P \cup P^{'})/\{worst(P)\}$
   **end for**
   **return** $\arg\max_{f \in \mathcal{F}} \phi(f)$

---

# 4 Experiments

## 4.1 Experimental Settings

We evaluate APD on 5 representative search spaces (e.g., NAS-Bench-201 Dong and Yang [2020], NAS-Bench-101 Ying et al. [2019], DARTS Liu et al. [2018], TransNAS-Bench-101–Micro Duan et al. [2021], OoD-ViT-NAS Bai et al. [2021]) across 4 tasks: image recognition, autoencoding, scene classification, and self-supervised jigsaw puzzles. We transfer the zero-cost proxy identified on CIFAR-10 within the NAS-Bench-201 search space to all datasets in both NAS-Bench-201 and NAS-Bench-101. However, due to the substantial search space disparities between TransNAS-Bench-101, OoD-ViT-NAS, and NAS-Bench-201, we search for new zero-cost proxies directly within these spaces to ensure optimal performance. In addition, we train APD with 7 mainstream LLMs (i.e., GPT4o Achiam et al. [2023], Claude 3.7 Anthropic [2025], Deepseek V3 Liu et al. [2024], Gemini flash Google Cloud [2025], Llama 4 Meta AI [2025], Grok 3 xAI [2025]). Full experimental details are provided in **App. B**.

## 4.2 Performance on NAS-Bench-201&101 search spaces

Table 2 compares APD with state-of-the-art training-free NAS methods on NAS-Bench-201 (CIFAR-10/100, ImageNet16-120) and NAS-Bench-101 (CIFAR-10). APD consistently achieves the highest test accuracy and ranking correlations (SPR/KT) across all datasets. Compared to the best existing methods, APD improves test accuracy by +0.07% on CIFAR-10 (vs. ZiCo), +0.52% on CIFAR-100 (vs. Synflow), and +0.69% on ImageNet16-120 (vs. AZ-NAS), while reducing runtime cost by over 50%. On ImageNet16-120, APD also outperforms Grasp by +14.18%, ZenNAS by +7.85%, ZiCo by +3.61%, and AZ-NAS by +0.69%. These results not only highlight the efficiency and robustness of APD across diverse search spaces, but also demonstrate the strong potential of LLM-driven ZCP design, advancing the development of training-free NAS.

Table 2: Performance comparison on NAS-Bench-201 (NB-201) search space with CIFAR-10/100, and ImageNet16-120 datasets, and NAS-Bench-101 (NB-101) search space with CIFAR-10 dataset. We report Kendall's $\tau$ (KT) and Spearman's $\rho$ (SPR) computed with all candidate architectures. In addition, we report the average independent runs results obtained from our method. The red, blue, and orange indicate the best, second-best, and third-best results, respectively.

| Method | CIFAR-10(NB-201) | | | CIFAR-100(NB-201) | | | ImageNet16-120(NB-201) | | | CIFAR10(NB-101) | | Runtime (ms/arch) |
|---|---|---|---|---|---|---|---|---|---|---|---|---|
| | SPR | KT | Test acc.(%) | SPR | KT | Test acc.(%) | SPR | KT | Test acc.(%) | SPR | Test acc.(%) | |
| Params | 0.751 | 0.576 | 93.61 ± 0.08 | 0.726 | 0.552 | 70.95 ± 0.37 | 0.690 | 0.519 | 41.67 ± 0.64 | 0.38 | 92.18 ± 1.53 | - |
| FLOPs | 0.733 | 0.541 | 93.61 ± 0.08 | 0.708 | 0.517 | 70.95 ± 0.37 | 0.691 | 0.517 | 41.67 ± 0.64 | 0.67 | 92.18 ± 1.53 | - |
| Snip Abdelfattah et al. [2021] | 0.614 | 0.455 | 86.63 ± 3.89 | 0.618 | 0.461 | 56.53 ± 7.56 | 0.545 | 0.409 | 15.13 ± 10.86 | 0.71 | 85.41 ± 1.38 | 45.78 |
| Grasp Abdelfattah et al. [2021] | 0.460 | 0.318 | 88.01 ± 3.14 | 0.470 | 0.329 | 61.43 ± 7.04 | 0.406 | 0.282 | 30.85 ± 5.66 | 0.45 | 87.08 ± 1.40 | 93.70 |
| Synflow Abdelfattah et al. [2021] | 0.769 | 0.571 | 93.67 ± 0.39 | 0.758 | 0.562 | 71.70 ± 0.94 | 0.745 | 0.553 | 43.39 ± 3.21 | 0.38 | 89.93 ± 2.97 | 78.26 |
| NWOT Mellor et al. [2021] | 0.743 | 0.557 | 91.95 ± 1.29 | 0.769 | 0.579 | 68.88 ± 1.60 | 0.760 | 0.573 | 42.31 ± 3.43 | 0.32 | 93.16 ± 0.36 | 36.54 |
| ZenNAS Lin et al. [2021] | 0.365 | 0.283 | 89.55 ± 1.12 | 0.338 | 0.245 | 64.69 ± 3.86 | 0.372 | 0.260 | 37.18 ± 3.17 | 0.65 | 93.06 ± 0.73 | 30.07 |
| ZiCo Li et al. [2023] | 0.784 | 0.589 | 93.69 ± 0.07 | 0.813 | 0.620 | 70.63 ± 1.08 | 0.804 | 0.614 | 41.42 ± 0.97 | 0.65 | 92.64 ± 0.99 | 75.50 |
| AZ-NAS Lee and Ham [2024] | 0.913 | 0.741 | 93.49 ± 0.30 | 0.900 | 0.723 | 70.33 ± 1.16 | 0.886 | 0.710 | 44.34 ± 1.26 | 0.42 | 92.01 ± 0.85 | 69.43 |
| SWAP Peng et al. [2024] | 0.810 | 0.634 | 90.48 ± 0.94 | 0.820 | 0.649 | 67.13 ± 1.83 | 0.774 | 0.610 | 35.40 ± 3.96 | 0.44 | 90.51 ± 2.08 | 47.61 |
| **APD** | 0.832 | 0.635 | 93.76 ± 0.09 ↑(0.07) | 0.843 | 0.654 | 72.22 ± 0.65 ↑(0.52) | 0.817 | 0.633 | 45.03 ± 0.76 ↑(0.69) | 0.73 | 93.49 ± 0.34 ↑(0.33) | 16.81 ↓(13.26) |
| Optimal | - | - | 94.33 ± 0.08 | - | - | 73.30 ± 0.20 | - | - | 46.97 ± 0.19 | - | 93.87 ± 0.08 | - |

Table 3: Comparison on the DARTS search space using CIFAR-10/100 (left) and ImageNet1k (right). "C10", "C100", and "Img" indicate search conducted on CIFAR-10, CIFAR-100, and ImageNet1k, respectively. All models are retrained using official code from Liu et al. [2018] for fair comparison. "Paper" shows results from original publications; "∗" denotes results from released training code Zheng et al. [2021]; "–" means unavailable or unreleased data.

| Method | CIFAR-10 Top-1 (%) | | CIFAR-100 Top-1 (%) | Params (M) | GPU-Days | Search method |
|---|---|---|---|---|---|---|
| | Retrain | Paper | Paper | | | |
| ResNet18 He et al. [2016] | - | - | 75.61 | 11.2 | - | Manual |
| AmoebaNet-A Zoph et al. [2018] | - | 97.45 ± 0.05 | 16.82 | - | 3150 | Evolution |
| ENAS Pham et al. [2018] | - | - | 81.09 | 4.6 | 0.5 | Evolution |
| PNAS Liu et al. [2018] | - | 96.59 | 82.37 | 3.2 | 225 | SMBO |
| NASNet-A Zoph et al. [2018] | - | 97.37 | 82.19 | 3.3 | 1,800 | RL |
| DARTS(2nd) Liu et al. [2018] | - | 97.24 ± 0.09 | 82.46 | 3.3 | 1 | Gradient |
| DARTS(1nd) Liu et al. [2018] | - | 97.00 ± 0.14 | 83.18 | 3.3 | 0.4 | Gradient |
| P-DARTS Chen et al. [2019] | 97.30 ± 0.15∗ | 97.50 | 83.37 | 3.4 | 0.3 | Gradient |
| PC-DARTS Xu et al. [2019] | 97.29 ± 0.11∗ | 97.43 ± 0.07 | 82.89 | 3.6 | 0.1 | Gradient |
| DARTS+ Liang et al. [2019] | - | 97.50 ± 0.11 | 83.72 | 3.7 | 0.4 | Gradient |
| DARTS- Chu et al. [2020] | 97.38∗ | 97.41 ± 0.08 | 82.49 | 3.5±0.13 | 0.4 | Gradient |
| FairDARTS-D Chu et al. [2020] | 97.29∗ | 97.46 ± 0.05 | - | 3.8 | 0.4 | Gradient |
| DARTS+PT Wang et al. [2021] | - | 97.39 ± 0.08 | - | 3.0 | 0.8 | Gradient |
| β-DARTS Ye et al. [2022] | - | 97.47 ± 0.08 | 83.48 | 3.8±0.15 | 0.4 | Gradient |
| Λ-DARTS Movahedi et al. [2023] | 97.48 ± 0.11 | 97.50 ± 0.05 | 83.47 | 3.5±0.13 | - | Gradient |
| FP-DARTS Wang et al. [2023] | - | 97.50 ± 0.05 | 83.50±0.05 | 3.9 | 0.08 | Gradient |
| DARTS-$AER^k$ Jing et al. [2023] | - | 97.47 ± 0.02 | - | 3.64 ± 0.18 | 0.3 | Gradient |
| IS-DARTS He et al. [2024] | - | 97.44 ± 0.04 | - | 4.25 ± 0.22 | 0.42 | Gradient |
| NWOT Mellor et al. [2021] | 95.73 | - | - | 5.0 | - | Training-free |
| TENAS Chen et al. [2021] | 97.37±0.064 | - | - | 3.8 | 0.05 | Training-free |
| NASI-ADA Shu et al. [2022] | - | 97.10±0.13 | - | 3.7 | 0.24 | Training-free |
| SWAP Peng et al. [2024] | - | 97.52±0.09 | - | 4.3 | 0.004 | Training-free |
| **APD(C10)** | 97.63±0.13 ↑(0.13) | - | 84.83±0.09 ↑(1.33) | 4.4 | 0.004 | Training-free |

| Method | Top-1(%) | Top-5(%) | Params (M) | FLOPs (M) | GPU-Days | Search method |
|---|---|---|---|---|---|---|
| ResNet50 He et al. [2016] | 75.3 | 92.2 | 25.6 | 4100 | - | Manual |
| AmoebaNet-A Zoph et al. [2018] | 74.5 | 92.4 | 6.4 | 555 | 3150 | Evolution |
| ProxylessNAS-RL Cai et al. [2018] | 74.6 | 92.3 | 5.8 | 465 | 8.3 | RL |
| EfficientNet-B0 Tan and Le [2019] | 76.3 | 93.2 | 5.3 | 390 | ≈3000 | RL |
| NASNet-A Zoph et al. [2018] | 74.0 | 91.6 | 5.3 | 564 | 2000 | RL |
| DARTS Liu et al. [2018] | 73.3 | 91.3 | 4.7 | 574 | 4 | Gradient |
| FBNet Wu et al. [2019] | 74.9 | - | 5.5 | 375 | 216 | Gradient |
| P-DARTS(C100) Chen et al. [2019] | 75.3 | 92.5 | 5.1 | 577 | 0.3 | Gradient |
| PC-DARTS(Img) Xu et al. [2019] | 75.8 | 92.7 | 5.3 | 597 | 3.7 | Gradient |
| DARTS+ Liang et al. [2019] | 76.3 | 92.8 | 5.1 | 591 | 0.2 | Gradient |
| DARTS-(Img) Chu et al. [2020] | 76.2 | 93.0 | 4.9 | 467 | 4.5 | Gradient |
| FairDARTS-B(Img) Chu et al. [2020] | 75.1 | 92.5 | 4.8 | 541 | - | Gradient |
| DARTS+PT(C10) Wang et al. [2021] | 74.5 | 92.0 | 4.6 | - | 0.8 | Gradient |
| β-DARTS(C100) Ye et al. [2022] | 75.8 | 92.9 | 5.4 | 597 | 0.4 | Gradient |
| Λ-DARTS Movahedi et al. [2023] | 75.7 | - | 5.2 | - | - | Gradient |
| FP-DARTS(C10) Wang et al. [2023] | 75.7 | 92.7 | 5.4 | - | 0.08 | Gradient |
| PDARTS-$AER^{ud}$ Jing et al. [2023] | 76.0 | 92.8 | 5.1 | 578 | 2.0 | Gradient |
| IS-DARTS He et al. [2024] | 75.9 | 92.9 | 6.4 | - | 0.42 | Gradient |
| NAO Luo et al. [2018] | 74.3 | 91.8 | 11.4 | 584 | 200 | Proxy |
| TENAS Chen et al. [2021] | 75.5 | 92.5 | 5.4 | - | 0.17 | Training-free |
| NASI-ADA(C10) Shu et al. [2022] | 75.0 | 92.2 | 4.9 | 559 | 0.01 | Training-free |
| SWAP Shu et al. [2022] (Img) | 76.0 | 92.4 | 5.8 | - | 0.006 | Training-free |
| **APD(Img)** | 76.9 ↑(0.6) | 94.0 ↑(0.8) | 6.3 | 695 | 0.004 ↓(1.5×) | Training-free |

## 4.3 Performance on DARTS Search Space

**CIFAR-10 and CIFAR-100 Datasets.** As shown in Table 3, APD achieves state-of-the-art performance with significantly reduced search costs. On CIFAR-10, APD attains a top-1 accuracy of 97.63% using merely 0.004 GPU-days, surpassing representative gradient-based methods such as DARTS and advanced training-free methods like TENAS. Similarly, on CIFAR-100, APD achieves 84.83% accuracy, outperforming leading competitors including DARTS+ and FP-DARTS. These results clearly demonstrate that APD sets a new benchmark in both accuracy and search efficiency across datasets.

**ImageNet1k Dataset**. Table 3 (right) summarizes results on ImageNet1k, demonstrating the strong generalization capability of our method. APD achieves 76.9% top-1 accuracy, improving over the leading training-free method (e.g. NASI-ADA and SWAP) and the existing best one-shot method. Importantly, APD achieves

Table 4: Results on TransNAS-Bench-101-Micro.

| Method | Autoencoding | Scene Classification | Jigsaw |
|---|---|---|---|
| | SSIM | Accuracy (%) | Accuracy (%) |
| Ground Truth | 0.58 | 54.9 | 95.4 |
| Grad_norm Abdelfattah et al. [2021] | 0.36± 0.03 | 48.7±0.7 | 80.3±0.3 |
| SNIP Abdelfattah et al. [2021] | 0.33±0.04 | 48.7±1.1 | 80.3±0.1 |
| Grasp Abdelfattah et al. [2021] | 0.33±0.06 | 50.2±1.6 | 91.1±0.3 |
| Fisher Abdelfattah et al. [2021] | 0.49±0.01 | 48.7±0.6 | 83.5±1.2 |
| Synflow Abdelfattah et al. [2021] | 0.46±0.07 | 53.7±1.2 | 90.9±0.4 |
| NWOT Mellor et al. [2021] | 0.43±0.02 | 53.2±0.6 | 92.3±0.3 |
| Zen-score Lin et al. [2021] | 0.46±0.01 | 53.7±0.2 | 87.5±0.4 |
| GradSign Zhang and Jia [2021] | 0.35±0.03 | 53.6±0.4 | 93.1±0.4 |
| Params | 0.46 | 53.7 | 85.9 |
| FLOPs | 0.46 | 53.7 | 85.9 |
| ZiCo Li et al. [2023] | 0.48±0.02 | 53.7±0.4 | 93.2±0.4 |
| SWAP Peng et al. [2024] | 0.42±0.02 | 45.0±10.9 | 89.8±5.6 |
| **APD** | 0.54±0.01 ↑(0.06) | 54.0±0.6 ↑(0.3) | 91.2±0.1 ↓(2) |

these improvements with a minimal search cost of only 0.004 GPU-days, which is 1.5× less than SWAP. These findings clearly demonstrate APD's excellent performance and generalization efficiency across different datasets.

## 4.4 Results on TransNAS-Bench-101–Micro

As summarized in Table 4, our method consistently achieves state-of-the-art performance across all three tasks on TransNAS-Bench-101-Micro, highlighting its strong generalization capability heterogeneous downstream tasks. In particular, for the Autoencoding task, our approach attains an SSIM score of 0.54, surpassing leading methods such as ZiCo (0.48), NWOT (0.43), and SWAP (0.42). Similar performance improvements are observed in the Scene Classification and Jigsaw tasks, demonstrating APD's robustness and effectiveness across diverse downstream tasks.

## 4.5 Generalizability on OoD-ViT-NAS-Ti

To further evaluate the generalizability of our method on vision transformers under Out-of-Distribution (OoD) conditions, we conduct experiments on the OoD-ViT-NAS-Ti search space Ho et al. [2025] across five datasets: ImageNet1k, ImageNet-A, ImageNet-R, ImageNet-D/Texture, and ImageNet-D/Material. All results are averaged over 5 independent runs. As shown in Table 5, APD consistently outperforms prior methods across all OoD settings. Specifically, APD achieves +0.04 higher $\rho$ than NWOT on ImageNet1k, and +0.01 over DSS on ImageNet-D/Texture. On ImageNet-R, APD surpasses all methods with a highest $\rho$ of 0.88, outperforming AutoProx by +0.10 and DSS by +0.07. These results highlight the strong generalizability of APD under distribution shifts, and further confirm the effectiveness of leveraging LLMs to design accurate and robust zero-cost proxies for transformer-based NAS.

Table 5: Correlation on OoD-ViT-NAS-Ti search space.

| Method | ImageNet1k | ImageNet-A | ImageNet-R | ImageNet-D/Texture | ImageNet-D/Material |
|--------|------------|------------|------------|--------------------|--------------------|
| SNIP | 0.38 | 0.51 | 0.55 | -0.06 | 0.11 |
| Grasp | -0.03 | -0.06 | -0.07 | -0.01 | 0.03 |
| MeCo | 0.48 | 0.40 | 0.33 | 0.09 | 0.08 |
| CroZe | 0.40 | 0.54 | 0.60 | 0.01 | 0.12 |
| DSS | 0.62 | 0.82 | 0.81 | 0.02 | 0.17 |
| AutoProx | 0.67 | 0.82 | 0.78 | 0.05 | 0.15 |
| NWOT | 0.75 | 0.76 | 0.74 | 0.11 | 0.12 |
| **APD** | 0.79↑(0.04) | 0.82 | 0.88 | 0.12↑(0.01) | 0.15 |

## 4.6 Ablation studies

**The impact of various LLMs:** To scrutinize the impact of various LLMs (i.e., GPT4o, Llama 4), we perform an ablation study of LLMs on the NAS-Bench-201 search space in the CIFAR-10 dataset (As shown Table 6 and the left panel of Fig. 5) and the TransNAS-Bench-101 Micro search space (as shown Fig. 5 right). Specifically, we conduct experiments by averaging 4 independent runs to keep a fair comparison. From Table 6 and Fig. 5, we draw two conclusions: (1) Our APD demonstrates strong robustness across different LLMs. Specifically, APD achieves highly consistent results under different LLMs, showing minimal performance fluctuations across models. Moreover, APD is stable to noise caused by different training settings, further enhancing its reliability in real-world deployment. (2) Compared to Naive method, our APD significantly improves the performance in terms of accuracy. For example, the accuracy of APD with Claude 3.7 is 21.26% higher than the Naive method. The performance improvement can be attributed to our proposed actor–critic reinforcement learning framework, which provides key reasoning and feedback for proxy generation.

Table 6: Comparison of APD with different LLMs on NAS-Bench-201 search space in CIFAR-10 dataset.

| Method | LLMs | Run 1 | Run 2 | Run 3 | Run 4 | Average ($\rho$) |
|--------|------|-------|-------|-------|-------|------------------|
| Naive | GPT4o | 60.37 | 59.74 | 60.52 | 61.40 | 60.51 |
| APD | Claude 3.7 | 80.90 | 80.91 | 82.72 | 81.91 | 81.14 |
| APD | Deepseek V3 | 79.77 | 80.78 | 79.77 | 79.77 | 80.24 |
| APD | Gemini flash | 75.47 | 73.31 | 73.50 | 74.66 | 75.22 |
| APD | Llama 4 | 73.04 | 72.70 | 72.32 | 72.95 | 72.87 |
| APD | GPT4o | 81.59 | 80.96 | 81.75 | 82.62 | 81.10 |
| APD | Grok 3 | 72.23 | 79.77 | 80.78 | 84.51 | 79.32 |

**The impact of various components:** We conduct ablation studies on NAS-Bench-201 (CIFAR-10/100) to evaluate the effectiveness of key components in APD. As shown in Table 7, the naive version without Evolution and Actor-Critic modules performs poorly (82.04% in CIFAR-10 and 47.62% in CIFAR-100). Introducing Evolution

Table 7: Ablation study of various components.

| | Naive | Evolution | Actor-Critic | CIFAR-10(NB-201) | CIFAR-100(NB-201) |
|---|-------|-----------|--------------|------------------|-------------------|
| ① | ✓ | ✗ | ✗ | 82.04 | 47.62 |
| ② | ✓ | ✓ | ✗ | 88.53 | 61.16 |
| ③ | ✓ | ✓ | ✓ | 93.76 | 72.22 |

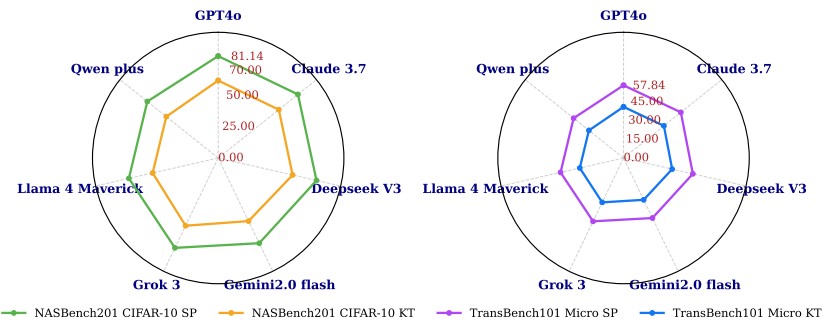

Figure 5: A comparison of ranking correlations of our method designed by 7 mainstream LLMs including GPT4o, Claude 3.7, Deepseek V3, Gemini2.0 flash, Grok 3, Llama 4 Maverick, and Qwen plus. Specifically, we conduct the empirical validation on NAS-Bench-201 (Left) and TransNAS-Bench-101 (Right) search spaces in the CIFAR-10 dataset.

alone significantly boosts performance, yielding +6.49% and +13.54% improvements, respectively. In addition, we observe that there is a joint contribution between Actor-Critic and Evolution. Specifically, ③ obtains 93.76% in CIFAR-10 and 72.22% in CIFAR-100. Those results validate the effectiveness of our APD.

## 4.7 Discussion

We are surprised by the remarkable performance of zero-cost proxies designed by LLMs for training-free NAS, in this section, we conduct an in-depth analysis of APD. APD achieves impressive performance primarily due to the effectiveness of the reinforcement learning strategy. The RL policy enables adaptive selection and refinement of zero-cost proxies, guiding the LLM toward generating more task-relevant strategies. This interaction mechanism significantly enhances the synergy between LLMs and NAS, ensuring that the discovered proxies are both generalizable and high-performing.

Despite involving LLMs, our framework mitigates the typical "black-box" concerns by embedding the LLM within an RL-guided optimization loop. This interaction provides a structured feedback signal, reducing the opaqueness of the overall process and making the optimization more transparent and controllable. Moreover, our approach redefines the traditional NAS design paradigm by shifting from handcrafted heuristics to LLM-guided discovery. This not only introduces a new learning framework for NAS but also offers a transferable methodology for other deep learning domains.

## 4.8 Additional Evaluation

Due to the page limit, we provide more experimental results in **App. A-H**. ❶ **Detailed Prompt Engineering of APD** are presented in **App. C**. ❷ **More ablation studies** are presented in **App. D**. ❸ **Limitations** are presented in **App. E**. ❹ **Visualizations of designed AZPs** are presented in **App. F**. ❺ **Visualizations of searched architectures** are presented in **App. G**.

**Related Work.** The related work is provided in **App. H**.

## 5 Conclusion and Future Works

In this work, we observe that existing zero-cost proxies in training-free NAS are manually crafted, inefficient, and poorly correlated with final performance. To address this, we propose APD, an LLM-driven framework that automatically discovers high-quality proxies without human effort. By incorporating an actor-critic reinforcement learning strategy, APD iteratively refines prompts to generate better proxies. The proposed method achieves state-of-the-art performance and efficiency on multiple NAS benchmarks. We believe APD offers a novel design paradigm and will inspire broader applications of LLMs in automated machine learning. In the future, we plan to further improve the performance of APD.

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

# A   A Simple Prompt

To empirically validate the hypothesis presented in Section 2, we employ the prompt depicted in Figure 6 to assess the capability of LLMs in generating effective ZCPs.

Figure 6: A representative minimal prompt used in our naive experiments.

**Prompt for Naive**

Please help me design **five novel Zero-Cost Proxies** for evaluating the **expressive performance** of a neural network on a given batch of image data using **training-free metrics**.

These proxies should require **no training**, and must be computationally cheap to evaluate, similar to existing zero-cost proxy methods such as **NWOT**, **SynFlow**, or **Jacobian Covariance**. However, you should not **copy or imitate** these methods — your goal is to **creatively propose original ideas** that align with the following requirements:

For each of the **five proxies**, do the following:
**Firstly**, provide a **one-paragraph description** of the idea, including how the score is computed.
**Secondly**, Implement the proxy in **Python**, as a function named 'evaluate(model, data)' that: Takes a neural network object ('model')(pytorch model) and image batch ('data')(tensor) as input. Returns a scalar score ('score') reflecting expressive performance.

**Note**: Do **not** use training. Avoid random sampling or stochastic components. The heuristic must work on general-purpose image data and standard neural networks (e.g., CNNs). **Do not** provide any additional explanation outside of the required description and code.

# B   Experimental settings

## B.1   Evolution Settings

Table 8 summarizes the default hyper-parameter configuration used in APD evolutionary experiments. Searches are conducted on a single RTX4090 GPU with a fixed random seed of 0. Each candidate proxy's performance is averaged over 5 independent forward passes using a batch of 16 random samples. We adopt CIFAR-10 as the primary benchmark and report transfer results on CIFAR-10 and ImageNet-16. The actor-critic controller is a two layer MLP with 256 hidden units. The actor and critic learning rates are set to 1e-3 and 1e-2, respectively, with a discount factor $\gamma = 0.9$. During search we run 10 episodes of 100 steps search and test every individual for 200 architectures. The controller retains a history window of 5.

Table 8: Hyper-parameters for Automatic Proxy Discovery.

| Parameter | Value | Parameter | Value |
|---|---|---|---|
| Episodes | 10 | Steps | 100 |
| History windows | 5 | Discount factor | 0.9 |
| Actor learning rate | 1e-3 | Critic learning rate | 1e-2 |
| Input batch size | 16 | Repeats | 5 |
| Hidden size | 256 | Number of layers | 2 |
| Population size | 5 | $\beta$ | 1 |

## B.2   DARTS Settings

We list the training hyper-parameters we adopt for all experiments on DARTS. Figure 27 shows the optimal architecture obtained by APD in the DARTS search space.

As detailed in Table 9, models are trained for 600 epochs with a fixed random seed of 42. We employ SGD with momentum 0.9, an initial learning rate of 0.025 that is decayed to zero via cosine annealing, weight decay of 5e-4, and gradient clipping at a maximum norm of 5. Architectures begin with 36

Table 9: Hyper-parameters for DARTS training.

| Parameter | Value | Parameter | Value |
|---|---|---|---|
| Seed | 42 | Batch size | 96 |
| CutOut length | 16 | Initial channels | 36 |
| Cells / layers | 20 | Aux. loss weight | 0.4 |
| Learning rate | 0.025 | Momentum | 0.9 |
| Weight decay | 5e-4 | Grad clip (norm) | 5 |
| DropPath prob. | 0.2 | Lr scheduler | CosineAnnealing |

channels and stack 20 cells. An auxiliary classifier is attached with a loss weight of 0.4. We apply CutOut regularization (length=16) and linearly increase DropPath probability 0.2 over the course of training. All runs use a batch size of 96.

### B.3 Autoformer Settings

For all ImageNet-1k training runs, we adhere to the core setup prescribed by AutoFormer [69], which is listed in Table 10. The subnet discovered by APD are reported in Figure 28, Figure 29 and Figure 30.

We retrain discovered AutoFormer architectures for 300 epochs on ImageNet-1k using AdamW with an initial learning rate of 5e-4, cosine decay to 1e-5, and a 5 epoch warm-up. The network consumes 224×224 images split into 16×16 patches, and optimization runs with a batch of 128. Regularization follows the AutoFormer baseline. RandAugment (9 transformations, magnitude 0.5), mixup-cutmix blending ($\alpha = 0.8$, switch probability 0.5), random erasing (probability 0.25, pixel model), and a linearly ramped drop-path of 0.1. All experiments use the retrain mode, thereby only the chosen subnet's weights are updated while the supernet is frozen.

Table 10: Hyper-parameter configuration used to retrain the subnet on ImageNet-1K.

| Category | Parameter | Value | Notes |
|---|---|---|---|
| Data | Input resolution | $224 \times 224$ | — |
| | Patch size | 16 | ViT patch size |
| | Batch size | 128 | — |
| Optimiser | Optimizer | AdamW | $\beta = (0.9, 0.999), \epsilon = 10^{-8}$ |
| | Initial LR | 5e-4 | Scaled by batch size / 512 |
| | Weight decay | 0.05 | Decoupled |
| | LR schedule | Cosine | 5 warm-up + 300 epochs |
| Regularisation | Drop-path rate | 0.10 | Linearly increased |
| | RandAugment | `rand-m9-mstd0.5-inc1` | Timm default |
| | Mixup / CutMix | $\alpha$=0.8, prob. = 1 | Switch prob. = 0.5 |
| Model | Training mode | *retrain* | Only subnet weights updated |
| | Max rel. position | 14 | Bias radius |

### B.4 LLMs Settings

For proxy generation, we interface with GPT-4o via the Chat Completion API [2] and evaluate APD across multiple LLMs, including Claude 3.7 [4], Grok 3 [66], Gemini 2.0 Flash [21], Deepseek V3 [38], and Qwen Plus [51]. To balance diversity and syntactic correctness, APD is performed with a temperature of 0.5 and the output length is capped at 8192 tokens. We supply a specialized system prompt that constrains LLM to emit JSON object with two fields: thought + code.

## C Prompt Engineering

In this section, we outline the guiding principles underlying the design of our prompts. Given that the effectiveness of LLM-driven heuristic discovery critically hinges on prompt quality, our prompts are carefully crafted to maintain clarity, specificity, and structured output. Particularly, we ensure clarity and specificity by precisely defining the computational boundaries and structural properties of

candidate proxies within each prompt, clearly delineating the allowable search space of the metrics to facilitate efficient exploration by the LLM. Inspired by previous auto proxy discovery research, we prompt the LLM to construct compositional proxies represented as directed acyclic graphs that integrate network-derived signals (e.g., gradients, activations, or weights) with basic statistical and arithmetic operations. Collectively, this rigorous prompt engineering strategy significantly enhances the efficiency, validity, and innovativeness of the ZCPs by APD.

Moreover, through extensive empirical experimentation, we observe that the initial code representation of a generated ZCP typically fails to fully realize its underlying conceptual potential. Specifically, the initial instantiation of the proxy, directly produces from a single prompt, often exhibits suboptimal correlation with the final performance metrics. It is only after applying evolutionary strategies such as mutation and crossover operations that iteratively refine the proxy structure and composition that the intrinsic effectiveness of the proxy becomes evident. This empirical insight underscores the necessity of iterative exploration and refinement within the APD, highlighting mutation and crossover as critical mechanisms for achieving robust and effective ZCPs.

### C.1 Prompt For NAS-Bench-201

We provide the detailed prompts employed in our experiments on NAS-Bench-201 [18]. These prompts serve as explicit instructions guiding the LLM to systematically generate novel ZCPs. Specifically, we design three variants of prompts as depicted in Figure 7, 8, and 9. Each prompt variant progressively introduces structured definitions, constraints on computational operations, and explicit guidelines to ensure that the generated proxies adhere to our framework requirements, thereby facilitating effective exploration of the proxy search space and enhancing the quality of the discovered ZCPs.

### C.2 Prompt For TransNAS-Bench-101

We detail the specific prompts utilized for proxy discovery experiments on TransNAS-Bench-101 [20]. These prompts are carefully crafted to instruct the large language model (LLM) to produce novel zero-cost proxies tailored specifically for their diverse tasks, including JIGSAW, autoencoding, scene classification, and object classification. To accommodate these distinct downstream tasks, we designed three prompt variants as shown in Figure 10, 11 and 12.

### C.3 Prompt For AutoFormer

We present the detailed prompts employed for proxy discovery experiments conducted on the AutoFormer architecture search space. Given AutoFormer's specific structural characteristics and its transformer-based design for vision tasks, we carefully tailor three distinct prompt variants following previous designs in Figure 13, 14 and 15.

## D Extended Experimental Results

### D.1 Additional Ablation Studies

**Analysis on Mutation and Crossover operations** To figure out the roles of the mutation and crossover operations, we evaluate the individual contribution of the mutation and crossover operation by selectively disabling each component during proxy search and observing the resulting performance changes. Figure 11 illustrates that omitting the mutation prompt reduces the proxy's correlation with ground-truth metrics while removal of the crossover operation narrows the architectural exploration space and weakens proxy accuracy. These findings confirm that mutation and crossover provide distinct but complementary functions. collectively enabling the effective of high-quality zero-cost proxies.

**Analysis on population size** To understand how the size of the evolving population affects the quality and convergence of our zero-cost proxy search, we perform an ablation study over three different population sizes: 1, 2, 3, 4, 5 and 10. A larger population can offer richer diversity and robustness but incurs greater computational overhead and makes it harder for the LLM to follow context, thereby reducing ZCPs quality.In contrast, a smaller population accelerates each generation

Figure 7: Prompt used for intialization in NAS-Bench-201.

**Prompt for Initialization**

**Your Task**: Please design **5 Zero-Cost Proxies** to evaluate the representation capability of different convolutional network architectures on a given dataset. The final goal of each proxy is to provide a **scalar score** that reflects the performance of a given network. You should generate Zero-Cost Proxies according to the following specifications, which include: **proxy requirements**, **description of the proxy search space** (defining the structure of how the proxy is computed), **the search space of the networks being evaluated**, the **given dataset**, and the **output format**, as described below:

**Proxy Requirements**:
**Training-free**: No gradient descent, weight updates, or learned parameters
**Efficient**: Low computational cost, suitable for early-stage model selection
**Dxeterministic**: No stochastic elements or randomness
**Input-aware**: The proxy should utilize a batch of input image data
**Model-sensitive**: The proxy should reflect meaningful differences between models.

**Proxy Search Space**: The computation of the proxy can be represented as a **directed acyclic graph (DAG)**. For any given node, its in-degree must satisfy $0 \leq degree_{in} \leq 2$.
Nodes with only outgoing edges and no incoming edges are called **inputs**. Inputs consist of two parts: the input module and the input property. The input module refers to different types of network layers (e.g., convolutional layers, BatchNorm, activation layers, etc.), and the input property is one of the following: gradient $G$, weights $W$, or output feature maps $Z$. An input is defined as a combination of these two components. For example, (convolutional layer + gradient $G$) refers to the gradients of the convolutional layer's parameters. A proxy can have multiple inputs.
Nodes with both incoming and outgoing edges are referred to as **operations**, which are further divided into two types:

- When $degree_{in} = 1$, the node typically performs a **statistical operation** such as computing the mean or standard deviation of the preceding node's output.

- When $degree_{in} = 2$, the node typically performs a **binary operation**, such as addition, subtraction, or division between two inputs.

- Nodes with only incoming edges are **outputs**. There must be **only one output node** in the DAG. The input to this final node must be a **scalar**.

**Network Search Space**: Networks drawn from NASBench-201, varying only in cell structure. Macro architecture: input → conv → cell_n → residual (stride=2) → cell_n → residual (stride=2) → cell_*n → global average pooling. Cells: DAGs of four nodes with operations: zeroize, skip-connection, 1×1 conv, 3×3 conv, 3×3 avg-pool.

**Given Dataset**: CIFAR-10 classification with input shape (3×32×32) and 10-dimensional output.

**Output**: For each proxy, provide a **Description** paragraph and **Code** demo implementing `evaluate(model, inputs, targets)` in PyTorch. Ensure numerical stability, deterministic outputs, and move any returned tensors to CPU.

**Note**: Your Zero-Cost Proxies must not be the same as any existing ones (such as nwot, snip, etc.), though you may draw inspiration from them. Ensure numerical stability in your functions (avoid inf or nan during computation). Do not provide any additional explanation outside of the required description and code.

but risks premature convergence. We fix all other settings and report the average Spearman correlation between proxy scores in Table 12.

**Analysis on hyperparameter** To assess how the discount factor $\gamma$ and the history window size affect proxy quality, we conduct two separate ablations, one varying $\gamma$ over $\{0.5, 0.7, 0.9, 0.99\}$, and the other varying the history window over $\{1, 3, 5, 10\}$. All other settings remain fixed. Results are reported in Table 13 and Table 14.

**Analysis on actor-critic** To evaluate the effect of varying the number of hidden layers in the actor-critic controller on proxy search performance and convergence speed, we conduct an ablation over depths of 2, 3, 4, and 5 layers. Table 15 reports the final Spearman correlation achieved after

Figure 8: Prompt used for mutation in NAS-Bench-201.

---

**Prompt for Mutation**

**Your Task**: Building upon the initial set of **Zero-Cost Proxies**, you will now perform a **mutation** step to generate novel and distinct proxies. You should introduce creative variations into existing proxies through systematic modifications. Follow the mutation rules and constraints defined below to design **5 distinct mutated proxies**:

**Mutation Rules**: When mutating a proxy, you must choose at least **one** and up to **four** of the following mutations to apply:

- **Input Mutation**: Replace, add, or remove one type of input (e.g., switch from convolutional layer gradients (G) to activation outputs (Z)). Consider using less commonly utilized inputs such as BatchNorm statistics or residual block outputs.

- **Operation Mutation**: Change statistical operations (mean, variance, max, min) to another statistic. Substitute binary operations (+, -, , /) with different arithmetic operations, ensuring numerical stability.

- **Structure Mutation**: Adjust the computational graph by altering the connections (edges) between nodes, ensuring the DAG property remains valid. Introduce an additional intermediate node for richer representations.

- **Aggregation Mutation**: Modify how multiple layer values are aggregated (e.g., sum, mean, weighted average).

**Mutated Proxy Search Space**:

- Each node has an in-degree $0 \leq degree_{in} \leq 2$.
- Exactly **one output node** producing a scalar value.
- Inputs: Combinations of modules (conv, BatchNorm, activation layers, residual block outputs) and properties (gradients $G$, weights $W$, outputs $Z$).
- Operations: Statistical (mean, variance, max, min, sum) or binary (+, -, , /).

**Network Search Space**: Networks drawn from NASBench-201, varying only in cell structure. Macro architecture: input → conv → cell_n → residual (stride=2) → cell_n → residual (stride=2) → cell_*n → global average pooling. Cells: DAGs of four nodes with operations: zeroize, skip-connection, 1×1 conv, 3×3 conv, 3×3 avg-pool.

**Given Dataset**: CIFAR-10 classification with input shape (3×32×32) and 10-dimensional output.

**Output**: For each proxy, provide a **Description** paragraph and **Code** demo implementing `evaluate(model, inputs, targets)` in PyTorch. Ensure numerical stability, deterministic outputs, and move any returned tensors to CPU.

**Note**: Do **not** use training. Avoid random sampling or stochastic components. The heuristic must work on general-purpose image data and standard neural networks (e.g., CNNs). **Do not** provide any additional explanation outside of the required description and code.

**Existing Proxies**:
<Existing Proxies>

---

convergence and the generation at which the search stabilized for each depth. The results indicate that evolution planning itself is relatively straightforward. A two-layer actor-critic controller already achieves convergence within 50 generations. Employing additional layers results in an increased number of ineffective generations, incurring unnecessary computational overhead without further improving performance (e.g., in the later stages of evolution when proxy quality improvements become minimal, continuing with fixed initialization procedures can result in diminishing returns, incurring unnecessary computational overhead).

Figure 9: Prompt used for crossover in NAS-Bench-201.

**Prompt for Crossover**

**Your Task**: Expanding upon the previously defined Zero-Cost Proxies, your next step is to implement a **crossover** operation to create novel proxies. The crossover process involves combining aspects of **two parent proxies** to produce **5 distinct crossover proxies**. Adhere to the crossover rules and constraints defined below:

**Crossover Rules**: To generate crossover proxies, select two parent proxies from previously defined proxies. Apply at least **one** and up to **two** of the following crossover techniques:

- **Input Crossover**: Exchange input nodes between parent proxies, such as combining gradients (G) from one proxy and outputs (Z) from another.
- **Operation Crossover**: Merge operation nodes by replacing a statistical or binary operation from one parent with an operation from the other parent.
- **Structure Crossover**: Blend the computational graph structures of two proxies by interchanging node connections, ensuring the resulting DAG remains valid.
- **Aggregation Crossover**: Combine aggregation strategies (e.g., sum, mean, weighted average) from the two parent proxies.

**Crossover Proxy Search Space**:

- Nodes must have an in-degree $0 \leq degree\_in \leq 2$.
- Exactly **one output node** producing a scalar value.
- Inputs: combinations of modules (conv, BatchNorm, activation layers, residual block outputs) and properties (gradients $G$, weights $W$, outputs $Z$).
- Operations: statistical (mean, variance, max, min, sum) or binary (+, -, , /).

**Network Search Space**: Networks drawn from NASBench-201, varying only in cell structure. Macro architecture: input → conv → cell_n → residual (stride=2) → cell_n → residual (stride=2) → cell_*n → global average pooling. Cells: DAGs of four nodes with operations: zeroize, skip-connection, 1×1 conv, 3×3 conv, 3×3 avg-pool.

**Output**: For each proxy, provide a **Description** paragraph and **Code** demo implementing `evaluate(model, inputs, targets)` in PyTorch. Ensure numerical stability, deterministic outputs, and move any returned tensors to CPU.

**Note**: Do **not** use training. Avoid random sampling or stochastic components. The heuristic must work on general-purpose image data and standard neural networks (e.g., CNNs). **Do not** provide any additional explanation outside of the required description and code.

**Existing Proxies**:
<Existing Proxies>

## D.2 Detailed Cross-LLM Performance

We track the evolution of average Spearman correlation over 20 generations for each LLM backbone under. Table 16 reports the per-generation correlation on NAS-Bench-201 for all seven models.

## D.3 AutoFormer Accuracy Results

We evaluate the architectures discovered by APD in the AutoFormer search space on ImageNet–1k classification. Table 17 reports the Top-1 accuracies for the tiny, small, and base variants: the tiny model achieves 76.1 %, the small model 81.5 %. These results demonstrate that our search not only produces proxies with high correlation to ground truth but also yields architectures that deliver competitive performance on a large-scale vision benchmark.

Figure 10: Prompt for initialization in TransNAS-Bench-101.

**Prompt for Initialization**

**Your Task**: Please design **5 novel Zero-Cost proxies** to evaluate the representation capability of different convolutional network architectures on a given downstream task. The final goal of each proxy is to provide a **scalar score** that reflects the performance of a given network. You should generate Zero-Cost Proixes according to the following specifications, which include: **proxy requirements**, **description of the proxy search space** (defining the structure of how the proxy is computed), the **search space of the networks being evaluated**, the **given dataset**, and the **output format**, as described below:

**Proxy Requirements**:

- **Training-free**: No gradient descent, weight updates, or learned parameters.

- **Efficient**: Low computational cost, suitable for early-stage model selection.

- **Deterministic**: No stochastic elements or randomness.

- **Model-sensitive**: The proxy should reflect meaningful differences between models.

**Proxy Search Space**: The computation of the proxy can be represented as a **directed acyclic graph (DAG**. For any given node, its in-degree must satisfy $0 \leq degree_{in} \leq 2$.

Nodes with only outgoing edges and no incoming edges are called **inputs**. Inputs consist of two parts: the input module and the input property. The input module refers to different types of network layers (e.g., convolutional layers, BatchNorm, activation layers, etc.), and the input property is one of the following: gradient $G$, weights $W$, or output feature maps $Z$. An input is defined as a combination of these two components. For example, (convolutional layer + gradient $G$) refers to the gradients of the convolutional layer's parameters. A proxy can have multiple inputs.

Nodes with both incoming and outgoing edges are referred to as **operations**, which are further divided into two types:

- When $degree_{in} = 1$, the node typically performs a **statistical operation** such as computing the mean or standard deviation of the preceding node's output.

- When $degree_{in} = 2$, the node typically performs a **binary operation**, such as addition, subtraction, or division between two inputs.

Nodes with only incoming edges are **outputs**. There must be **only one output node** in the DAG. The input to this final node must be a **scalar**.

**Network Search Space**: The networks to be evaluated come from **TransNAS-Bench-101**, which includes variations in both **cell-level** and **macro-level structures**.

**Macro architecture**: img → searched backbone (composed of stacked cells) → task-specific decoder. The macro structure consists of **three stages**, each containing **modules** made of **residual blocks**. After each stage, downsampling and channel doubling occur. For example: Stage 1 (Module 1) → residual blocks → Stage 2 (Module 2, 3) → residual blocks → Stage 3 (Module 4) → residual blocks → task-specific head.

**Cell structure**: Each cell is modeled as a **directed acyclic graph (DAG)** with six nodes. For any $v_i, v_j \in V$, if $i < j$, then $e_{ij} \in E$. Each node represents a latent feature tensor, and each edge represents a candidate operation from the following set:

- zeroize

- skip-connection

- 1 × 1 convolution

- 3 × 3 convolution

Each cell forms the base unit of the backbone and is repeatedly stacked according to the macro-level configuration. This enables flexible network designs across different tasks such as object classification, semantic segmentation, surface normal estimation, and more.

Given Downstream Task:
<Given Downstream Task>

Figure 11: Prompt for mutation in TransNAS-Bench-101.

**Prompt for Mutation**

**Your Task**: Building upon the initial set of **Zero-Cost Proxies**, you will now perform a **mutation** step to generate novel and distinct proxies. You should introduce creative variations into existing proxies through systematic modifications. Follow the mutation rules and constraints defined below to design **5 distinct mutated proxies**:

**Mutation Rules**: When mutating a proxy, you must choose at least **one** and up to **four** of the following mutations to apply:

- **Input Mutation**: Replace, add, or remove one type of input (e.g., switch from convolutional layer gradients (G) to activation outputs (Z)). Consider using less commonly utilized inputs such as BatchNorm statistics or residual block outputs.

- **Operation Mutation**: Change statistical operations (mean, variance, max, min) to another statistic. Substitute binary operations (+, -, , /) with different arithmetic operations, ensuring numerical stability.

- **Structure Mutation**: Adjust the computational graph by altering the connections (edges) between nodes, ensuring the DAG property remains valid. Introduce an additional intermediate node for richer representations.

- **Aggregation Mutation**: Modify how multiple layer values are aggregated (e.g., sum, mean, weighted average).

**Proxy Search Space**: The computation of the proxy can be represented as a **directed acyclic graph (DAG**. For any given node, its in-degree must satisfy $0 \leq degree_{in} \leq 2$.

Nodes with only outgoing edges and no incoming edges are called **inputs**. Inputs consist of two parts: the input module and the input property. The input module refers to different types of network layers (e.g., convolutional layers, BatchNorm, activation layers, etc.), and the input property is one of the following: gradient $G$, weights $W$, or output feature maps $Z$. An input is defined as a combination of these two components. For example, (convolutional layer + gradient $G$) refers to the gradients of the convolutional layer's parameters. A proxy can have multiple inputs.

Nodes with both incoming and outgoing edges are referred to as \*\*operations\*\*, which are further divided into two types:

- When $degree_{in} = 1$, the node typically performs a **statistical operation** such as computing the mean or standard deviation of the preceding node's output.

- When $degree_{in} = 2$, the node typically performs a **binary operation**, such as addition, subtraction, or division between two inputs.

Nodes with only incoming edges are **outputs**. There must be **only one output node** in the DAG. The input to this final node must be a **scalar**.

**Network Search Space**: The networks to be evaluated come from **TransNAS-Bench-101**, which includes variations in both **cell-level** and **macro-level structures**.

**Macro architecture**: img → searched backbone (composed of stacked cells) → task-specific decoder. The macro structure consists of **three stages**, each containing **modules** made of **residual blocks**. After each stage, downsampling and channel doubling occur. For example: Stage 1 (Module 1) → residual blocks → Stage 2 (Module 2, 3) → residual blocks → Stage 3 (Module 4) → residual blocks → task-specific head.

**Cell structure**: Each cell is modeled as a **directed acyclic graph (DAG)** with six nodes. For any $v_i, v_j \in V$, if $i < j$, then $e_{ij} \in E$. Each node represents a latent feature tensor, and each edge represents a candidate operation from the following set:

- zeroize

- skip-connection

- $1 \times 1$ convolution

- $3 \times 3$ convolution

Each cell forms the base unit of the backbone and is repeatedly stacked according to the macro-level configuration. This enables flexible network designs across different tasks such as object classification, semantic segmentation, surface normal estimation, and more.

Given Downstream Task:
<Given Downstream Task>

Figure 12: Prompt for crossover in TransNAS-Bench-101.

**Prompt for Crossover**

**Your Task**: Expanding upon the previously defined Zero-Cost Proxies, your next step is to implement a **crossover** operation to create novel proxies. The crossover process involves combining aspects of **two parent proxies** to produce **5 distinct crossover proxies**. Adhere to the crossover rules and constraints defined below:

**Crossover Rules**: To generate crossover proxies, select two parent proxies from previously defined or mutated proxies. Apply at least **one** and up to **two** of the following crossover techniques:

- **Input Crossover**: Exchange input nodes between parent proxies, such as combining gradients (G) from one proxy and outputs (Z) from another.

- **Operation Crossover**: Blend the computational graph structures of two proxies by interchanging node connections, ensuring the resulting DAG remains valid.

- **Structure Crossover**: Blend the computational graph structures of two proxies by interchanging node connections, ensuring the resulting DAG remains valid.

- **Aggregation Crossover**: Combine aggregation strategies (e.g., sum, proxies.

**Proxy Search Space**: The computation of the proxy can be represented as a **directed acyclic graph (DAG**. For any given node, its in-degree must satisfy $0 \leq degree_{in} \leq 2$.
Nodes with only outgoing edges and no incoming edges are called **inputs**. Inputs consist of two parts: the input module and the input property. The input module refers to different types of network layers (e.g., convolutional layers, BatchNorm, activation layers, etc.), and the input property is one of the following: gradient $G$, weights $W$, or output feature maps $Z$. An input is defined as a combination of these two components. For example, (convolutional layer + gradient $G$) refers to the gradients of the convolutional layer's parameters. A proxy can have multiple inputs.
Nodes with both incoming and outgoing edges are referred to as \*\*operations\*\*, which are further divided into two types:

- When $degree_{in} = 1$, the node typically performs a **statistical operation** such as computing the mean or standard deviation of the preceding node's output.

- When $degree_{in} = 2$, the node typically performs a **binary operation**, such as addition, subtraction, or division between two inputs.

Nodes with only incoming edges are **outputs**. There must be **only one output node** in the DAG. The input to this final node must be a **scalar**.

**Network Search Space**: The networks to be evaluated come from **TransNAS-Bench-101**, which includes variations in both **cell-level** and **macro-level structures**.
**Macro architecture**: img → searched backbone (composed of stacked cells) → task-specific decoder. The macro structure consists of **three stages**, each containing **modules** made of **residual blocks**. After each stage, downsampling and channel doubling occur. For example: Stage 1 (Module 1) → residual blocks → Stage 2 (Module 2, 3) → residual blocks → Stage 3 (Module 4) → residual blocks → task-specific head.
**Cell structure**: Each cell is modeled as a **directed acyclic graph (DAG)** with six nodes. For any $v_i, v_j \in V$, if $i < j$, then $e_{ij} \in E$. Each node represents a latent feature tensor, and each edge represents a candidate operation from the following set:

- zeroize

- skip-connection

- 1 × 1 convolution

- 3 × 3 convolution

Each cell forms the base unit of the backbone and is repeatedly stacked according to the macro-level configuration. This enables flexible network designs across different tasks such as object classification, semantic segmentation, surface normal estimation, and more.

Given Downstream Task:
<Given Downstream Task>

Figure 13: Prompt for initialization in AutoFormer.

**Prompt for Initialization**

**Your Task**: Please design **5 novel Zero-Cost Proxies** to evaluate the representation capability of different transformer network architectures on a given dataset. The final goal of each proxy is to provide a **scalar score** that reflects the performance of a given network. You should generate zero-cost proxies according to the following specifications, which include: **proxy requirements**, **description of the proxy search space** (defining the structure of how the metric is computed), the **search space of the networks being evaluated**, the **given dataset**, and the **output format**, as described below:

**Proxy Requirements**:

- **Training-free**: No gradient descent, weight updates, or learned parameters.
- **Efficient**: Low computational cost, suitable for early-stage model selection.
- **Deterministic**: No stochastic elements or randomness.
- **Model-sensitive**: The proxy should reflect meaningful differences between models.

**Proxy Search Space**: The computation of the proxy can be represented as a **directed acyclic graph (DAG)**. For any given node, its in-degree must satisfy $0 \leq degree_{in} \leq 2$.

Nodes with only outgoing edges and no incoming edges are called **inputs**. Inputs consist of two parts: the input module and the input property. The input module refers to different types of network layers, and the input property is one of the following: gradient $G$, weights $W$, or output feature maps $Z$. An input is defined as a combination of these two components. For example, (convolutional layer + gradient $G$) refers to the gradients of the convolutional layer's parameters. A proxy can have multiple inputs.

Nodes with both incoming and outgoing edges are referred to as **operations**, which are further divided into two types:

- When $degree_{in} = 1$, the node typically performs a **statistical operation** such as computing the mean or standard deviation of the preceding node's output.
- When $degree_{in} = 2$, the node typically performs a **binary operation**, such as addition, subtraction, or division between two inputs.

Nodes with only incoming edges are **outputs**. There must be **only one output node** in the DAG. The input to this final node must be a **scalar**.

**Search Space**: The search space defines a set of possible architectural configurations derived from the supernet. Instead of fixed parameters, it provides discrete choices or ranges for each architectural hyperparameter, enabling the selection or evaluation of optimal subnetworks.

A typical search space includes the following parameters with explicit candidate values:

- **EMBED DIM**: A set of embedding dimensions ($D \in \{D_1, D_2, \ldots, D_n\}$), e.g., ($D \in \{528, 576, 624\}$).
- **NUM HEADS**: Choices for attention heads ($H \in \{H_1, H_2, \ldots, H_m\}$), e.g., ($H \in \{9, 10\}$).
- **MLP RATIO**: Options for the MLP expansion ratio ($R_{mlp} \in \{R_1, R_2, \ldots, R_k\}$), e.g., ($R_{mlp} \in \{3.0, 3.5, 4.0\}$).
- **DEPTH**: Number of transformer blocks ($L \in \{L_1, L_2, \ldots, L_s\}$), e.g., ($L \in \{14, 15, 16\}$).

Formally, the search space $\mathcal{A}$ can be defined as a Cartesian product of discrete hyperparameter sets: $\mathcal{A} = \{(D, H, R_{mlp}, L) \mid D \in \{528, 576, 624\}, H \in \{9, 10\}, R_{mlp} \in \{3.0, 3.5, 4.0\}, L \in \{14, 15, 16\}\}$

**Given Dataset**: The architectures will be evaluated on an image classification task using **IMAGE-NET**. The input image shape is **(3, 224, 224)**, and the network outputs a **1000-dimensional vector**.

**Note**: Your Zero-Cost Proxies must not be the same as any existing ones (such as nwot, snip, etc.), though you may draw inspiration from them. Ensure numerical stability in your functions (avoid inf or nan during computation).

Figure 14: Prompt for mutation in AutoFormer.

**Prompt for Mutation**

**Your Task**: Given **5 existing Zero-Cost proxies**, **mutate** each of these proxies to create **5 new ones** (one mutated version per original proxy). Each mutated proxy should still be **training-free**, **computationally efficient**, **deterministic**, **input-aware**, and **model-sensitive**. The final goal is to produce **5 updated or extended training-free proxies** that remain valid for early-stage model selection but differ in approach from the original set.

When generating each mutated proxy, you may:

- Slightly alter or combine existing nodes in the computation DAG
- Introduce new operations or input types (still abiding by the in-degree constraints)
- Change the nature of the final scalar computation
- Ensure numerical stability (avoid division by zero, log of non-positive numbers, etc.)

**Proxy Requirements**:

- **Training-free**: Do **not** use any form of parameter updates.
- **Efficient**: Maintain low overhead in computation.
- **Deterministic**: Must produce consistent outputs for the same model and data.
- **Model-sensitive**: Should vary meaningfully with different subnetworks sampled from the supernet.

**Proxy Search Space**: All mutated proxies should still form a **directed acyclic graph (DAG)** in their computation. For any given node:

- **In-degree Constraint**: $0 \leq degree_{in} \leq 2$.
- **Input Nodes**: Consist of a network module. Tied to a property ($W$, $G$, or $Z$). You may now choose to introduce new input properties that are still relevant to a training-free scenario (e.g., a logarithm of weights, or a combined dimension of feature maps), but be sure to keep them deterministic and valid.
- **Operation Nodes (single-operand)**: Perform statistical transformations (e.g., mean, variance, norms, etc.).
- **Operation Nodes (binary)**: Perform pairwise operations (e.g., addition, multiplication, ratio).
- **Output Node**: Must be a single scalar value (with no outgoing edges).

**Search Space**: The search space defines a set of possible architectural configurations derived from the supernet. Instead of fixed parameters, it provides discrete choices or ranges for each architectural hyperparameter, enabling the selection or evaluation of optimal subnetworks.

A typical search space includes the following parameters with explicit candidate values:

- **EMBED DIM**: A set of embedding dimensions ($D \in \{D_1, D_2, \ldots, D_n\}$), e.g., ($D \in \{528, 576, 624\}$).
- **NUM HEADS**: Choices for attention heads ($H \in \{H_1, H_2, \ldots, H_m\}$), e.g., ($H \in \{9, 10\}$).
- **MLP RATIO**: Options for the MLP expansion ratio ($R_{mlp} \in \{R_1, R_2, \ldots, R_k\}$), e.g., ($R_{mlp} \in \{3.0, 3.5, 4.0\}$).
- **DEPTH**: Number of transformer blocks ($L \in \{L_1, L_2, \ldots, L_s\}$), e.g., ($L \in \{14, 15, 16\}$).

Formally, the search space $\mathcal{A}$ can be defined as a Cartesian product of discrete hyperparameter sets: $\mathcal{A} = \{(D, H, R_{mlp}, L) \mid D \in \{528, 576, 624\}, H \in \{9, 10\}, R_{mlp} \in \{3.0, 3.5, 4.0\}, L \in \{14, 15, 16\}\}$

**Given Dataset**: The architectures will be evaluated on an image classification task using **IMAGE-NET**. The input image shape is **(3, 224, 224)**, and the network outputs a **1000-dimensional vector**.

**Note**: Your Zero-Cost Proxies must not be the same as any existing ones (such as nwot, snip, etc.), though you may draw inspiration from them. Ensure numerical stability in your functions (avoid inf or nan during computation).

**Existing Proxies**:
<Existing Proxies>

Figure 15: Prompt for crossover in AutoFormer.

**Prompt for Crossover**

**Your Task**: Please design **5 novel Zero-Cost Proxies** to evaluate the representation capability of different transformer network architectures on a given dataset, **using a crossover-based approach**. This means each of your new proxies should be created by **combining at least two distinct ideas, components, or sub-nodes** from hypothetical or existing parent proxies. For instance, if you have two parent proxies A and B with specific inputs or operations, you must form a new child proxy by **merging** relevant parts from each parent, ensuring the final proxy remains logically consistent and provides a **scalar score** that reflects the performance of a given network. You should generate Zero-Cost Proxies according to the following specifications, which include: **proxy requirements**, **description of the proxy search space**, the **search space of the networks being evaluated**, the **given dataset**, and the **output format**, as described below:

**Proxy Requirements**:

- **Training-free**: Do **not** use any form of parameter updates.
- **Efficient**: Maintain low overhead in computation.
- **Deterministic**: Must produce consistent outputs for the same model and data.
- **Model-sensitive**: Should vary meaningfully with different subnetworks sampled from the supernet.
- **Crossover-based**: Each proxy must **inherit or merge** conceptual "genes" (e.g., particular input sources or operations) from **at least two parent proxies**, forming a new, valid proxy design.

**Proxy Search Space**: All mutated proxies should still form a **directed acyclic graph (DAG)** in their computation. For any given node:

- **In-degree Constraint**: $0 \leq degree_{in} \leq 2$.
- **Input Nodes**: Consist of a network module. Tied to a property ($W$, $G$, or $Z$). You may now choose to introduce new input properties that are still relevant to a training-free scenario (e.g., a logarithm of weights, or a combined dimension of feature maps), but be sure to keep them deterministic and valid.
- **Operation Nodes (single-operand)**: Perform statistical transformations (e.g., mean, variance, norms, etc.).
- **Operation Nodes (binary)**: Perform pairwise operations (e.g., addition, multiplication, ratio).
- **Output Node**: Must be a single scalar value (with no outgoing edges).

**Search Space**: The search space defines a set of possible architectural configurations derived from the supernet. Instead of fixed parameters, it provides discrete choices or ranges for each architectural hyperparameter, enabling the selection or evaluation of optimal subnetworks.
A typical search space includes the following parameters with explicit candidate values:

- **EMBED DIM**: A set of embedding dimensions ($D \in \{D_1, D_2, \ldots, D_n\}$), e.g., ($D \in \{528, 576, 624\}$).
- **NUM HEADS**: Choices for attention heads ($H \in \{H_1, H_2, \ldots, H_m\}$), e.g., ($H \in \{9, 10\}$).
- **MLP RATIO**: Options for the MLP expansion ratio ($R_{mlp} \in \{R_1, R_2, \ldots, R_k\}$), e.g., ($R_{mlp} \in \{3.0, 3.5, 4.0\}$).
- **DEPTH**: Number of transformer blocks ($L \in \{L_1, L_2, \ldots, L_s\}$), e.g., ($L \in \{14, 15, 16\}$).

Formally, the search space $\mathcal{A}$ can be defined as a Cartesian product of discrete hyperparameter sets: $\mathcal{A} = \{(D, H, R_{mlp}, L) \mid D \in \{528, 576, 624\}, H \in \{9, 10\}, R_{mlp} \in \{3.0, 3.5, 4.0\}, L \in \{14, 15, 16\}\}$

**Given Dataset**: The architectures will be evaluated on an image classification task using **IMAGE-NET**. The input image shape is **(3, 224, 224)**, and the network outputs a **1000-dimensional vector**.

**Note**: Your Zero-Cost Proxies must not be the same as any existing ones (such as nwot, snip, etc.), though you may draw inspiration from them. Ensure numerical stability in your functions (avoid inf or nan during computation).

Table 11: Average proxy correlation over 20 generations for different prompt operations on NAS-Bench-201. Full denotes the complete operation configuration, incorporating both mutation and crossover.

| Generation | No Crossover | No Mutation | Full |
|---|---|---|---|
| 1 | 0.444 | 0.193 | **0.246** |
| 2 | 0.533 | 0.237 | **0.470** |
| 3 | 0.533 | 0.349 | **0.617** |
| 4 | 0.533 | 0.439 | **0.628** |
| 5 | 0.541 | 0.463 | **0.662** |
| 6 | 0.541 | 0.463 | **0.662** |
| 7 | 0.541 | 0.463 | **0.705** |
| 8 | 0.572 | 0.463 | **0.705** |
| 9 | 0.627 | 0.547 | **0.754** |
| 10 | 0.640 | 0.547 | **0.794** |
| 11 | 0.640 | 0.547 | **0.800** |
| 12 | 0.640 | 0.669 | **0.800** |
| 13 | 0.640 | 0.713 | **0.800** |
| 14 | 0.640 | 0.761 | **0.800** |
| 15 | 0.640 | 0.761 | **0.800** |
| 16 | 0.680 | 0.780 | **0.800** |
| 17 | 0.714 | 0.780 | **0.803** |
| 18 | 0.739 | 0.780 | **0.803** |
| 19 | 0.739 | 0.780 | **0.813** |
| 20 | 0.758 | 0.780 | **0.813** |

Table 12: Effect of population size on the average Spearman correlation of the full prompt configuration over 20 generations.

| Population Size | Avg. Spearman $\rho$ |
|---|---|
| 1 | 0.6023 |
| 2 | 0.6837 |
| 3 | 0.7541 |
| 4 | 0.8013 |
| 5 | **0.8124** |
| 10 | 0.7895 |

# E   Limitations

While APD markedly advances training-free NAS, several caveats remain. We group them here for clarity.

**Black-box optimization** Because APD delegates the generation of every candidate proxy to a LLM, the token-level reasoning that maps a prompt to executable code is not observable, raising black-box concerns. We acknowledge that the LLM itself remains opaque. Nevertheless, APD is designed so that its outer optimization loop is inspectable. The LLM's outputs are evaluated by a public fitness function, and the actor–critic scheduler is trained only on this scalar reward. All decisions that influence the search trajectory therefore pass through a measurable, task-specific signal rather than hidden logits. Thus, while some inner LLM reasoning remains inaccessible, the information APD exposes is already adequate for transparency and troubleshooting, so the residual opacity of LLM is unlikely to undermine the method's value.

**Need for a small ground-truth subset** Computing fitness requires baseline accuracies for roughly 2-3% of each benchmark, which is standard in NAS studies and inexpensive compared with full training. Although a small ground-truth subset is indispensable for APD, we have empirically shown that the ZCPs obtained on one such subset continue to generalize when transferred to search space with no significant differences. Consequently, even for previously unseen NAS tasks, the overhead introduced by APD remains minimal and well within acceptable bounds.

**Prompt-context length growth on very large searches** The generator receives at most $C_t$ proxies per step, and both prompt and context are explicitly bounded to fit the LLM window. Empirically the

Table 13: Effect of discount factor $\gamma$ on average Spearman correlation. Best value is highlighted in yellow.

| Discount Factor $\gamma$ | Avg. Spearman $\rho$ |
|:---:|:---:|
| 0.50 | 0.7124 |
| 0.70 | 0.7823 |
| 0.90 | **0.8137** |
| 0.99 | 0.8015 |

Table 14: Effect of history window size on average Spearman correlation. Best value is highlighted in yellow.

| History Window | Avg. Spearman $\rho$ |
|:---:|:---:|
| 1 | 0.6987 |
| 3 | 0.8042 |
| 5 | **0.8259** |
| 10 | 0.7791 |

combined length stayed under 6k tokens in specific population size, and we have to down-sample the context if future domains require longer code.

**Security and stability when executing LLM-generated code** Each proxy is arbitrary code and could, in principle, issue unsafe or resource-hungry calls. Therefore, code has to be run inside a resource-capped sandbox.

# F Discovered AZPs

## F.1 Proxy for NAS-Bench-201

To demonstrate the effectiveness of our approach, we present three representative proxies that achieved strong performance on NAS-Bench-201, which are illustrated in Figures 16, 17, and 18.

## F.2 Proxy for TransNAS-Bench-101

We present the zero-cost proxies discovered on TransNAS-Bench-101 by APD in Figure 19, 20, and 21.

## F.3 Proxy for AutoFormer

In Figure 22, 23, and 24, we present the zero-cost proxies discovered through APD on AutoFormer.

# G Searched Architectures

In the DARTS search space, the optimal architecture discovered by APD is shown in Figure 27, with its normal and reduction cells visualized in Figures 25 and 26, respectively. In the AutoFormer setting, the architectures found for the tiny, small, and base variants are presented in Figure 28, 29, and 30, respectively.

# H Related Work

## H.1 Neural Architecture Search

Neural Architecture Search (NAS) aims to seek high-performance neural networks in an automatic manner. Early works cast the problem as black-box reinforcement learning or evolutionary optimization. NASNet [77] and AmoebaNet [53] trained thousands of architectures from scratch on large GPU clusters, demonstrating the promise of search but at prohibitive cost. Subsequent one-shot methods

Table 15: Impact of A2C network depth on proxy search quality and convergence.

| Hidden Layers | Final Spearman $\rho$ | Convergence Generation |
|---|---|---|
| 2 | 0.8216 | 42 |
| 3 | 0.8367 | 79 |
| 4 | 0.8175 | 96 |
| 5 | 0.8230 | 103 |

Table 16: Evolution of average Spearman correlation over 20 generations on NAS-Bench-201 for different LLM backbones.

| Gen. | Claude 3.7 | Gemini 2.0 Flash | GPT-4o | Deepseek V3 | Qwen Plus | Grok 3 | Llama4 |
|---|---|---|---|---|---|---|---|
| 1 | 0.0000 | 0.3460 | 0.0989 | 0.0853 | 0.0228 | 0.1293 | 0.0000 |
| 2 | 0.5040 | 0.4290 | 0.1091 | 0.1720 | 0.0732 | 0.1293 | 0.1319 |
| 3 | 0.5040 | 0.5027 | 0.1816 | 0.1720 | 0.1292 | 0.1293 | 0.1319 |
| 4 | 0.5040 | 0.5560 | 0.3909 | 0.1720 | 0.3230 | 0.1293 | 0.1319 |
| 5 | 0.6553 | 0.5560 | 0.4312 | 0.2126 | 0.3807 | 0.1293 | 0.1319 |
| 6 | 0.7087 | 0.6033 | 0.4312 | 0.2126 | 0.3807 | 0.1293 | 0.1319 |
| 7 | 0.7304 | 0.6212 | 0.5036 | 0.3189 | 0.3962 | 0.2669 | 0.1319 |
| 8 | 0.7304 | 0.6714 | 0.6028 | 0.4923 | 0.5244 | 0.2669 | 0.1319 |
| 9 | 0.7304 | 0.6714 | 0.6278 | 0.4969 | 0.5518 | 0.2669 | 0.1319 |
| 10 | 0.7304 | 0.6714 | 0.6806 | 0.6531 | 0.6109 | 0.4071 | 0.1605 |
| 11 | 0.7713 | 0.6869 | 0.7411 | 0.6531 | 0.6303 | 0.6933 | 0.1605 |
| 12 | 0.7713 | 0.6869 | 0.7411 | 0.6531 | 0.6303 | 0.6933 | 0.1605 |
| 13 | 0.7713 | 0.6869 | 0.7744 | 0.7944 | 0.6303 | 0.6933 | 0.2809 |
| 14 | 0.7713 | 0.6986 | 0.7898 | 0.7944 | 0.6303 | 0.7081 | 0.2809 |
| 15 | 0.7728 | 0.7317 | 0.7898 | 0.7944 | 0.6439 | 0.7395 | 0.3011 |
| 16 | 0.7728 | 0.7317 | 0.8090 | 0.8024 | 0.6439 | 0.7395 | 0.4225 |
| 17 | 0.8114 | 0.7317 | 0.8090 | 0.8024 | 0.6994 | 0.7395 | 0.4348 |
| 18 | 0.8114 | 0.7517 | 0.8090 | 0.8024 | 0.7212 | 0.7395 | 0.5598 |
| 19 | 0.8114 | 0.7517 | 0.8090 | 0.8024 | 0.7212 | 0.7697 | 0.6495 |
| 20 | 0.8114 | 0.7522 | 0.8110 | 0.8024 | 0.7212 | 0.7697 | 0.6973 |

reduce cost by sharing parameters into a single supernet. ENAS [75] introduces weight sharing with a recurrent controller, while DARTS [40] relaxes the discrete search space to a differentiable mixture, enabling gradient-based updates but inheriting optimization bias and collapse issues that spawned a line of robust variants. However, performance estimation remains the key throughput bottleneck [44].

## H.2 Zero-Cost Proxies

NAS traditionally relies on partial or full training to judge candidate models, making large-scale exploration prohibitive. Zero-Cost Proxies (ZCPs) address this bottleneck by estimating an architecture's potential from its randomly initialized weights in significantly less time and constitute a standard component of training-free NAS. The earliest ZCPs exploit gradient saliency. SNIP [34] evaluates weights with the first-order loss sensitivities, GraSP [1] extends the idea to second-order information, and SynFlow [1] removes data dependence by propagating an all-ones input, yielding strong correlations in deep convolutional spaces. A second strand replaces gradients with cheap statistics of activations or parameter layouts. NWOT [44] measures neural-tangent-kernel overlap, Zen-score [37] evaluates Jacobian log-determinants, while ZiCo [35] shows that even simple norms such as parameter count or FLOPs can outperform many hand-crafted heuristics once appropriately normalized.

Despite steady progress, two limitations persist. Firstly, all existing proxies are manually specified. Their functional form rarely transfers intact across tasks (e.g., from NAS-Bench-201 CIFAR10 to TransNAS-Bench-101 autoencoding). Secondly, correlation improvements have largely plateaued, and gains often come at the cost of additional forward passes or Jacobian computations.

## H.3 Automatic Proxy Discovery

Automatic Proxy Discovery aims to relieve experts from hand-crafting ZCPs by treating the proxy itself as the search object. The most systematic effort so far is Auto-Prox [64]. It encodes a proxy as a computation graph whose nodes are primitive arithmetic or operations (e.g., exp, log). An evolutionary algorithm with elitism preservation mutates these graphs to maximize a joint correlation

Table 17: The comparison results on the Autoformer search space in the ImageNet dataset. ∗ denotes the results reported by [28].

| Models | #Param (M) | FLOPS (B) | Top-1 (%) | Top-5 (%) | Model Type | Design Type | Years | GPU Days |
|---|---|---|---|---|---|---|---|---|
| **Tiny search space** | | | | | | | | |
| ResNet-18∗ [26] | 11.7 | 1.8 | 72.5 | - | CNNs | Manual | 2015 | - |
| MobileNet-V3 [54] | 5.5 | - | 75.2 | - | CNNs | Manual | 2015 | - |
| Deit-Ti [59] | 5.7 | 1.2 | 72.2 | 91.1 | Transformer | Manual | 2015 | - |
| TNT-Ti [24] | 6.1 | 1.4 | 73.9 | 91.9 | Transformer | Manual | 2015 | - |
| ViT-Ti [19] | 5.7 | - | 74.5 | - | Transformer | Manual | ICLR2020 | - |
| CPVT-Ti [17] | 6.0 | - | 74.9 | 92.6 | Transformer | Manual | 2015 | - |
| PVT-Tiny [62] | 13.2 | 1.9 | 75.1 | - | Transformer | Manual | 2015 | - |
| AutoFormer-Ti [11] | 5.7 | 1.3 | 74.7 | 92.6 | Transformer | Auto | CVPR2021 | 24 |
| GLiT-Ti [10] | 7.2 | 1.4 | 76.3 | - | Hybrid | Auto | ICCV2021 | N/A |
| ViTAS-C [56] | 5.6 | 1.3 | 74.7 | 91.6 | Transformer | Auto | ECCV2022 | 32 |
| TF-TAS-Ti [76] | 5.9 | 1.4 | 75.3 | 92.8 | Transformer | Auto | CVPR2022 | 0.5 |
| Auto-Prox [64] | 6.4 | - | 75.6 | - | Transformer | Auto | AAAI2024 | 0.1 |
| AZ-NAS [33] | 6.2 | 1.4 | 76.4 | - | Transformer | Auto | CVPR2024 | 0.04 |
| CET-TAS (Ours) | 8.4 | 1.9 | 76.1 | 93.1 | Transformer | Auto | - | 0.25 |
| **Small search space** | | | | | | | | |
| ResNet-50∗ [26] | 25.6 | 4.1 | 80.2 | - | CNNs | Manual | 2015 | - |
| RegNetY-4GF [27] | 20.6 | - | 79.4 | - | CNNs | Manual | 2015 | - |
| DeiT-S [59] | 22.1 | 4.7 | 79.9 | 95.0 | Transformer | Manual | 2015 | - |
| ViT-S/16 [19] | 22.1 | 4.7 | 78.8 | - | Transformer | Manual | 2015 | - |
| PVT-Small [62] | 24.5 | 3.8 | 79.8 | - | Transformer | Manual | 2015 | - |
| Swin-T [41] | 29.0 | 4.5 | 81.3 | - | Transformer | Manual | 2015 | - |
| TNT-S [24] | 23.8 | 5.2 | 81.5 | 95.7 | Transformer | Manual | 2015 | - |
| CPVT-S [17] | 23.0 | - | 81.5 | 95.7 | Transformer | Manual | 2015 | - |
| T2T-ViT_t-14 [72] | 21.5 | - | 81.7 | - | Transformer | Manual | 2015 | - |
| AutoFormer-S [11] | 22.9 | 5.1 | 81.7 | 95.7 | Transformer | Auto | CVPR2021 | 24 |
| GLiT-S [10] | 24.6 | 4.4 | 80.5 | - | Hybrid | Auto | ICCV2021 | N/A |
| ViTAS-F [56] | 27.6 | 6.0 | 80.5 | 95.1 | Transformer | Auto | ECCV2022 | 32 |
| TF-TAS-S [76] | 22.8 | 5.0 | 81.9 | 95.8 | Transformer | Auto | CVPR2022 | 0.5 |
| AZ-NAS [33] | 23.8 | 5.1 | 82.2 | - | Transformer | Auto | CVPR2024 | 0.07 |
| CET-TAS (Ours) | 30.9 | 6.3 | 81.5 | 95.3 | Transformer | Auto | - | 0.25 |

objective. Auto-Prox, however, is specialized to ViTs. Its search space hard-codes transformer-specific primitives. Similar variants, such as EZNAS [3] and Auto-DAS [57] follow identical graph enumeration but remain tied to fixed settings, respectively, limiting cross-domain portability and placing heavy reliance on human expertise.

## H.4 LLM for Neural Architecture Search

Recent research has explored leveraging large language models to assist or even automate the neural architecture search process. EvoPrompting [9] pioneers this direction by treating NAS as a code-level program synthesis task, where LLMs evolve architecture generation programs through an evolutionary prompt-tuning mechanism. By iteratively mutating and refining code snippets that specify architectural design choices, EvoPrompting enables LLMs to autonomously discover performant neural architectures without explicit gradient-based optimization. Building on this paradigm, LLMatic [47] combines LLM-driven architecture generation with quality-diversity optimization, allowing the search process to balance exploration and exploitation more effectively. Through LLM-guided mutation and recombination, LLMatic produces diverse yet high-performing architectures across vision and reinforcement learning benchmarks. Collectively, these works demonstrate that LLMs hold substantial potential for application in neural architecture search, suggesting that language-driven reasoning can serve as a viable mechanism for exploring the search space.

However, a critical limitation of these approaches lies in their reliance on evolutionary operations and crossover to progressively refine architectures. This paradigm inevitably necessitates the instantiation, training, and subsequent evaluation of each generated model to ascertain its performance, a process that is notoriously resource-intensive. The iterative cycle of generating a population of architectures, training them for even a few epochs, and measuring their fitness creates a significant computational bottleneck, rendering the search process both time-consuming and expensive. Consequently, the practical application of such methods is limited, particularly in scenarios demanding efficient and lightweight architecture discovery.

Figure 16: zero-cost proxy1 discovered by APD in the NAS-Bench-201.

```python
import torch
import torch.nn as nn

def proxy1(model, inputs, targets):
    bn_ranks = []
    ratios = []
    hooks = []

    def bn_hook(module, inp, out):
        if isinstance(out, torch.Tensor):
            B, C, H, W = out.shape
            mat = out.view(B, C, -1).permute(0, 2, 1).reshape(-1, C)
            frob_norm = torch.linalg.matrix_norm(mat, ord='fro')**2
            spec_norm = torch.linalg.matrix_norm(mat, ord=2)**2
            stable_rank = frob_norm / (spec_norm + 1e-6)
            bn_ranks.append(stable_rank.mean())

    for layer in model.modules():
        if isinstance(layer, nn.BatchNorm2d):
            hooks.append(layer.register_forward_hook(bn_hook))
        elif isinstance(layer, nn.Conv2d):
            weights = layer.weight
            l1_norm = weights.abs().sum(dim=(1,2,3)).mean()
            l2_norm = weights.norm(p=2, dim=(1,2,3)).mean()
            ratios.append((l1_norm / l2_norm).item())

    with torch.no_grad():
        model(inputs)

    for hook in hooks:
        hook.remove()

    bn_sum = torch.stack(bn_ranks).sum().item() if bn_ranks else 0.0
    ratio_sum = sum(ratios) if ratios else 0.0

    return bn_sum * ratio_sum
```

Figure 17: zero-cost proxy2 discovered by APD in the NAS-Bench-201.

```python
import torch
import torch.nn as nn

def proxy2(model, inputs, targets):
    ratios = []
    spatial_stds = []

    activations = []
    def hook_fn(module, input, output):
        activations.append((module.weight.detach(), output.detach()))

    hooks = []
    for layer in model.modules():
        if isinstance(layer, nn.Conv2d):
            hooks.append(layer.register_forward_hook(hook_fn))

    with torch.no_grad():
        model(inputs)

    for hook in hooks:
        hook.remove()

    for weight, act in activations:
        weight_norm = torch.norm(weight, p='fro')
        act_norm = torch.norm(act, p='fro')
        if act_norm != 0:
            ratios.append((weight_norm / act_norm).item())

        spatial_std = act.std(dim=1, keepdim=True)
        spatial_stds.append(spatial_std.mean().item())

    avg_ratio = sum(ratios) / len(ratios) if ratios else 0.0
    avg_std = sum(spatial_stds) / len(spatial_stds) if spatial_stds
        else 0.0
    return -2 * (avg_ratio * avg_std) / (avg_ratio + avg_std + 1e-10)
```

Figure 18: zero-cost proxy3 discovered by APD in the NAS-Bench-201.

```python
import torch
import torch.nn as nn

def proxy3(model, inputs, targets):
    activations = []

    def hook_cnn(module, input, output):
        activations.append(output.detach().flatten(2))

    hooks = []
    for layer in model.modules():
        if isinstance(layer, nn.Conv2d):
            hooks.append(layer.register_forward_hook(hook_cnn))

    with torch.no_grad():
        _ = model(inputs)

    for h in hooks:
        h.remove()

    scores = []
    for act in activations:
        act = act - act.mean(dim=-1, keepdim=True)
        channel_mean = act.abs().mean(dim=(0, 2))
        if channel_mean.mean() > 1e-6:
            cv = channel_mean.std() / channel_mean.mean()
            scores.append(cv.item())

    return -sum(scores) / len(scores) if scores else 0.0
```

Figure 19: zero-cost proxy1 discovered by APD in the TransNAS-Bench-101.

```python
import torch
import torch.nn.functional as F

def proxy1(model, inputs, targets):
    inputs.requires_grad_(True)

    with torch.enable_grad():
        output = model(inputs)
        loss = F.cross_entropy(output, targets)
        grad_clean = torch.autograd.grad(loss, inputs)[0]

        output = model(inputs + 0.01)
        loss = F.cross_entropy(output, targets)
        grad_noisy = torch.autograd.grad(loss, inputs)[0]

    inputs.requires_grad_(False)
    cos_sim = F.cosine_similarity(grad_clean.flatten(), grad_noisy.
        flatten(), dim=0)
    return cos_sim.cpu().item()
    return score.cpu()
```

Figure 20: zero-cost proxy2 discovered by APD in the TransNAS-Bench-101.

```python
import torch

def proxy2(model, inputs, targets):
    with torch.no_grad():
        clean_output = model(inputs)
        scaled_inputs = inputs * 1.01  # Small fixed scaling
        scaled_output = model(scaled_inputs)

    logit_diff = torch.norm(clean_output - scaled_output, p=2, dim=1)
        .mean()
    return logit_diff.cpu().item()
```

Figure 21: zero-cost proxy3 discovered by APD in the TransNAS-Bench-101.

```python
import torch
import torch.nn.functional as F

def proxy3(model, inputs, targets):
    with torch.no_grad():
        clean_output = model(inputs)
        noise = torch.zeros_like(inputs)
        for i in range(inputs.shape[2]):
            for j in range(inputs.shape[3]):
                noise[:, :, i, j] = 0.01 * ((i + j) % 2 * 2 - 1)
        noisy_output = model(inputs + noise)

    clean_probs = F.softmax(clean_output, dim=1)
    noisy_probs = F.softmax(noisy_output, dim=1)

    kl_div = F.kl_div(clean_probs.log(), noisy_probs, reduction='
        batchmean')
    return kl_div.cpu().item()
    return score.cpu()
```

Figure 22: zero-cost proxy1 discovered by APD in the AutoFormer.

```python
import torch
import torch.nn.functional as F

def heuristic_5(model, inputs, targets, loss_fn):
    with torch.no_grad():
        clean_output = model(inputs)
        noise_levels = [0.01, 0.05, 0.1]

        kl_values = []
        clean_probs = F.softmax(clean_output, dim=1)

        for level in noise_levels:
            noise = torch.randn_like(inputs) * level
            noisy_output = model(inputs + noise)
            noisy_probs = F.softmax(noisy_output, dim=1)
            kl_div = F.kl_div(clean_probs.log(), noisy_probs,
                reduction='batchmean')
            kl_values.append(kl_div)

        score = torch.prod(torch.stack(kl_values)) ** (1/len(
            kl_values))
    return score.cpu().item()
```

Figure 23: zero-cost proxy2 discovered by APD in the AutoFormer.

```python
import torch
import torch.nn.functional as F

def proxy2(model, inputs, targets, loss_fn):
    with torch.no_grad():
        # Input perturbation component
        noise = torch.randn_like(inputs) * 0.01
        perturbed_inputs = inputs + noise
        original_outputs = model(inputs)
        perturbed_outputs = model(perturbed_inputs)
        diff = torch.norm(original_outputs - perturbed_outputs, p=2,
            dim=1).mean()

        # KL divergence component
        original_softmax = original_outputs.softmax(dim=1)
        perturbed_softmax = perturbed_outputs.softmax(dim=1)
        kl_div = F.kl_div(original_softmax.log(), perturbed_softmax,
            reduction='batchmean')

    return (diff + kl_div).cpu().item()
```

Figure 24: zero-cost proxy3 discovered by APD in the AutoFormer.

```python
import torch

def proxy2(model, inputs, targets):
    with torch.no_grad():
        clean_output = model(inputs)
        scaled_inputs = inputs * 1.01  # Small fixed scaling
        scaled_output = model(scaled_inputs)

    logit_diff = torch.norm(clean_output - scaled_output, p=2, dim=1)
        .mean()
    return logit_diff.cpu().item()
```

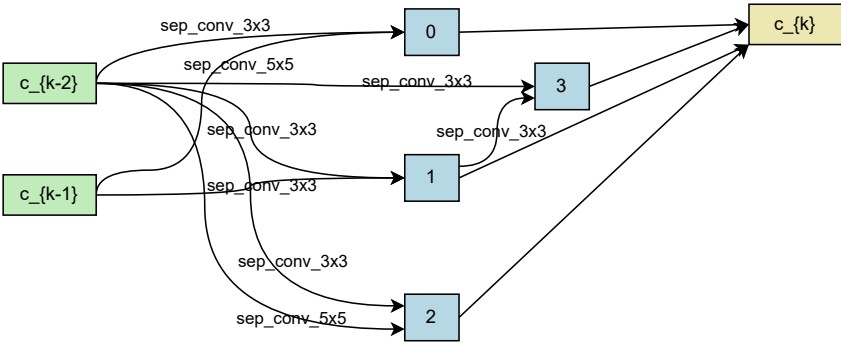

Figure 25: Normal cell searched by APD.

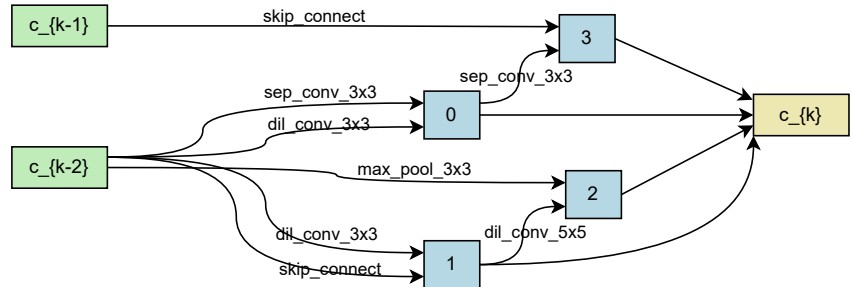

Figure 26: Reduction cell searched by APD.

Figure 27: Optimal structure found in the DARTS search space by APD.

```
Genotype(normal=[('sep_conv_3x3', 0), ('sep_conv_5x5', 1),
('sep_conv_3x3', 0), ('sep_conv_3x3', 1),
('sep_conv_5x5', 0), ('sep_conv_5x5', 0),
('sep_conv_3x3', 3), ('sep_conv_3x3', 0)], normal_concat=range(2, 6),
reduce=[('sep_conv_3x3', 0), ('dil_conv_3x3', 0),
('dil_conv_3x3', 0), ('skip_connect', 0),
('max_pool_3x3', 0), ('dil_conv_5x5', 3),
('sep_conv_3x3', 2), ('skip_connect', 1)], reduce_concat=range(2, 6))
```

Figure 28: Subnet discovered in the AutoFormer tiny search space by APD.

```
RETRAIN:
  MLP_RATIO:
  - 4
  - 3.5
  - 3.5
  - 3.5
  - 3.5
  - 4
  - 4
  - 4
  - 4
  - 3.5
  - 3.5
  - 3.5
  NUM_HEADS:
  - 4
  - 3
  - 4
  - 4
  - 4
  - 4
  - 4
  - 4
  - 3
  - 3
  - 3
  - 4
  DEPTH: 12
  EMBED_DIM: 240
```

Figure 29: Subnet discovered in the AutoFormer small search space by APD.

```
RETRAIN:
  MLP_RATIO:
    - 3.5
    - 3.0
    - 4.0
    - 4.0
    - 4.0
    - 3.0
    - 3.0
    - 3.0
    - 4.0
    - 4.0
    - 4.0
    - 3.0
    - 4.0
    - 3.0
  NUM_HEADS:
    - 7
    - 6
    - 7
    - 5
    - 7
    - 7
    - 6
    - 7
    - 7
    - 6
    - 5
    - 5
    - 7
    - 5
  DEPTH: 14
  EMBED_DIM: 448
```

Figure 30: Subnet discovered in the AutoFormer base search space by APD.

```
RETRAIN:
  MLP_RATIO:
    - 3.0
    - 3.0
    - 3.0
    - 4.0
    - 4.0
    - 3.5
    - 3.0
    - 4.0
    - 3.5
    - 3.0
    - 4.0
    - 3.0
    - 3.0
    - 4.0
    - 3.0
  NUM_HEADS:
    - 9
    - 10
    - 9
    - 10
    - 10
    - 9
    - 10
    - 9
    - 9
    - 10
    - 9
    - 9
    - 10
    - 9
    - 10
  DEPTH: 15
  EMBED_DIM: 576
```

