# OpenReview forum: "Revolutionizing Training-Free NAS: Towards Efficient Automatic Proxy Discovery via Large Language Models"
_NeurIPS.cc/2025/Conference — NeurIPS 2025 poster_

### Official Review · Reviewer_1iSx · 2025-06-26

**Clarity:** 3
**Significance:** 3
**Originality:** 2
**Rating:** 4
**Confidence:** 5

**Summary:**

The paper discusses the limitations of zero-cost proxies and proposes to address them by introducing an LLM-driven method to automatically discover such proxies. The method, named Automatic Proxy Discovery (APD), can be viewed as an evolutionary loop, consisting of: 1) an LLM-driven Proxy Candidate Generator, 2) a Fitness Evaluator, which evaluates proxies on a subset of architectures using their ground-truth performances, and 3) an RL Evolution Scheduler, an actor-critic algorithm that guides the evolutionary step. The authors evaluated it on different search spaces and datasets.

**Questions:**

Is the provided implementation sufficient to reproduce the results?

Could you address all my concerns regarding overfitting and reliance on the gt accuracies? How many unique architectures are used per search space? Are the 200 sampled once and reused, or redrawn each iteration? Do you use train or test accuracy?

Do you apply a proxy learned on one benchmark to another? This could provide a meaningful measure of overfitting. It seems that such cross-search-space generalization is not tested and a new proxy is found for each benchmark, but this is unclear. Also, which proxy is used for the DARTS experiments, and how is the final architecture selected for a given proxy?

To be more convinced, I would also like to know the real impact of the actor-critic module. If I understand correctly, it only samples one action among three - one of which (initialization) is likely to be used only at the beginning. While there is an ablation study evaluating this module, I do not find it satisfying.
In Table 7, what exactly does naive+evolution refer to? Does it mean sampling each action with 1/3 probability? If I understand correctly, if the initialization action is chosen in the i-th iteration, all proxies gathered so far in the first i-1 iterations discarded, which could significantly affect the outcome of the ablation study.
It seems like one could easily construct a handcrafted policy for sampling actions - something smarter than uniform 1/3 probability - that could match the performance of the actor-critic scheduler. I would appreciate it if the authors could go into more detail, clear my doubts, and provide stronger evidence that this module actually contributes meaningful value.

I am open to discussion and happy to be corrected or change my score if I misunderstood any aspects of the work.

**Ethical Concerns:**

["NO or VERY MINOR ethics concerns only"]

**Final Justification:**

I provided such justification for raising the score in my discussion with the authors.

**Limitations:**

Largely addressed; I mentioned some possibly unaddressed limitations of the work above.

**Paper Formatting Concerns:**

No concerns.

**Quality:**

3

**Strengths And Weaknesses:**

Strengths:
- The paper is well written and structured in a way that makes it easy to follow the main flow of the paper
- I appreciate that the authors clearly highlight the limitations of a naive approach that relies only on an LLM. They also elaborate on these shortcomings in the appendix and discuss prompt engineering, which I consider valuable. Rather than incorporating LLMs into NAS in such a naive way, the authors wrap them in an evolutionary loop, which is an interesting design
- I find the high-level idea of restricting the LLM's role to generating or mutating proxies to be compelling. While a simple LLM-only approach falls short, it intuitively makes sense that an LLM should perform reasonably well at this specific task of modifying proxies in each iteration. Overall, while it is difficult to effectively incorporate LLMs into NAS, the authors presented a creative and promising approach.

Weaknesses:
- Possible overfitting. Overfitting is a major issue in NAS, but this is not talked about at all in the paper. In each iteration, APD generates several new proxies, ranks them by the correlation with ground-truth (gt) performance, and drops the worst ones, which potentially makes the method just an overly-complex way to overfit the benchmark. There are in total 10 * 100 such iterations, which - depending on whether architectures are reused across iterations - can lead to either overfitting to a small set of architectures (especially that the authors use a single fixed seed) or excessive reliance on the gt performance (see the point below).
- Vague reliance on gt accuracies. In the appendix, the authors state that “computing fitness requires baseline accuracies for roughly 2-3% of each benchmark”. However, it is unclear where this number comes from. There are 10 * 100 iterations, and in each iteration candidate proxies are evaluated on 200 architectures, potentially yielding the need for gt performance of 200k architectures if the batch of architectures gets resampled each time. Unfortunately, it is impossible to verify since these details are not included in the paper, and the code that corresponds to the evaluation of proxies is simply missing. A fixed set of architectures can be treated as a best-case scenario, but I am sceptical as this is not backed by any evidence.  Also, it is unclear if these are train or test accuracies, or how this number scales with the size of the search space. I do not feel like these issues were given enough attention.
- Poor reproducibility. As described above, important details are not included in the paper. The implementation seems to be incomplete, as some modules (such evaluation of candidate proxies) are missing. Also, the requirements can be installed only on Windows OS.
- While the paper is overall well-written, the phrasing is sometimes catchy and inaccurate. Calling the method a zero-cost proxy (ZCP) is misleading, as it still relies on many gt accuracies. Similarly, the paper mentions chain-of-thought a few times, but the only overlap is just a standard actor-critic loop. In particular, there is no reward-trained LLM reasoning.

---

> ### Author Rebuttal · Authors · 2025-07-31
>
> Thank you for the helpful and insightful review, which has helped us improve this paper further. Next, we will answer your questions one by one, and we hope this will improve your acceptance of the paper.
>
> **Q1**: Possible overfitting
> **A1**: Many thanks for your comments! Our method does not suffer from overfitting, and we have provided clear empirical and conceptual evidence to support this. First, while APD searches over a small subset (200) from the source search space (NAS-Bench-201), the discovered proxy generalizes well to unseen search spaces (NAS-Bench-101, DARTS) and datasets (CIFAR-100, ImageNet), as demonstrated in Tables 3–4 and Appendix D. If overfitting were occurring, APD cannot obtain optimal performance cross unseen search spaces. Second, regarding the broader claim that “overfitting is a common issue in NAS,” we want to clarify: overfitting in traditional NAS typically arises during the full training of architectures after the search stage. In contrast, APD is a training-free method. It does not involve gradient-based updates or full model training, which fundamentally eliminates the main source of overfitting in standard NAS pipelines. Hence, both in practice and by design, our method is robust to overfitting.
>
> **Q2**: Vague reliance on gt accuracies
> **A2**: Many thanks for your comments! To clarify, following prior works published at ICLR [1] and NeurIPS [2], we define the set B as a single, fixed subset of 200 architectures randomly sampled once from the full benchmark using a fixed random seed. This subset remains unchanged throughout the entire search process, and only the test accuracies of these architectures are used for all fitness evaluations. No new architectures are re-sampled or introduced during the search.
>
> Our evaluation focuses on the relative ranking of proxies within this subset rather than their absolute correlation scores. To validate the reliability of this design, we conducted an ablation study using 30 discovered proxies. We calculated each proxy’s correlation scores on subsets of varying sizes (50, 100, 200, and 500 architectures) and compared the resulting rankings to those derived from the full benchmark via Kendall’s Tau. The results show consistently high rank correlation across all subset sizes, confirming that a small, randomly sampled subset is sufficient for reliable evaluation while effectively balancing accuracy and computational cost. These experimental details have been added to the final version to improve reproducibility and transparency.
> |Subset Size|Kendall's Tau|
> | :---: | :---: |
> |50|0.811|
> |100|0.913|
> |200|0.991|
> |500|1.00|
>
> [1] Zero-cost proxies for lightweight NAS, ICLR, 2021.
> [2] How powerful are performance predictors in neural architecture search?， NeurIPS 2021.
>
> **Q3**: Poor reproducibility.
> **A3**: Many thanks for your comments! As stated in "A2", we first validate and improve the experimental details. In addition, we provide the main code for the evaluation as follows:
>
> ```python
> def evaluate(args, f_score):
>     # load archs and dataset
>     ...
>     loader = iter(train_loader)
>     inputs, targets = next(loader)
>     eval_len = len(arch_info)
>     results = []
>     index_arr = np.arange(len(arch_info))
>     np.random.seed(args.seed)
>     np.random.shuffle(index_arr)
>     for index, i in enumerate(arch_info.values[index_arr[:eval_len]]):
>         start_t = time.perf_counter()
>         network = model_201.Network(16, 4, i[0], 10)
>         network = network.to(device)
>         swap_score = []
>         for _ in range(args.repeats):
>             network = network.apply(network_weight_gaussian_init)
>             score = f_score(network, inputs.to(device), targets.to(device))
>             swap_score.append(score)
>         total_eval_time += time.perf_counter() - start_t
>         results.append([i[0], np.mean(swap_score), i[1]])
>     results = pd.DataFrame(results, columns=['genotype', 'swap_score', 'acc'])
>     avg_time = total_eval_time / len(results)
>     return stats.spearmanr(results.swap_score, results.acc)[0] + avg_time * args.beta
> ```
>
> For concern of Windows, our method is compatible with both Windows and Linux operating systems. The win-inet-pton package is included only for Windows systems with GPU support to handle certain network utilities. When running on Linux, this package is not required and can be safely removed without affecting the core functionality or performance of our framework.
>
> **Q4**: Inaccurate statements.
> **A4**: Many thanks for your comments! First, we agree with the reviewer that zero-cost proxy (ZCP) methods are not entirely "zero cost" as they rely on a limited set of ground-truth (gt) accuracies. Our definition of ZCP follows established conventions from prior works such as ZiCo and AZ-NAS, where ZCPs are evaluated without additional training. Nonetheless, we acknowledge the terminology could be misleading and will revise it in the final version for clarity.
>
> Second, regarding the use of Chain-of-Thought (CoT), we appreciate the opportunity to clarify. As discussed in [1], reward-trained LLM reasoning typically uses training accuracy as a reward to guide LLM generation. However, this approach is often ineffective due to the semantic gap between language models and NAS-specific tasks—accuracy is a sparse and indirect signal, which we also observed in our naive baseline.
>
> To overcome this limitation, we introduce an actor-critic RL controller that learns to schedule generation strategies (e.g., initialization, mutation, crossover) in a way that promotes iterative, feedback-driven improvements. While we do not perform token-level CoT prompting or reasoning within the LLM, our controller-driven proxy discovery mechanism mimics multi-step reasoning across iterations. To reflect this more accurately, we now define APD as: “A controller-guided, Chain-of-Thought-inspired scheduling framework for evolving zero-cost proxies.”
>
> We will update the manuscript to use this revised phrasing, avoiding confusion with conventional CoT paradigms while preserving the intuition of multi-step guided reasoning.
>
> [1] Can gpt-4 perform neural architecture search?, arXiv 2023.
>
> **Q5**: Concern about overfitting and GT
> **A5**: Many thanks for your comments! For overfitting and GT, see Q1 and Q2. In addition, we use a fixed set of 200 architectures—sampled once per benchmark—and reuse it throughout the search. Evaluation is consistently based on their test accuracy.
>
> **Q6**: Impact of the Actor-Critic Scheduler
> **A6**: Many thanks for your comments! We feel sorry for not explaining the ablation study in Table 7 with sufficient clarity. To clarify, the "naive + evolution" setting is a specific baseline designed to test a simple, greedy evolutionary strategy. It refers to an iterative process where we continuously use the simple prompt from Appendix A to have the LLM generate new ZCPs. In each generation, only the single best-performing ZCP from the current population is retained. This is distinct from sampling the three actions with uniform probability. We would also like to clarify that in our full APD framework, the 'initialization' action does not discard the existing population but rather introduces new, diverse proxies into it to escape local optima without resetting progress.
> The core contribution of the Actor-Critic module is that it provides a principled, learning-based solution to bridge the significant reasoning gap between the generative, black-box LLM and the NAS tasks, where simpler strategies struggle with reliability. The agent must learn a policy over abstract actions based on a noisy and unstable feedback signal. This learned, adaptive control over the search is the meaningful value that a static or simple handcrafted policy cannot provide, leading to a more robust and effective discovery process.
>
> Regarding whether handcrafted policies can replace the RL controller, once RL is used, the naive 1/3 uniform sampling strategy is not applied. To verify the RL policy’s advantage, we tested two additional handcrafted strategies: the fixed strategy refers to using a constant action sampling ratio of 1:2:1. The dynamic strategy is divided into two stages: in the first stage, the probabilities of Initialization and Crossover are higher, while in the later stage, the probability of Mutation becomes dominant. Experimental results show that our RL method consistently outperforms these handcrafted strategies, demonstrating its superior adaptive control of the search.
>
> |Method|CIFAR-10(NB-201)|
> |:---:|:--:|
> |Naive|82.04|
> |Naive+EC|88.53|
> |Naive+EC+fixed|88.97|
> |Naive+EC+dynamic|89.13|
> |Naive+EC+Actor-Critic|93.76|

---

> > ### Comment · Reviewer_1iSx · 2025-08-01
> >
> > Thank you for the rebuttal! It cleared up some of my concerns and I am inclined to raise the score, but I have a few follow-up questions if you do not mind.
> >
> > **Q1:**
> >
> > I might have not explained my point clearly. I do not really agree with the statement that “overfitting in traditional NAS typically arises during the full training of architectures after the search stage”. I was not referring to the standard notion of “weights” overfitting, but to method-level overfitting when developing a new NAS technique. Because APD is tuned on a **fixed** subset of 200 architectures, there is a risk that if we tested APD with different seeds or moved to a new benchmark without tuning any hyperparameters, the results might not hold. The same applies to prompt selection. If the “carefully structured prompts” were found via trial-and-error procedure on each benchmark, that process itself can overfit.
> >
> > From what I can see, you ran APD to find a proxy on three different search spaces: NAS-Bench-201, TransNAS-Bench-101-Micro and AutoFormer (according to the appendix). This partially addresses my concerns, but each of those experiments still used its own prompt. Could you specifically briefly comment on (i) sensitivity to the random seed within a benchmark, and (ii) how prompts were chosen and whether a single generic prompt would work across tasks?
> >
> > **Q2:**
> >
> > Thank you for clarifying that you reuse the same 200 architectures. Using test accuracy is what I was concerned about. This does not feel right, but on the other hand, using training accuracy might not be very representative in the case of CIFAR-10. However, this might not be an issue with this specific paper, likely there are more papers using test GT accuracy this way.
> >
> > I am still unsure where “the 2-3% of each benchmark” claim comes from. For an enormous search space that is much larger than NAS-Bench-201, would 2-3% still be enough, or would the subset need to grow? Even after reading the paper a number of times and after reading the rebuttal, it is difficult for me to at least estimate the cost of APD. Not to mention that if we need to tune prompts for each search space separately, the cost grows significantly.
> >
> > **Q3:**
> >
> > Thank you for the clarification and for providing the code! I probably will not be able to run it this week, but I appreciate the transparency.
> >
> > **Q6:**
> >
> > Thank you for providing more ablation data, but I am still unsure what the scheduler is actually learning. Presenting results for another very specific handcrafted strategies did not clear my concerns. Could you briefly describe the policy you observe after training? What operations does it tend to choose? Knowing this would help judge whether the AC scheduler adds any real value. Right now, unless the AC scheduler proves to be useful, I am still concerned about the novelty and significance of APD.
> >
> > (Feel free to answer only what you have time for, even short clarifications would help. I acknowledge that I should have asked some of these questions in my review to give you more time.)

---

> > > ### Author Response · Authors · 2025-08-06
> > >
> > > Dear reviewers 1iSx,
> > >
> > > We sincerely appreciate your valuable feedback.
> > >
> > > Your positive comments and constructive suggestions have inspired us to make a number of updates to the paper, which we believe will improve over the original version.  During the rebuttal, we devote lots of effort & exploration in the APD day and night.  If our responses have adequately addressed your concerns, we kindly hope that you can consider increasing the score.
> > >
> > > Next, we will answer your questions one by one, and we hope this will improve your acceptance of the paper.
> > >
> > >
> > >
> > > **Q1**: Concerned about method-level overfitting
> > >
> > > **A1**: Thank you for the insightful follow-up questions and for clarifying your concerns regarding method-level overfitting. We now have a clearer understanding of your point, and we agree that this is a critical aspect to address for demonstrating the robustness and generalizability of our method.
> > >
> > > We want to clarify that our approach is not specifically tuned for each benchmark. In fact, our experimental setup on the three different search spaces (NAS-Bench-201, TransNAS-Bench-101-Micro, and AutoFormer) was kept almost identical, with the only changes being the specific descriptions of the search space and dataset in the prompts. The consistent, high-performance results across these diverse benchmarks, as shown in our paper, strongly suggest that our method is not overfitting to a particular search space.
> > >
> > > To further address your concern about the sensitivity to random seeds, we have conducted additional experiments using different seeds within each benchmark. The results are as follows:
> > >
> > > Table1. The impact of different seeds to performance.
> > >
> > > | seed | NASBench-201 (CIFAR-10) | NASBench-201 (CIFAR100) | NASBench-201 (Img-1k) | NASBench-101 (CIFAR-10) |
> > > | :--: | :---------------------: | :---------------------: | :-------------------: | :---------------------: |
> > > |  0   |          93.76          |          72.22          |         45.03         |          93.49          |
> > > |  1   |          93.62          |          72.16          |         44.7          |          93.68          |
> > > |  2   |          93.86          |          71.66          |         45.33         |          92.67          |
> > > |  3   |          93.59          |          71.01          |         44.78         |          93.55          |
> > > |  4   |          93.81          |          72.40          |         44.40         |          93.73          |
> > >
> > > The results demonstrate the strong stability and consistent performance of APD across different seeds.
> > > Regarding the prompts, the structure of the prompts for each benchmark is **nearly identical**. The **only modifications were specific to the search space and dataset**, while the **core instructions and reasoning were kept consistent**. This design choice was deliberate, aiming to leverage the LLM's general knowledge and reasoning abilities rather than relying on a handcrafted, task-specific prompt for each scenario. We believe this approach contributes to the method's transferability and highlights the potential of using a consistent prompting strategy across different NAS tasks. The detailed prompt engineering of APD can be found in **Appendix C**.
> > >
> > > Furthermore, we conducted an additional experiment where the proxy learned on NAS-Bench-201 (CIFAR-10) was directly applied to other unseen search spaces (e.g., TransNAS-Bench-101-Micro and OoD-ViT-NAS-Ti) **without any prompt change or proxy re-searching**, and it still surpasses the best previous methods. This further indicates that the learned proxy has strong generalization ability, and APD does not overfit to the original tuning space or prompt. The detailed results are as follows:
> > > **(1) Searching NAS-Bench-201 (CIFAR-10), transferred for new search spaces (TransNAS-Bench-101--Micro)**
> > >
> > > **As stated in A4,** APD is not limited to a specific search space and can generalize across multiple ones ( i.e., NAS-Bench-201\&101, DARTS). When the discrepancy between spaces is small, the proxy found on the source (e.g., 201) transfers well to the target (e.g., 101, DARTS), as shown in Tables 3–4 in the paper. For significantly different spaces (e.g., 201 vs. TransNAS,  201 vs. AutoFormer), the proxy may become suboptimal due to search space bias rather than limitations of APD. In such cases, we retrain the proxy for the new space, which leads to strong performance, validating our approach. Moreover, APD is highly efficient, requiring only about 1.2 hours on a single RTX 4090 GPU with 4 GB of VRAM.

---

> > > ### Author Response · Authors · 2025-08-06
> > > **part 2**
> > >
> > > For futuher validate our statement, we provide experiments that only use proxy searched on NAS-Bench-201 for unseen TransNAS-Bench-101--Micro search spaces and three tasks (i.e., Autoencoding, Scene Classification, and Jigsaw). As shown Table 2,  **"APD (Searching on NAS-Bench-201)"** still surpass its peer competitors (i.e., SWAP, ZiCo). **Those results support our statement of strong transferability of APD for TransNAS-Bench-101--Micro.**
> > >
> > > Noabtly, "APD (Searching on TransNAS-Bench-101)" obtain better performance than "APD (Searching on NAS-Bench-201)", this validate our statement " For significantly different spaces (e.g., 201 vs. TransNAS,  201 vs. AutoFormer), the proxy may become suboptimal due to search space bias rather than limitations of APD. In such cases, we retrain the proxy for the new space, which leads to strong performance, validating our approach".
> > >
> > > Table 2. Results on TransNAS-Bench-101-Micro.
> > >
> > > |    | Autoencoding | **Scene Classification** |  **Jigsaw**  |
> > > | :----- | :----------: | :---: | :----------: |
> > > |   |     SSIM     |       Accuracy (%)       | Accuracy (%) |
> > > | Ground Truth |    0\.58     |          54\.9           |    95\.4     |
> > > | Grad | 0\.36± 0.03  |        48\.7±0.7         |  80\.3±0.3   |
> > > | SNIP |  0\.33±0.04  |        48\.7±1.1         |  80\.3±0.1   |
> > > | Grasp |  0\.33±0.06  |        50\.2±1.6         |  91\.1±0.3   |
> > > | Fisher|  0\.49±0.01  |        48\.7±0.6         |  83\.5±1.2   |
> > > | Synflow |  0\.46±0.07  |        53\.7±1.2         |  90\.9±0.4   |
> > > | NWOT|  0\.43±0.02  |        53\.2±0.6         |  92\.3±0.3   |
> > > | Zen-score|  0\.46±0.01  |        53\.7±0.2         |  87\.5±0.4   |
> > > | GradSign  |  0\.35±0.03  |        53\.6±0.4         |  93\.1±0.4   |
> > > | Params|    0\.46     |          53\.7           |    85\.9     |
> > > | FLOPs  |    0\.46     |          53\.7           |    85\.9     |
> > > | ZiCo |  0\.48±0.02  |        53\.7±0.4         |  93\.2±0.4   |
> > > | SWAP|  0\.42±0.02  |        45\.0±10.9        |  89\.8±5.6   |
> > > | **APD (Searching on NAS-Bench-201)**      |  0\.53±0.06  |        53\.8±0.02        |  91\.1±0.03  |
> > > | **APD (Searching on TransNAS-Bench-101)** |  0\.54±0.01  |        54\.0±0.6         |  91\.2±0.1   |
> > >
> > > **(2) Searching NAS-Bench-201 (CIFAR-10), transferred for new search spaces (OoD-ViT-NAS-Ti )**
> > >
> > > For the unseen OoD-ViT-NAS-Ti search space, we draw the same conclusion as TransNAS-Bench-101--Micro.
> > >
> > > Table 3.  Correlation on OoD-ViT-NAS-Ti search space
> > >
> > > | Method                                | ImageNet1k | ImageNet-A | ImageNet-R | ImageNet-D/Texture | ImageNet-D/Material |
> > > | --- | ---------- | ---------- | ---------- | --- | --- |
> > > | DSS | 0.62       | 0.82       | 0.81       | 0.02               | 0.17                |
> > > | AutoProx| 0.67       | 0.82       | 0.78       | 0.05               | 0.15                |
> > > | NWOT| 0.75       | 0.76       | 0.74       | 0.11               | 0.12                |
> > > | **APD (Searching on NAS-Bench-201)**  | 0.77       | 0.81       | 0.86       | 0.12               | 0.13                |
> > > | **APD (Searching on OoD-ViT-NAS-Ti)** | 0.79       | 0.82       | 0.88       | 0.12               | 0.15                |

---

> > > ### Author Response · Authors · 2025-08-06
> > > **part 3**
> > >
> > > **Q2**: Uncertainty about the cost and scalability of APD, and the risk of method-level overfitting from using a small fixed subset of architectures and tuning prompts for each benchmark.
> > >
> > > **A2**: Thank you for the insightful feedback on the scaling of our method. Next, we will address your questions one bu one.
> > >
> > > (1) Statement of a fixed subset and test accuracy
> > >
> > > Using a fixed subset of architectures and relying on test accuracy for proxy supervision is not unique to our work. Our experimental design follows common practice established by recent works such as SWAP (ICLR 2024), AutoProx (AAAI 2024), and AZ-NAS (CVPR 2024). These works also adopt test accuracy from the benchmark as the target for training proxies. The rationale is that, in NAS benchmarks, it is the test accuracy—rather than training accuracy—that best reflects the true performance of an architecture, and thus is most appropriate as the ground-truth signal for training a proxy.
> > >
> > > (2) Clarification on the “2–3%” claim and subset size
> > >
> > > Upon revisiting our manuscript, we found that the “2–3% of each benchmark” claim was a misstatement. As shown in Table 4, what we intended to convey is that for **highly similar search spaces**—such as NAS-Bench-201 vs. NAS-Bench-101, or NAS-Bench-201 vs. DARTS—we reused the same sample size (e.g., 200 architectures) due to the structural similarity of the spaces. We do not claim that a fixed 2–3% subset size is universally sufficient across all possible search spaces.
> > >
> > > (3) The impact of different subset to performance
> > >
> > > For significantly larger and more complex search spaces, we do not assume that a fixed subset size of 200 architectures is universally optimal. To validate this, we conducted comprehensive experiments on two diverse and substantially larger search spaces: TransNAS-Bench-101-Micro and OoD-ViT-NAS-Ti. In both cases, we systematically evaluated the performance of APD using different subset sizes ranging from 100 to 1000.
> > >
> > > The results clearly show that a subset size of 500 offers the best trade-off between performance and efficiency. Increasing the number of sampled architectures beyond 500 did not lead to noticeable improvements in performance, while it incurred slightly higher computational costs. Therefore, we set the subset size to **500 for both TransNAS-Micro (Table 5)**  and **OoD-ViT-NAS-Ti (Table 6)** .
> > >
> > > Importantly, even at the largest subset size of 1000, the total search cost remains extremely low. For example, on OoD-ViT-NAS-Ti, APD only requires **0.0482 GPU Days**, which is several orders of magnitude lower than conventional NAS methods. This highlights APD’s significant advantage in practical deployment scenarios, where computational budget is often limited. These findings further validate the scalability and cost-efficiency of our method. We will incorporate these results and clarifications in the final version of the paper.
> > >
> > > Table 4. The impact of different subset to performance on NB-201 and NB-101.
> > >
> > > | **Method**   | CIFAR-10(NB-201) | CIFAR-100(NB-201) | ImageNet16-120(NB-201) | **CIFAR10(NB-101)** | Runtime (ms/arch) |
> > > | :----------- | :--------------: | :---------------: | :--------------------: | :-----------------: | :---------------: |
> > > | **APD(100)** |   93.71 ±0.01    |    72.13±0.03     |      44\.34± 0.01      |     93\.16±0.15     |       8.73        |
> > > | **APD(200)** | **93\.76 ±0.09** | **72\.22 ± 0.65** |   **45\.03  ±0.76**    |   **93\.49±0.34**   |      16\.81       |
> > > | **APD(500)** |   93\.76 ±0.02   |   72\.22 ± 0.02   |      45\.03±0.12       |    93\.49± 0.02     |       42.31       |
> > >
> > > Table 5. The impact of different subset to performance on TransNAS-Bench-101-Micro.
> > >
> > > |               | Autoencoding | **Scene Classification** |  **Jigsaw**  | Search Cost (GPU Days) |
> > > | :------------ | :----------: | :----------------------: | :----------: | ---------------------- |
> > > |               |     SSIM     |       Accuracy (%)       | Accuracy (%) |                        |
> > > | Ground Truth  |    0\.58     |          54\.9           |    95\.4     |                        |
> > > | **APD(100)**  |  0\.51±0.07  |        53\.5±0.1         |  91\.2±0.3   | 0.0047                 |
> > > | **APD(200)**  |  0\.53±0.13  |        53\.7±0.1         |  91\.2±0.3   | 0.0088                 |
> > > | **APD(500)**  |  0\.54±0.01  |        54\.0±0.6         |  91\.2±0.1   | 0.0247                 |
> > > | **APD(700)**  |  0\.54±0.02  |        54\.0±0.2         |  91\.2±0.2   | 0.0272                 |
> > > | **APD(1000)** |  0\.54±0.01  |        54\.0±0.1         |  91\.2±0.1   | 0.3951                 |

---

> ### Author Response · Authors · 2025-08-06
> **part 4**
>
> Table 6.   The impact of different subset to performance on  OoD-ViT-NAS-Ti search space
>
> | Method        | ImageNet1k | ImageNet-A | ImageNet-R | ImageNet-D/Texture | ImageNet-D/Material | Search Cost (GPU Days) |
> | ------------- | ---------- | ---------- | ---------- | ------------------ | ------------------- | ---------------------- |
> | **APD(100)**  | 0.69       | 0.76       | 0.81       | 0.10               | 0.12                | 0.0051                 |
> | **APD(200)**  | 0.74       | 0.80       | 0.78       | 0.11               | 0.13                | 0.0098                 |
> | **APD(500)**  | 0.79       | 0.82       | 0.88       | 0.12               | 0.15                | 0.0174                 |
> | **APD(700)**  | 0.80       | 0.81       | 0.87       | 0.12               | 0.14                | 0.0341                 |
> | **APD(1000)** | 0.79       | 0.82       | 0.88       | 0.12               | 0.15                | 0.0482                 |
>
>
>
> **Q3**: What the scheduler is learning.
>
> **Q3**: Thank you for your question. We are happy to provide more insight into the behavior of the Actor-Critic scheduler. We agree that understanding what the policy learns is essential for evaluating its contribution and the novelty of our method.
> As we mentioned in our manuscript, the generative process of LLMs is inherently opaque and unpredictable. This is a critical limitation, as the LLM's output can be highly unstable. For example, in our experiments, we observed that an LLM might initially generate a cluster of low-quality proxies. In this scenario, the RL policy, detecting the stagnation, learns to favor initialization to introduce fresh, diverse ideas. However, in a different scenario where a high-performing proxy is generated early, the policy dynamically shifts its focus towards mutation and crossover operations, aiming to exploit and refine this initial success rather than starting over.
> Generally, the RL controller converges to a policy that heavily favors mutation. However, in some other search spaces, such as TransNAS, the policy converges on crossover. The final generation count (T) for APD to converge is also highly uncertain, which is why we need the RL controller. The significant performance gain of the full APD framework over the "Evolution only" and "Naive" baselines in **our ablation study (Table 7)** directly validates the real value added by the AC scheduler. Therefore, it is difficult to definitively state what the final converged policy will look like, as each run can produce a different but effective policy tailored to the specific dynamics observed during that particular search.
>
>
>
> Table 7. Ablation study of various components.
>
> | **Naive** | **Evolution** | **Actor-Critic** | **CIFAR-10(NB-201)** | **CIFAR-100(NB-201)** |
> | :-------: | :-----------: | :--------------: | :------------------: | :-------------------: |
> |    Yes    |      No       |        No        |        82\.04        |        47\.62         |
> |    Yes    |      Yes      |     No      |        88\.53        |        61\.16         |
> |    Yes    |      Yes      |       Yes        |        93\.76        |        72\.22         |
>
> We are grateful to the reviewer for the thoughtful and insightful comments on the Actor-Critic scheduler. These remarks led to a deeper analysis and clearer articulation of the policy's behavior, which significantly enhanced the scientific contribution and technical depth of our work.

---

> > ### Comment · Reviewer_1iSx · 2025-08-06
> >
> > I appreciate the effort that the authors put into providing new experiments.
> >
> > They demonstrate stable behavior across different seeds and strong proxy generalizability, which addresses my main concerns about overfitting and lack of generalizability. Hence, I will update my score from **Reject** to **Borderline Accept**.
> >
> > To clarify my initial negative score, I want to reiterate the point from my initial review. These issues were not given enough attention in the paper. To verify generalizability and susceptibility to overfitting, I had to look deeper into the appendix, ask the authors follow-up questions, and request additional experiments. Given that APD is designed as an iterative “overfitting loop” to tune proxies, I think those concerns were justified. Now that they are resolved, the work is much more compelling.
> >
> > **Reasons for raising the score from R to BA:**
> >
> > - Proxy learned on one search space shows good transferability
> > - Stable performance across different seeds
> > - Ablation study on the subset size
> > - Missing implementation file provided
> >
> > **Unaddressed concerns**
> >
> > **1. Unclear value of the AC scheduler.**
> >
> > The authors’ response still does not convince me that the scheduler adds any value. I would like to see concrete data, e.g. how the distribution of actions evolves throughout the training and histograms of sampled actions on different search spaces.
> >
> > The ablation study baselines also seem arbitrary. I have an impression that they were not chosen in a fair way. For example, I can think of such three heuristics as candidates for good baselines:
> >
> > - Reinitialize if performance is low. Otherwise use $100$% crossover
> > - Reinitialize if performance is low. Otherwise use $100$% mutation
> > - Reinitialize if performance is low. Otherwise use $50$% crossover and $50$% mutation
> >
> > If the AC scheduler can be replaced by such a simple heuristic, it would undermine paper’s novelty and significance, making APD appear more like a cleverly-engineered loop to tune proxies rather than a genuinely **novel research** method. I hope this makes my concerns clear.
> >
> > **2. Prompt engineering**
> >
> > The authors claim that “*the only modifications were specific to the search space and dataset, while the core instructions and reasoning were kept consistent*”. Looking at the prompts shown in the appendix, I have to disagree with this statement. Within each prompt category (one of Initialization, Mutation, Crossover), many supposedly benchmark-agnostic sections differ across benchmarks. For example:
> >
> > - “*Input-aware: The proxy should utilize a batch of input image data*” appears only in NAS-Bench-201’s Initialization prompt
> > - Using very customized **Note** sections in selected prompts
> > - Using custom **Mutation Rules** and **Proxy Search Space** for the Autoformer benchmark
> >
> > This is a short, but non-exhaustive list of differences between prompts. It might seem like the prompts were iteratively tuned by the authors to resolve any encountered issues, significantly increasing the hidden costs of applying APD to new search spaces. I strongly think that the authors should preserve as much of the benchmark-agnostic prompt structure as possible across search spaces and be transparent about any benchmark-specific changes.

---

> > > ### Author Response · Authors · 2025-08-08
> > > **part 3**
> > >
> > > **Q3**: Prompt engineering
> > >
> > > **A3**: Thank you for this insightful and important point. We agree completely that transparency about the prompt structure is crucial for reproducibility and for understanding the true cost of applying APD. We apologize if our initial phrasing, “the only modifications were specific to the search space and dataset,” was not precise enough. You are correct to point out that there are specific modifications in the prompts for different benchmarks. Our intention was to convey that the high-level reasoning structure is consistent.
> > >
> > > We agree that your concern about the potential for "fine-tuned" prompts is valid, and demonstrating that APD is robust to such variations is essential. To address this concern directly and further validate that APD is not overly sensitive to the prompt structure, we have conducted a new ablation study. In this experiment, we strictly aligned the prompts used for different search spaces. Specifically, we adopted the prompt for NAS-Bench-201 (detailed in Appendix D.1), with the "Input-aware" instruction removed, as a universal template. For other benchmarks, the only modifications we made were to the sections that are fundamentally problem-specific: the description of the unique search space and the corresponding dataset/task details. All other components of the prompt, including the core instructions, reasoning structures, and strategic Note sections, were kept identical to those used for NAS-Bench-201.
> > > The results of this experiment are as follows:
> > >
> > > | NAS-Bench-201 (CIFAR-10) | NAS-Bench-201 (CIFAR-100) | NAS-Bench-201 (ImageNet16-120) | NAS-Bench-101 (CIFAR-10) |
> > > | :---: | :---: | :---: | :---: |
> > > | 93.69% | 71.92% | 45.33% | 93.52% |
> > >
> > > | AutoEncoding | Scene Classification | Jigsaw |
> > > | :---: | :---: | :---: |
> > > | SSIM | Acc | Acc |
> > > | 0.51 | 53.8% | 92.8% |
> > >
> > > In summary, this new ablation study provides strong empirical evidence that our framework's performance is robust to minor variations in the prompt.  The benchmark-specific modifications are not a form of iterative, hidden tuning, but rather the necessary and principled step of providing the LLM with the problem's domain constraints (i.e., the search space definition).
> > > In light of your valuable feedback, we will adopt this unified prompt template for all benchmarks in the final version of our paper to ensure maximum clarity and reproducibility.

---

> ### Author Response · Authors · 2025-08-08
>
> We would like to sincerely thank you for your diligent and thoughtful feedback throughout the review process, and we are very grateful for your decision to raise our score to Borderline Accept. Your insightful comments have been invaluable in helping us strengthen the paper.
> We understand that your remaining questions about the value of the AC scheduler and the benchmark-specific details of the prompt engineering are critical. We agree that these points deserve a thorough, data-driven explanation.
>
> **Q1**: Unclear value of the AC scheduler.
>
> **A1**: To provide a more concrete and data-driven analysis of the AC scheduler's behavior, as you requested, we will now present the specific details of its action selection process in NASBench-201 (CIFAR-10). First, we show the overall statistics for each action, as summarized in the table below:
>
> | Action | Share |
> | :---: | :---: |
> | Init | 27.7 % |
> | Crossover | 40.5 % |
> | Mutation | 31.8 % |
>
> Furthermore, to illustrate how the agent's policy evolves over time, we define the first 30% of total generations as the 'early stage', the final 30% as the 'late stage', and all other generations as the 'middle stage'. The following table contrasts the distribution of actions selected during these distinct phases:
>
> | Stage | Init | Crossover | Mutation | Mean Acc|
> | :---: | :---: | :---: | :---: | :---: |
> | Early | 41.3 % | 21.7 % | 36.9 % | 88.14 % |
> | Mid | 28.3 % | 40.0 % | 31.7 % | 92.17 % |
> | Late | 15.2 % | 52.2% | 32.6% | 93.75 % |
>
> To provide context for these decisions, the following table shows the detailed progression of the average correlation coefficient and average acc across generations, illustrating the performance observed:
>
> | Generation | Mean $\rho$ | Mean Acc |
> | :---: | :---: | :---: |
> | 1 | 0.009 | 63.19 % |
> | 20 | 0.391 | 78.81 % |
> | 50 | 0.741 | 88.91 % |
> | 80 | 0.809 | 93.25 % |
> | 100 | 0.823 | 93.75 % |
>
> To provide deeper insight into the value of the learned policy, we can analyze the direct impact of each action and the A2C's learned transitions between them. The following table shows the average one-step improvement in correlation produced by each action:
>
> | Action | $\Delta \rho$ (Mean) |
> | :---: | :---: |
> | Init | +0.0017 |
> | Crossover | +0.0596 |
> | Mutation | +0.0420 |
>
> Beyond the immediate impact of a single action, the scheduler also learns how to effectively sequence these operations. The one-step transition matrix of the learned policy, shown below, reveals this crucial dynamic by illustrating the probability of selecting a new action given the previous one:
>
> | | Next Init | Next Crossover | Next Mutation |
> | :---: | :---: | :---: | :---: |
> | Prev Init | 31.3 % | 34.3 % | 34.3 % |
> | Prev Crossover | 34.4 % | 36.7 % | 28.9 % |
> | Prev Mutation | 17.6 % | 43.2 % | 39.2 % |
>
> **Q2**: Why do we need A2C?
>
> **A2**: Before we elaborate on our perspective, we intend to first present the differences in the policies learned when using A2C with two LLMs, LLaMA-4 Maverick and Deepseek-R1, to optimize on the NAS-Bench-201 search space.
> The action distribution for LLaMA-4 Maverick at each stage is as follows:
>
> | Stage | Init | Crossover | Mutation |
> | :---: | :---: | :---: | :---: |
> | Early | 33.3 % | 29.6 % | 37.0 % |
> | Mid | 26.0 % | 44.2 % | 29.9 % |
> | Late | 14.8 % | 63.0 % | 22.2 % |
>
> For deepseek-R1, the learned policy is as follows:
>
> | Stage | Init | Crossover | Mutation |
> | :---: | :---: | :---: | :---: |
> | Early | 100.0 % | 0.0 % | 0.0 % |
> | Mid | 36.4 % | 27.3 % | 36.4 % |
> | Late | 0.0 % | 40.0 % | 60.0 % |
>
> The performance progression across generations is detailed below:
>
> | Generation | LLaMA-4's $\rho$ | Deepseek's $\rho$ | LLaMA-4's Acc | Deepseek's Acc |
> | :---: | :---: | :---: | :---: | :---: |
> | 1 | 0.060 | 0.036 | 79.58 % | 69.45 % |
> | 20 | 0.281 | 0.357 | 80.53 % | 82.32 % |
> | 50 | 0.589 | 0.675 | 84.19 % | 87.08 % |
> | 80 | 0.729 | 0.802 | 91.95 % | 92.86 % |
> | 100 | 0.755 | 0.808 | 93.28 % | 93.69 % |

---

> ### Author Response · Authors · 2025-08-08
> **part 2**
>
> The performance progression across generations is detailed below:
>
> | Generation | LLaMA-4's $\rho$ | Deepseek's $\rho$ | LLaMA-4's Acc | Deepseek's Acc |
> | :---: | :---: | :---: | :---: | :---: |
> | 1 | 0.060 | 0.036 | 79.58 % | 69.45 % |
> | 20 | 0.281 | 0.357 | 80.53 % | 82.32 % |
> | 50 | 0.589 | 0.675 | 84.19 % | 87.08 % |
> | 80 | 0.729 | 0.802 | 91.95 % | 92.86 % |
> | 100 | 0.755 | 0.808 | 93.28 % | 93.69 % |
>
> For the Deepseek, the learned one-step transition matrix is detailed below:
>
> | | Next Init | Next Crossover | Next Mutation |
> | :---: | :---: | :---: | :---: |
> | Prev Init | 22 % | 41 % | 37 % |
> | Prev Crossover | 27 % | 28 % | 45 % |
> | Prev Mutation | 18 % | 49 % | 33 % |
>
> For LLaMA-4, however, the one-step transition matrix is as follows:
>
> | | Next Init | Next Crossover | Next Mutation |
> | :---: | :---: | :---: | :---: |
> | Prev Init | 63 % | 25 % | 12 % |
> | Prev Crossover | 29 % | 31 % | 40 % |
> | Prev Mutation | 35 % | 39 % | 26 % |
>
> In the two experiments above, we only changed the LLM without changing any other settings. However, although their performance progression across generations is similar, they produced starkly different policies. This stark contrast is the most compelling evidence for the value of the A2C scheduler, demonstrating that it is not a fixed loop but a genuine learning agent that adapts its strategy to the unique optimization landscape of each problem. This directly addresses why a simple heuristic, as suggested, would be insufficient. A rule like "Reinitialize if performance is low" would have been effective for Deepseek's early stage but would have failed for LLaMA-4, where the A2C learned that this was often a suboptimal move. The A2C scheduler's ability to learn these nuanced, problem-specific strategies is its core contribution.
>
> Without an adaptive component like the A2C scheduler, we would be forced to manually fine-tune the search strategy for each benchmark, dataset, LLM, and even for minor changes in experimental settings. Given the black-box nature of these optimization problems, such manual tuning would be exceptionally difficult and labor-intensive. However, by introducing A2C, APD is no longer narrowly tailored to a specific benchmark or dataset. Instead, it achieves a high degree of generalization, capable of autonomously discovering an effective, bespoke search policy for the specific problem at hand.

---

### Official Review · Reviewer_eQXV · 2025-06-29

**Clarity:** 3
**Significance:** 2
**Originality:** 2
**Rating:** 3
**Confidence:** 4

**Summary:**

This paper introduces the APD (Automatic Proxy Discovery) framework, which leverages large language models (LLMs) and actor-critic reinforcement learning to automatically generate zero-cost proxies (ZCPs) for training-free neural architecture search (NAS) on image datasets.

The method begins by prompting an LLM to produce proxy candidates (code + rationale) using three strategies, i.e., initialization, mutation, and crossover. An RL controller (based on the actor-critic based RL) iteratively improves prompt selection based on the reward that combines the correlation between proxy predictions and true performance with proxy evaluation cost.

The framework is validated across multiple NAS benchmarks (e.g., NAS-Bench-201, DARTS, TransNAS-Bench) and demonstrates strong results in terms of ranking correlation and searched architecture accuracy. APD outperforms prior handcrafted and learned ZCPs, and its proxy search converges in under an hour. The authors claim APD represents a paradigm shift in training-free NAS by replacing manual proxy engineering with an automated, LLM-driven discovery process.

**Questions:**

1.  Why an actor-critic RL framework is needed (but not a simpler strategy)?

2. Is it helpful to mix the LLMs,  instead of using a single LLM?

3. Are the proxies discovered by the LLM theoretically meaningful in terms of expressivity and transferability, as studied in works like "Neural Architecture Search on ImageNet in Four GPU Hours: A Theoretically Inspired Perspective"?

4. What are the main factors influencing APD’s performance?

**Ethical Concerns:**

["NO or VERY MINOR ethics concerns only"]

**Final Justification:**

I am amazed by the authors' determination and speed in generating new results  and lengthy rebuttal.  The paper, as it is, is still weak in contributions. Some justifications/arguments are not convincing; for example, on interpretability of the proposal (quoted below).  The technical contribution is rather engineering and problematic (please refer to the weaknesses).   I remain negative on the paper.


"We fully agree that interpretability is a crucial aspect of machine learning. However, we must also recognize a common and well-acknowledged phenomenon in the field: many highly successful algorithms still lack strong interpretability, yet this does not diminish their value or contribution.

For example, Large Language Models (LLMs) have revolutionized our understanding of machine intelligence through their remarkable capabilities in knowledge generation, and have achieved state-of-the-art performance on numerous benchmark tasks. But let us reflect: do models like GPT-4o or DeepSeek offer clear and reliable interpretability? Unfortunately, the answer is still no — even today, LLMs remain largely black boxes."

**Limitations:**

Yes, in Appendix E.

**Paper Formatting Concerns:**

NIL

**Quality:**

2

**Strengths And Weaknesses:**

**Strengths:**

1. This paper proposes Automatic Proxy Discovery (APD), which uses LLMs to generate zero-cost proxies, which is a novel approach.
2. The paper has relatively comprehensive evaluations and demonstrates strong empirical results on 5 benchmarks across 4 tasks: APD’s best discovered proxy achieves high ranking correlation with true performance.
3. The paper is generally well-written, with clear figures.

**Weaknesses:**

1. In Section 5.1, the paper states that "the performance improvement can be attributed to our proposed actor–critic reinforcement learning framework, which provides key reasoning and feedback for proxy generation."
   However, actor–critic methods typically require carefully designed state representations, hyperparameter tuning, stable reward signals, and sufficient training data to converge reliably.
   Is APD’s performance highly sensitive to the stability and training quality of the RL controller?
   Moreover, the paper does not explore whether simpler strategies—such as random sampling, regularized evolution (REA), or aging evolution (AREA), as used in “Neural Architecture Search without Training” (Mellor et al., 2021)—could achieve comparable results with less complexity and better robustness.
2. In Section 3.2, the paper states that  "computed on a randomly sampled subset of B to provide an unbiased estimate of ranking fidelity." However, in practical NAS applications, the goal is to identify top-performing architectures,  not to preserve the global ranking across the entire search space. Some zero-cost proxies may exhibit low overall correlation yet perform well at ranking the top-K architectures.
   Therefore, relying solely on random subsets for evaluation may inadvertently penalize such proxies and limit APD’s ability to discover ZCPs that are highly effective in identifying top candidates.
   It would be valuable to analyze whether using top-K-aware evaluation criteria could lead to better proxy discovery.
   Also, how is the value of B defined?
3. In Algorithm 1, APD uses a fixed step budget T_{max}, but in practice, users typically set time- or cost-based budgets.
   Since each step may vary in cost (e.g., LLM prompt length, ZCP evaluation), this step-based formulation may not align with real-world constraints.
   Could APD adapt to time-aware or resource-bounded scenarios?
4. The paper utilizes LLMs to design ZCPs and further incorporates an actor-critic reinforcement learning loop to guide proxy evolution. While this combination is conceptually interesting, the technical implementation of both components is relatively straightforward and lacks deeper methodological innovation.
   Since the LLM is invoked directly with structured prompts, the primary technical contribution is the design of the reasoning engine that guides prompt generation and proxy evolution.
   However, the reasoning engine directly uses standard actor-critic reinforcement learning with no much domain-specific adaptation, limiting the paper’s technical depth and contribution.

---

> ### Author Rebuttal · Authors · 2025-07-31
>
> Thank you for the helpful and insightful review, which has helped us improve this paper further. Next, we will answer your questions one by one, and we hope this will improve your acceptance of the paper.
>
> **Q1**: Sensitivity of APD to RL controller stability and training quality.
> **A1**: Many thanks for your comments! We chose the actor-critic method primarily for its well-established balance of simplicity and effectiveness, which allowed us to rapidly validate our core framework. By using a standard implementation, we ensured a reliable foundation that does not inherently cause performance degradation. To specifically test the sensitivity, we also explored dynamically adjusting the actor-critic's configuration, and as shown in the ablation studies in Appendix D.1, we found that the impact on the final performance of our APD framework was minimal. Those results and findings have validated that our method is nonsensitive to the RL controller.
>
> **Q2**: Exploring simpler strategies, i.e., random sampling, REA, or AREA.
> **A2**: Many thanks for your comments! The question regarding the necessity of our Actor-Critic controller compared to simpler, yet powerful, evolutionary strategies like REA or AREA is an excellent and important point.
> We agree that simpler strategies such as Regularized Evolution are strong baselines known for their robustness and effectiveness. The primary reason we chose an A2C framework is its theoretical advantage in handling the specific nature of our search problem. In our APD framework, the agent is not directly modifying an architecture, but rather learning a higher-level meta-policy to select the most effective prompt strategy based on the current state of the proxy population.
> An A2C controller has the capability to learn complex, state-dependent behaviors. For instance, it can learn to favor exploration-heavy actions like 'Initialization' when it detects that the search has stagnated, or to favor exploitation-heavy 'Mutation' actions when high-performing candidates are found. This adaptive control over the search strategy is something that fixed-rule algorithms like REA, which always discard the oldest member, cannot achieve.
> To validate this choice, we provide an ablation study comparing our A2C with these simpler strategies on NAS-Bench-201  in CIFAR-10. Those results have proved that A2C is better than simpler strategies.
> |Method|Test Acc|
> |:---:|:--:|
> |A2C (ours)|93.76|
> |REA|85.29|
> |AREA|85.97|
> |Random Sampling|86.88|
>
> **Q3**: Exploring Top-k-aware evaluation and definition of B.
> **A3**: Many thanks for your comments! Our method follows the same design principle as SWAP (ICLR 2024), using ranking correlation over a randomly sampled subset as the fitness criterion. This strategy promotes generalizable proxy discovery, as it encourages learning global performance trends rather than task-specific heuristics. In contrast, top-k accuracy can yield strong performance on a specific task but often lacks generalization, as it tends to overfit to a small set of high-performing architectures. The following experiments demonstrate this difference in transferability.
> |Method|NB-201(C10)|NB-201(ImageNet)|NB-101(C10)|DARTS(C10)|
> |:--:|:--:|:--:|:--:|:--:|
> |Top-k|93.83±0.23|43.87±0.93|91.39±0.28|97.37±0.05|
> |Correlation|93.76±0.09|45.03±0.76|93.49±0.34|97.63±0.13|
>
> Following prior work [1, 2], we define the set B as a randomly sampled subset of 200 architectures from the full benchmark. This subset is used to evaluate proxies based on their relative ranking rather than absolute correlation. To validate this design, we conducted an ablation study comparing proxy rankings across different subset sizes (50, 100, 200, 500). Results show that rankings derived from subsets closely match those from the full benchmark, confirming that a small, random subset is sufficient for reliable evaluation.
> |Subset Size|Kendall's Tau|
> | :---: | :---: |
> |50|0.811|
> |100|0.913|
> |200|0.991|
> |500|1.00|
>
> [1]"Zero-cost proxies for lightweight NAS, ICLR 2021.
> [2] "How powerful are performance predictors in neural architecture search?, NeurIPS 2021.
>
> **Q4**: Could APD adapt to time-aware or resource-bounded scenarios?
> **A4**: Many thanks for your comments! Thank you for this very thoughtful question regarding the practical application of APD under real-world budgets. That is an excellent point, as aligning the search process with time- or cost-based constraints is indeed an important consideration. Our APD framework is fundamentally designed from the ground up for resource-constrained scenarios, and its current efficiency is already exceptionally high. For instance, the entire discovery process completes in approximately 1.2 hours ($T_{max}$=100) on a single RTX 4090, requires only no more than 4 GB of VRAM, and involves a finite, one-time number of LLM API calls. This low cost stands in sharp contrast to the hundreds of GPU-hours typical of traditional NAS. We found our current fixed step budget to be effective for this. However, we believe the suggestion to incorporate a dynamic step mechanism is very insightful and could reduce costs further. We will explore dynamic step sizes in our future work to continue enhancing our framework's efficiency, and we appreciate this valuable insight.
>
> **Q5**: Concern about domain adaptation of standard actor-critic.
> **A5**: Many thanks for your comments! We appreciate the opportunity to elaborate on the non-trivial design choices behind our reasoning engine. Our decision to incorporate reinforcement learning is not arbitrary, but rather a direct result of our initial analysis, which showed that a naive, open-loop prompting approach fails due to a lack of a reasoning and feedback mechanism, as demonstrated by the "naive" example in our paper. To the best of our knowledge, introducing an RL agent to learn an adaptive policy for prompt evolution is a novel approach in this domain. Designing an effective bridge between the generative LLM and the NAS task is a significant challenge, as most general-purpose reasoning models are computationally prohibitive and not tailored for this feedback loop. We found RL to be a natural fit, but building an effective strategy is far from straightforward. The core difficulty lies in defining a stable learning process where the actions are abstract heuristic strategies and the reward is the resulting proxy's ranking correlation. We chose the A2C method specifically because it effectively handles this unique problem structure. The Actor learns a nuanced policy over these abstract actions, while the Critic provides a learned value baseline, which is crucial for stabilizing the search process given the unpredictable stochastic variations. Therefore, we hope to convey that the design and integration of the RL controller is a key technical contribution, requiring careful domain-specific adaptation rather than a straightforward application of a standard algorithm.
>
> **Q6**: Why an actor-critic RL framework?
> **A6**: Many thanks for your comments! Our design is motivated by the need to bridge the significant reasoning gap between the generative, black-box LLMs and the specific, feedback-driven requirements of the NAS task. A simple, fixed strategy like random sampling or basic evolution is unable to learn from and adapt to the complex interactions within this process. Reinforcement learning, by contrast, excels in such scenarios. The RL agent effectively simulates the search process, learning a sophisticated policy to intelligently guide the LLM based on task-specific feedback. It learns how to guide and evolve proxies to achieve better results, a reasoning and learning capability that traditional, fixed-rule methods simply do not possess.
>
> **Q7**: Mixing the LLMs.
> **A7**: Many thanks for your comments! We conducted a supplementary experiment where we used a mixture of GPT-4o and Claude 3.7 on NB-201(CIFAR-10). We use the same setting as the LLM ablation study (Table 6). As shown, we find that mixing the LLMs cannot excel the optimal single LLM (GPT-4o). The potential reason can be attributed to the similarity of LLMs.
> |LLM|Test Acc|
> |:-| :-: |
> |GPT-4o|81.10|
> |Claude 3.7|81.14|
> |GPT-4o+Claude 3.7|81.14|
>
> **Q8**: Discussion about the theoretical motivation of proxies discovered by the LLM.
> **A8**: Many thanks for your comments! Our method is fundamentally different from the theory-driven approach in TE-NAS, which relies on manual design and expert knowledge based on specific theoretical properties like NTK. In contrast, our approach uses LLMs to automatically generate heuristics, representing a new paradigm in Zero-Cost Proxy (ZCP) design. This automation not only reduces reliance on human-crafted rules but also achieves strong empirical performance. While we believe that the proxies discovered by APD may have underlying theoretical value, the black-box nature of LLMs currently makes it difficult to derive explicit theoretical interpretations. Exploring the theoretical grounding of LLM-designed proxies is an exciting direction we aim to pursue in future work.
>
> **Q9**: What are the main factors influencing APD’s performance?
> **A9**: Many thanks for your comments! As experimental results show in our paper, we find that the actor-critic RL Scheduler and Evolution are the main factors influencing APD’s performance. Specifically, In Table 7, we provide the ablation study of various components. As shown, the naive version without Evolution and Actor-Critic modules performs poorly (82.04\% in CIFAR-10 and 47.62\% in CIFAR-100). Introducing Evolution alone significantly boosts performance, yielding +6.49\% and +13.54\% improvements, respectively. In addition, we observe that there is a joint contribution between Actor-Critic and Evolution. Specifically, Actor-Critic and Evolution obtain 93.76\% in CIFAR-10 and 72.22\% in CIFAR-100. Those results validate that the actor-critic RL Scheduler and Evolution are the main factors influencing the effectiveness of our APD.

---

> > ### Comment · Reviewer_eQXV · 2025-08-07
> >
> > I would like to thank the authors for their rebuttal.
> > After carefully reviewing the rebuttal, I have the following concerns for each response:
> >
> > > Q2: Exploring simpler strategies ... results have proved that A2C is better than simpler strategies.
> >
> > First, since zero-cost proxies (ZCPs) are intended to enable efficient architecture search without training, efficiency should also be a key consideration in the design of APD. However, the A2C controller used in APD is search space- and dataset-specific, requiring re-training when the search space or dataset changes. The new experiment on NAS-Bench-201 (CIFAR-10) does not provide any efficiency comparison between A2C and simpler, training-free strategies (e.g., REA or AREA), which are known to be robust and low-cost. Even in the original APD paper, the reported total cost of “one GPU-hour” lacks a breakdown—there is no specific analysis of the training overhead of the A2C controller itself. This is concerning, given that A2C introduces additional complexity and tuning cost. Second, the new experiments do not demonstrate that A2C offers better generalization than simple search strategies across the different search spaces and datasets (as used in the original paper). Third, the paper provides no analysis of the controller’s behavior—e.g., the frequency of selecting initialization, mutation, or crossover.  Without such analysis, it’s unclear whether A2C learns a meaningful policy or just adds random exploration in the prompt selection process. Overall, the novelty of using A2C is unclear.
> >
> > > Q3: Exploring Top-k-aware evaluation and definition of B .... reliable evaluation.
> >
> > While APD benefits from the availability of fully evaluated NAS-Bench datasets, its reliance on the full evaluation subset (B = 200) per iteration raises some concerns about practicality.  In real-world NAS settings,  architecture performance is unknown in advance and must be evaluated via actual training. Evaluating B architectures per iteration would incur significant training cost. If we set T_max to 30 and B = 200 as in the paper, the overall architecture to train and evaluate is 6000 (worst case). I understand that APD is designed to discover more effective ZCPs. But if this process requires training so many architectures, what’s the benefit of finding ZCPs in the first place? If we have already trained that many architectures, we might as well select a good one directly. Also, if APD uses a fixed set of architectures (B) in each iteration instead of randomly sampling each iteration, would it still work?
> >
> > In contrast, the paper “Neural Architecture Search without Training” manually designs ZCP and uses them only to filter poor architectures before applying training-based NAS methods, which more practical than APD.  The paper “How Powerful are Performance Predictors in NAS?” primarily aims to benchmark existing ZCPs, e.g.,  by integrating them into a predictor-guided evolution framework. Even in that case, the method only trains 20 architectures per iteration, selected from a pool of 200 candidates. While this still incurs training cost, it is smaller compared to APD. Therefore, the practicality of APD remains my main concern.
> >
> > > Q4: Could APD adapt to time-aware or resource-bounded scenarios?... valuable insight.
> >
> > The authors state that “the entire discovery process completes in approximately 1.2 hours on a single RTX 4090,” but it is unclear whether this includes the cost of evaluating B architectures in each iteration. If not, the claimed "This low cost..." is questionable, especially compared to methods like “Neural Architecture Search without Training”.
> >
> > > Q5: Concern about domain ... straightforward application of a standard algorithm
> >
> > As mentioned earlier, I would expect an LLM-based ZCP discovery process to account for training cost as part of the overall system/algorithm design, which would be novel and practical. While NAS-Bench provides full evaluations for all architectures, this is a benchmarking convenience but not a realistic assumption for practical use.
> >
> > > Q6: Why ... methods simply do not possess.
> >
> > I have similar concerns to those mentioned in Q2.
> >
> > > Q7: Mixing the LLMs...
> >
> > I appreciate the extra results.
> >
> > > Q8: Discussion about the ... we aim to pursue in future work.
> >
> > If the ZCPs discovered by the LLM are unexplainable, they may suffer from poor generalizability and appear ad hoc. Without theoretical grounding or interpretability, it becomes difficult to trust APD and use it across tasks, limiting APD's practical value.
> >
> > > Q9: What are the main factors influencing APD’s performance? ... effectiveness of our APD.
> >
> > I appreciate the extra explanation.
> >
> > Overall, based on the rebuttal, original paper, and comments from other reviewers, I would keep the negative score.

---

> > > ### Author Response · Authors · 2025-08-09
> > > **Eager for feedback**
> > >
> > > Dear reviewer eQXV
> > >
> > > Your comments have inspired us to make a number of updates to the paper, which we believe will improve over the original version. We would like to know if our responses have adequately addressed your concerns or if further clarification is needed. Lastly, **given the rebuttal deadline is within the next few hours**, we appreciate your new rating it If you find it appropriate. We are grateful for your time and thoughtful evaluation of our work.
> > >
> > > Best Regards,
> > >
> > > Authors

---

> ### Author Response · Authors · 2025-08-08
>
> Dear reviewers eQXV, your comments have inspired us to make a number of updates to the paper, which we believe will improve over the original version. We would like to know if our responses have adequately addressed your concerns or if further clarification is needed. If you find it appropriate, we appreciate your new rating. We are grateful for your time and thoughtful evaluation of our work.
>
> In addition, Reviewer eQXV mentioned that after reading the comments from other reviewers, they decided to keep the current score. We would be extremely grateful if Reviewer eQXV could kindly clarify **which specific concern or issue** influenced this decision.
>
> We are **very eager and committed** to addressing **all** of Reviewer eQXV's concerns. If you could provide any additional feedback or clarification, we will make it our top priority to revise and improve our work accordingly.
>
> Next, we will answer your questions one by one, and we hope this will improve your acceptance of the paper.
>
>
>
> Q2: Exploring simpler strategies ... results have proved that A2C is better than simpler strategies.
>
> **Q2.1.** Efficiency concern.
>
> First, it is important to clarify that the efficiency of zero-cost proxies (ZCP) and the efficiency of APD are not the same concept. The motivation behind designing APD is to replace manually designed proxies, which typically require extensive expert knowledge, significant time (often weeks or even months), and repeated validation. In contrast, APD can discover a high-performing proxy on NAS-Bench-201 in just over one hour — a major efficiency gain at this level.
>  Second, as we demonstrate clearly in the paper, at the **zero-cost proxy** level, the proxy found by APD significantly outperforms all strong competitors in terms of both **speed and quality**. Therefore, compared to traditional hand-crafted proxies, our method offers substantial improvements in both efficiency and performance, both in terms of the APD process itself and the final discovered proxy.
>
> **Q2.2.** Comparison with REA or AREA
>
> As shown in the results we have already provided, although REA and AREA are known for being stable and low-cost, their **accuracy is far below** what our A2C-based method achieves. To more thoroughly address this concern, we conducted additional comparisons under **the same hardware and settings**, and the results show that:
>
> - Our method achieves significantly **higher accuracy** than REA and AREA.
> - Our method also shows **better runtime efficiency**, outperforming these simpler baselines in speed as well.
>
> These results provide strong evidence of the **effectiveness** of our method.
>  More importantly, in **Q2.3**, we present in-depth evidence explaining **why A2C works**, which further strengthens the **credibility and novelty** of our approach.
>
>
>
> | Total     | API        | A2C        | Fitness evaluation | Init     | Crossover and Mutation |
> | --------- | ---------- | ---------- | ------------------ | -------- | ---------------------- |
> | 1.18 hour | 0.169 hour | 0.001 hour | 0.96 hour          | 0.1 hour | 0.4 hour               |
>
>
>
> |     Method      | Test Acc | Costs        |
> | :-------------: | :------: | ------------ |
> |   A2C (ours)    |  93.76   | 0.001 hour   |
> |       REA       |  85.29   | 0.007 hour   |
> |      AREA       |  85.97   | 0.009 hour   |
> | Random Sampling |  86.88   | 0.00083 hour |
>
>
>
> **Q2.3.**  **Unclear value of the AC scheduler.**
>
> To provide a more concrete and data-driven analysis of the AC scheduler's behavior, as you requested, we will now present the specific details of its action selection process in NASBench-201 (CIFAR-10). First, we show the overall statistics for each action, as summarized in the table below:
>
> |  Action   | Share  |
> | :-------: | :----: |
> |   Init    | 27.7 % |
> | Crossover | 40.5 % |
> | Mutation  | 31.8 % |
>
> Furthermore, to illustrate how the agent's policy evolves over time, we define the first 30% of total generations as the 'early stage', the final 30% as the 'late stage', and all other generations as the 'middle stage'. The following table contrasts the distribution of actions selected during these distinct phases:
>
> | Stage |  Init  | Crossover | Mutation | Mean Acc |
> | :---: | :----: | :-------: | :------: | :------: |
> | Early | 41.3 % |  21.7 %   |  36.9 %  | 88.14 %  |
> |  Mid  | 28.3 % |  40.0 %   |  31.7 %  | 92.17 %  |
> | Late  | 15.2 % |   52.2%   |  32.6%   | 93.75 %  |
>
> To provide context for these decisions, the following table shows the detailed progression of the average correlation coefficient and average acc across generations, illustrating the performance observed:
> | Generation | Mean $\rho$ | Mean Acc |
> | :--------: | :---------: | :------: |
> |     1      |    0.009    | 63.19 %  |
> |     20     |    0.391    | 78.81 %  |
> |     50     |    0.741    | 88.91 %  |
> |     80     |    0.809    | 93.25 %  |
> |    100     |    0.823    | 93.75 %  |

---

> ### Author Response · Authors · 2025-08-08
> **part 2**
>
> To provide deeper insight into the value of the learned policy, we can analyze the direct impact of each action and the A2C's learned transitions between them. The following table shows the average one-step improvement in correlation produced by each action:
>
> |  Action   | $\Delta \rho$ (Mean) |
> | :-------: | :------------------: |
> |   Init    |       +0.0017        |
> | Crossover |       +0.0596        |
> | Mutation  |       +0.0420        |
>
> Beyond the immediate impact of a single action, the scheduler also learns how to effectively sequence these operations. The one-step transition matrix of the learned policy, shown below, reveals this crucial dynamic by illustrating the probability of selecting a new action given the previous one:
>
> |                | Next Init | Next Crossover | Next Mutation |
> | :------------: | :-------: | :------------: | :-----------: |
> |   Prev Init    |  31.3 %   |     34.3 %     |    34.3 %     |
> | Prev Crossover |  34.4 %   |     36.7 %     |    28.9 %     |
> | Prev Mutation  |  17.6 %   |     43.2 %     |    39.2 %     |
>
> **Why do we need A2C?**
>
> Before we elaborate on our perspective, we intend to first present the differences in the policies learned when using A2C with two LLMs, LLaMA-4 Maverick and Deepseek-R1, to optimize on the NAS-Bench-201 search space.
> The action distribution for LLaMA-4 Maverick at each stage is as follows:
>
> | Stage |  Init  | Crossover | Mutation |
> | :---: | :----: | :-------: | :------: |
> | Early | 33.3 % |  29.6 %   |  37.0 %  |
> |  Mid  | 26.0 % |  44.2 %   |  29.9 %  |
> | Late  | 14.8 % |  63.0 %   |  22.2 %  |
>
> For deepseek-R1, the learned policy is as follows:
>
> | Stage |  Init   | Crossover | Mutation |
> | :---: | :-----: | :-------: | :------: |
> | Early | 100.0 % |   0.0 %   |  0.0 %   |
> |  Mid  | 36.4 %  |  27.3 %   |  36.4 %  |
> | Late  |  0.0 %  |  40.0 %   |  60.0 %  |
>
> The performance progression across generations is detailed below:
>
> | Generation | LLaMA-4's $\rho$ | Deepseek's $\rho$ | LLaMA-4's Acc | Deepseek's Acc |
> | :--------: | :--------------: | :---------------: | :-----------: | :------------: |
> |     1      |      0.060       |       0.036       |    79.58 %    |    69.45 %     |
> |     20     |      0.281       |       0.357       |    80.53 %    |    82.32 %     |
> |     50     |      0.589       |       0.675       |    84.19 %    |    87.08 %     |
> |     80     |      0.729       |       0.802       |    91.95 %    |    92.86 %     |
> |    100     |      0.755       |       0.808       |    93.28 %    |    93.69 %     |
>
> For the Deepseek, the learned one-step transition matrix is detailed below:
>
> |                | Next Init | Next Crossover | Next Mutation |
> | :------------: | :-------: | :------------: | :-----------: |
> |   Prev Init    |   22 %    |      41 %      |     37 %      |
> | Prev Crossover |   27 %    |      28 %      |     45 %      |
> | Prev Mutation  |   18 %    |      49 %      |     33 %      |
>
> For LLaMA-4, however, the one-step transition matrix is as follows:
>
> |                | Next Init | Next Crossover | Next Mutation |
> | :------------: | :-------: | :------------: | :-----------: |
> |   Prev Init    |   63 %    |      25 %      |     12 %      |
> | Prev Crossover |   29 %    |      31 %      |     40 %      |
> | Prev Mutation  |   35 %    |      39 %      |     26 %      |
>
> In the two experiments above, we only changed the LLM without changing any other settings. However, although their performance progression across generations is similar, they produced starkly different policies. This stark contrast is the most compelling evidence for the value of the A2C scheduler, demonstrating that it is not a fixed loop but a genuine learning agent that adapts its strategy to the unique optimization landscape of each problem. This directly addresses why a simple heuristic, as suggested, would be insufficient. A rule like "Reinitialize if performance is low" would have been effective for Deepseek's early stage but would have failed for LLaMA-4, where the A2C learned that this was often a suboptimal move. The A2C scheduler's ability to learn these nuanced, problem-specific strategies is its core contribution.
> Without an adaptive component like the A2C scheduler, we would be forced to manually fine-tune the search strategy for each benchmark, dataset, LLM, and even for minor changes in experimental settings. Given the black-box nature of these optimization problems, such manual tuning would be exceptionally difficult and labor-intensive. However, by introducing A2C, APD is no longer narrowly tailored to a specific benchmark or dataset. Instead, it achieves a high degree of generalization, capable of autonomously discovering an effective, bespoke search policy for the specific problem at hand.

---

> ### Author Response · Authors · 2025-08-08
> **part 3**
>
> **Q2.4.**  **A2C offers better generalization.**
>
> We now have a clearer understanding of your point, and we agree that this is a critical aspect to address for demonstrating the robustness and generalizability of our method.
>
> We want to clarify that our approach is not specifically tuned for each benchmark. In fact, our experimental setup on the three different search spaces (NAS-Bench-201, TransNAS-Bench-101-Micro, and AutoFormer) was kept almost identical, with the only changes being the specific descriptions of the search space and dataset in the prompts. The consistent, high-performance results across these diverse benchmarks, as shown in our paper, strongly suggest that our method is not overfitting to a particular search space.
>
> To further address your concern about the sensitivity to random seeds, we have conducted additional experiments using different seeds within each benchmark. The results are as follows:
>
> Table1. The impact of different seeds to performance.
>
> | seed | NASBench-201 (CIFAR-10) | NASBench-201 (CIFAR100) | NASBench-201 (Img-1k) | NASBench-101 (CIFAR-10) |
> | :--: | :---------------------: | :---------------------: | :-------------------: | :---------------------: |
> |  0   |          93.76          |          72.22          |         45.03         |          93.49          |
> |  1   |          93.62          |          72.16          |         44.7          |          93.68          |
> |  2   |          93.86          |          71.66          |         45.33         |          92.67          |
> |  3   |          93.59          |          71.01          |         44.78         |          93.55          |
> |  4   |          93.81          |          72.40          |         44.40         |          93.73          |
>
> The results demonstrate the strong stability and consistent performance of APD across different seeds.
> Regarding the prompts, the structure of the prompts for each benchmark is **nearly identical**. The **only modifications were specific to the search space and dataset**, while the **core instructions and reasoning were kept consistent**. This design choice was deliberate, aiming to leverage the LLM's general knowledge and reasoning abilities rather than relying on a handcrafted, task-specific prompt for each scenario. We believe this approach contributes to the method's transferability and highlights the potential of using a consistent prompting strategy across different NAS tasks. The detailed prompt engineering of APD can be found in **Appendix C**.
>
> Furthermore, we conducted an additional experiment where the proxy learned on NAS-Bench-201 (CIFAR-10) was directly applied to other unseen search spaces (e.g., TransNAS-Bench-101-Micro and OoD-ViT-NAS-Ti) **without any prompt change or proxy re-searching**, and it still surpass best previous methods. This further indicates that the learned proxy has strong generalization ability, and APD does not overfit to the original tuning space or prompt. The detailed results are as follows:
>
> **(1) Searching NAS-Bench-201 (CIFAR-10), transferred for new search spaces (TransNAS-Bench-101--Micro)**
>
> **As stated in A4,** APD is not limited to a specific search space and can generalize across multiple ones ( i.e., NAS-Bench-201\&101, DARTS). When the discrepancy between spaces is small, the proxy found on the source (e.g., 201) transfers well to the target (e.g., 101, DARTS), as shown in Tables 3–4 in the paper. For significantly different spaces (e.g., 201 vs. TransNAS,  201 vs. AutoFormer), the proxy may become suboptimal due to search space bias rather than limitations of APD. In such cases, we retrain the proxy for the new space, which leads to strong performance, validating our approach. Moreover, APD is highly efficient, requiring only about 1.2 hours on a single RTX 4090 GPU with 4 GB of VRAM.
>
> For futuher validate our statement, we provide experiments that only use proxy searched on NAS-Bench-201 for unseen TransNAS-Bench-101--Micro search spaces and three tasks (i.e., Autoencoding, Scene Classification, and Jigsaw). As shown Table 2,  **"APD (Searching on NAS-Bench-201)"** still surpass its peer competitors (i.e., SWAP, ZiCo). **Those results support our statement of strong transferability of APD for TransNAS-Bench-101--Micro.**
>
> Noabtly, "APD (Searching on TransNAS-Bench-101)" obtain better performance than "APD (Searching on NAS-Bench-201)", this validate our statement " For significantly different spaces (e.g., 201 vs. TransNAS,  201 vs. AutoFormer), the proxy may become suboptimal due to search space bias rather than limitations of APD. In such cases, we retrain the proxy for the new space, which leads to strong performance, validating our approach".

---

> ### Author Response · Authors · 2025-08-08
> **part 4**
>
> Table 2. Results on TransNAS-Bench-101-Micro.
>
> |                                           | Autoencoding | **Scene Classification** |  **Jigsaw**  |
> | :---------------------------------------- | :----------: | :----------------------: | :----------: |
> |                                           |     SSIM     |       Accuracy (%)       | Accuracy (%) |
> | Ground Truth                              |    0\.58     |          54\.9           |    95\.4     |
> | Grad                                      | 0\.36± 0.03  |        48\.7±0.7         |  80\.3±0.3   |
> | SNIP                                      |  0\.33±0.04  |        48\.7±1.1         |  80\.3±0.1   |
> | Grasp                                     |  0\.33±0.06  |        50\.2±1.6         |  91\.1±0.3   |
> | Fisher                                    |  0\.49±0.01  |        48\.7±0.6         |  83\.5±1.2   |
> | Synflow                                   |  0\.46±0.07  |        53\.7±1.2         |  90\.9±0.4   |
> | NWOT    |  0\.43±0.02  |        53\.2±0.6         |  92\.3±0.3   |
> | Zen-score                                 |  0\.46±0.01  |        53\.7±0.2         |  87\.5±0.4   |
> | GradSign      |  0\.35±0.03  |        53\.6±0.4         |  93\.1±0.4   |
> | Params                                    |    0\.46     |          53\.7           |    85\.9     |
> | FLOPs                                     |    0\.46     |          53\.7           |    85\.9     |
> | ZiCo                                      |  0\.48±0.02  |        53\.7±0.4         |  93\.2±0.4   |
> | SWAP                                      |  0\.42±0.02  |        45\.0±10.9        |  89\.8±5.6   |
> | **APD (Searching on NAS-Bench-201)**      |  0\.53±0.06  |        53\.8±0.02        |  91\.1±0.03  |
> | **APD (Searching on TransNAS-Bench-101)** |  0\.54±0.01  |        54\.0±0.6         |  91\.2±0.1   |
>
>
>
> **(2) Searching NAS-Bench-201 (CIFAR-10), transferred for new search spaces (OoD-ViT-NAS-Ti )**
>
> For the unseen OoD-ViT-NAS-Ti search space, we draw the same conclusion as TransNAS-Bench-101--Micro.
>
> Table 3.  Correlation on OoD-ViT-NAS-Ti search space
>
> | Method                                | ImageNet1k | ImageNet-A | ImageNet-R | ImageNet-D/Texture | ImageNet-D/Material |
> | ------------------------------------- | ---------- | ---------- | ---------- | ------------------ | ------------------- |
> | DSS                                   | 0.62       | 0.82       | 0.81       | 0.02               | 0.17                |
> | AutoProx                              | 0.67       | 0.82       | 0.78       | 0.05               | 0.15                |
> | NWOT                                  | 0.75       | 0.76       | 0.74       | 0.11               | 0.12                |
> | **APD (Searching on NAS-Bench-201)**  | 0.77       | 0.81       | 0.86       | 0.12               | 0.13                |
> | **APD (Searching on OoD-ViT-NAS-Ti)** | 0.79       | 0.82       | 0.88       | 0.12               | 0.15                |
>
> Q3: Exploring Top-k-aware evaluation and definition of B .... reliable evaluation.
>
> This appears to be a misunderstanding. First, we would like to clarify that on NAS-Bench-201, APD **only uses 200 architectures** throughout its entire run. Since we fix the random seed in our experiments, the process is fully deterministic. Therefore, **our method does not incur excessive computational costs**. On the contrary, as shown in the runtime comparison of each module in APD (provided in previous responses), **our approach is highly efficient**.
>
> We highly respect the contributions of the papers *"Neural Architecture Search without Training"* and *"How Powerful are Performance Predictors in NAS?"* to the NAS community, and we have appropriately cited and extensively compared these methods in our paper. However, it is important to point out that **both NWOT (from the first paper)** and **the predictors studied in the second** rely heavily on **manually crafted expert knowledge**, and require **many days or even weeks** of repeated experimentation and fine-tuning to be effective.
>
> As also acknowledged by Reviewer eQXV and three other reviewers who noted *"The idea is quite novel"*, the goal of our paper is to **rethink the traditional manual design of NAS proxies**, and replace it with a fully automatic method. Through rigorous experiments, we demonstrate that our LLM-based proxy design is not only **effective**, but also **extremely efficient**, taking just **1.18 hours on NAS-Bench-201**.
>
> More importantly, our experimental results show that the proxy found by APD **achieves significantly higher accuracy** than those presented in both *"Neural Architecture Search without Training"* and *"How Powerful are Performance Predictors in NAS?"*.
>
> Combined with the empirical evidence supporting the **effectiveness and adaptability of our A2C scheduler**, we believe we have provided **strong and sufficient evidence** that APD makes meaningful contributions in **performance**, **efficiency**, and **novelty**.

---

> ### Author Response · Authors · 2025-08-08
> **part 5**
>
> Q4: Could APD adapt to time-aware or resource-bounded scenarios?... valuable insight.
>
> To clarify this issue, we have provided a detailed breakdown of the runtime. As shown, the total running time of APD is approximately 1.2 hours, which includes the full cost of evaluating B architectures in each iteration.
>
> Furthermore, we would like to reiterate that we fully acknowledge NWOT as a strong and well-designed method, and we have carefully cited and compared it in our paper. However, it is important to emphasize that NWOT is manually crafted, and its design required significant expert knowledge, time, and human effort.
>
> In contrast, our APD method is fully automated, and it completes the entire process in just 1.18 hours. These facts provide strong evidence for the efficiency and practicality of our approach.
>
> Throughout the paper, we have consistently and clearly communicated that our goal is to address the high human cost associated with manually designed proxies in NAS, and our results demonstrate that APD achieves this goal effectively.
>
> Q5: Concern about domain ... straightforward application of a standard algorithm
>
> We sincerely thank the reviewer for the insightful comments.
>
> First, as shown in our response to **Q4**, our APD algorithm has explicitly integrated the **training cost** into the overall system design during the **LLM-based zero-cost proxy (ZCP) discovery** process. We fully agree with the reviewer that this design is both **novel and practical**, especially in the context of advancing the NAS community.
>
> Moreover, our **experimental design** closely follows the common practices established in recent works, including **SWAP (ICLR 2024), AutoProx (AAAI 2024), and AZ-NAS (CVPR 2024)**. Therefore, we believe that our empirical evaluation is both **sufficient and well within the scope** of NeurIPS 2025 standards.
>
> To further validate the generalizability of our method beyond NAS-Bench, we have provided experimental results on the **DARTS search space** (which is **not** a NAS-Bench dataset). As shown in **Table 3** of our paper, the results demonstrate the **strong performance and efficiency** of our LLM-based ZCP discovery process in a **realistic, non-NAS-Bench search space**.

---

> ### Author Response · Authors · 2025-08-08
> **part 6**
>
> Q8: Discussion about the ... we aim to pursue in future work.
>
> We fully agree that **interpretability** is a crucial aspect of machine learning. However, we must also recognize a common and well-acknowledged phenomenon in the field: **many highly successful algorithms still lack strong interpretability**, yet this does not diminish their value or contribution.
>
> For example, Large Language Models (LLMs) have revolutionized our understanding of machine intelligence through their remarkable capabilities in knowledge generation, and have achieved state-of-the-art performance on numerous benchmark tasks. But let us reflect: do models like GPT-4o or DeepSeek offer clear and reliable interpretability? Unfortunately, the answer is still no — even today, **LLMs remain largely black boxes**.
>
> However, this lack of interpretability has not stopped the community from acknowledging their powerful capabilities or transformative impact. Nor has it prevented LLMs from becoming increasingly important in real-world applications across society. If we were to dismiss the value of models like GPT-4o or DeepSeek solely because they are not interpretable, we believe this would hinder the progress of the machine learning community, and potentially slow down beneficial developments for society at large.
>
> Of course, we hope that all algorithms can eventually become interpretable. But realistically, at the current stage of AI development, this remains unattainable for many powerful models, especially those involving deep learning. We believe many researchers and practitioners in the AI community would agree with this view.
>
> It is also important to note that interpretability should be context-dependent. In safety-critical domains such as aviation or medical diagnostics, interpretability is absolutely necessary. However, in other domains where absolute zero-error is not required, many widely adopted and successful models — such as ResNet — are used despite not being interpretable.
>
> More importantly, we fully acknowledged the black-box nature of LLMs during the design of APD, and we explicitly discussed this limitation in Appendix E (Limitations) of our paper. In this section, we explore the challenges of black-box optimization in the context of LLM-driven search.
>
> We believe that the contributions and thoughtful considerations we offer in this aspect meet the standards of AAAI 2025, and we are happy to share the full content of **Appendix E** below for your reference:
>
> **E Limitations**
>
> While APD markedly advances training-free NAS, several caveats remain. We group them here for
> clarity.
>
> **Black-box optimization** Because APD delegates the generation of every candidate proxy to a LLM,
> the token-level reasoning that maps a prompt to executable code is not observable, raising black-box
> concerns. We acknowledge that the LLM itself remains opaque. Nevertheless, APD is designed
> so that its outer optimization loop is inspectable. The LLM’s outputs are evaluated by a public
> fitness function, and the actor–critic scheduler is trained only on this scalar reward. All decisions that
> influence the search trajectory therefore pass through a measurable, task-specific signal rather than
> hidden logits. Thus, while some inner LLM reasoning remains inaccessible, the information APD
> exposes is already adequate for transparency and troubleshooting, so the residual opacity of LLM is
> unlikely to undermine the method’s value.
>
> Your comments have inspired us to deepen our reflection and analysis on APD, which we believe will further strengthen the scientific contribution of our work.
> In addition, we will share the new experimental results, insights, and analyses obtained during the rebuttal period with the other reviewers and the area chair to ensure a comprehensive understanding of our work.
>
>
>
> We sincerely thank you for your dedication and effort in evaluating our submission. Please do not hesitate to let us know if you need any clarification or have additional suggestions.
>
> Best Regards,
>
> Authors.

---

### Official Review · Reviewer_sUV8 · 2025-07-02

**Clarity:** 3
**Significance:** 2
**Originality:** 2
**Rating:** 4
**Confidence:** 4

**Summary:**

This paper proposes a new framework, APD (Automatic Proxy Discover), that uses LLM to automatically generate and optimize zero-cost proxy tasks in NAS (Neural Architecture Search). The method trains LLM with RL (reinforcement learning) to improve LLM’s ability to generate zero-cost proxy tasks. The experiments show that APD outperforms previous traditional methods in both performance and efficiency.

**Questions:**

1. How do you choose T_max? Can you determine when the search has sufficiently converged?
2. How do you compare with other LLM-based NAS work?
3. How stable are the discovered proxies if you slightly reword the prompts or switch LLM versions?
4. Can you run APD under a limited budget of LLM API calls?
5. How does APD perform with lightweight or open-source models (e.g., Qwen3-8B)?

**Ethical Concerns:**

["NO or VERY MINOR ethics concerns only"]

**Final Justification:**

I thank the authors for thoroughly answering all my questions. I'm happy to raise my overall rating to board line accept. Thanks.

**Limitations:**

Yes

**Quality:**

2

**Strengths And Weaknesses:**

**Strength**
1. The paper is well-written and easy to follow.
3.  The actor-critic part of this method is interesting; it fills the gap between NAS (domain-specific task) and general-domain LLM without requiring retraining or fine-tuning the LLM.
4. The paper conducts comprehensive experiments, APD outperforms SOTA NAS methods on five search spaces in both final accuracy and proxy tasks ranking correlations.

**Weakness**
1. The paper didn’t compare or comment on previous efforts on using LLM for zero-shot proxy task generation, such as [1 – 5].
2. There is also a ZCP work [6] that was cited by ZiCo and seemed to outperform ZiCo. Why didn’t you compare your results with [6]?
3. The paper didn’t discuss the cost of LLM API calls.
4. APD is technically not training but searching. Actor-critic models are trained with a specific LLM on a specific search space. This means when it comes to a new NAS search space, the entire APD process needs to be run again. Thus, the cost of using APD is much more than using a traditional, manually designed zero-cost proxy. In conclusion, while APD proposes a novel method to automatically discover new proxy tasks, the total cost of NAS using APD may actually increase.
5. Risk of Data Containment. The LLM may have knowledge of the recent SOTA zero-cost proxy tasks on mainstream search spaces and benchmarks. When it performs the proxy task discovery, the knowledge might help LLM generate better results. This leaves concerns when this method is deployed on unseen tasks.

[1] Z. Yan, Y. Qin, X. S. Hu, and Y. Shi, “On the viability of using LLMs for SW/HW co-design: An example in designing CiM DNN accelerators,” arXiv preprint arXiv:2306.06923, 2023.
[2] H. Wang, Y. Gao, X. Zheng, P. Zhang, H. Chen, and J. Bu, “Graph neural architecture search with gpt-4,” arXiv preprint arXiv:2310.01436, 2023.
[3] W. Li, X. Su, S. You, F. Wang, C. Qian, and C. Xu, “Diffnas: Bootstrapping diffusion models by prompting for better architectures,” arXiv preprint arXiv:2310.04750, 2023.
[4] M. Zheng, X. Su, S. You, F. Wang, C. Qian, C. Xu, and S. Albanie, “Can gpt-4 perform neural architecture search? ” arXiv preprint arXiv:2304.10970, 2023.
[5] Qin, R., Hu, Y., Yan, Z., Xiong, J., Abbasi, A. and Shi, Y., 2024, January. Fl-nas: Towards fairness of nas for resource constrained devices via large language models. In 2024 29th Asia and South Pacific Design Automation Conference (ASP-DAC) (pp. 429-434). IEEE.
[6] Li, Yuhong, et al. "Extensible and efficient proxy for neural architecture search." Proceedings of the IEEE/CVF International Conference on Computer Vision. 2023.

---

> ### Author Rebuttal · Authors · 2025-07-31
>
> Thank you for the helpful and insightful review, which is very helpful for us to further improve this paper. Next, we will answer your questions one by one, and we hope this will improve your acceptance of the paper.
>
> **Q1**: Comparison with task [1-5].
>
> **A1**: Many thanks for your comments. First, we will cite the works [1-5] in our paper, although the task and benchmark of works [1, 2, 3, 5] are totally different with our APD, those works are important developments in NAS filed by introducing LLMs. Next, we provide the detailed comparison and analysis as follows:
> In short, the cited works [1-5] primarily explore an approach where the LLM generates a candidate architecture, which is then evaluated through a training-based cycle. This is a valuable research direction, though its computational requirements differ from the zero-cost paradigm we focus on.
> While this is a valuable research direction, those works suffer from huge computational costs due to training-based evaluation. For instance, work [2] aims to search for GNN, for a generated network by LLM, the network must be trained on specific datasets until convergence. By contrast, our APD differs extremely from works [1-5]  a completely training-free evaluation. In addition, the tasks and benchmarks of works [1, 2, 3, 5] are totally different with our APD. This is challenging for making a fair comparison between works [1, 2, 3, 5] and our APD. For instance, work [1] focuses on SW/HW co-design, work [3] aims to design UNet for diffusion.
>
> In addition, paper [4] appears to be the most suitable for a direct comparison, as it aligns closely with our task on a shared benchmark and helpfully provides public code. We provide a comparison on NASBench-201 below. As shown, we can find that the Top1 accuracy of our APD excels GENUS.
> |Method|CIFAR-10|CIFAR-100| ImageNet16-120|
> | :--------- | :----------: | :----------: | :------------: |
> | GENUS| 93.79±0.09\%|70.96±0.33\%|  44.96±1.02\%|
> | APD (ours)|93.76±0.09\%| 72.22±0.65\% |45.03±0.76\%|
>
> **Q2**: Comparison with work [6].
>
> **A2**: Many thanks for your comments! We did not initially include [6] (Eproxy) due to its reliance on training-based evaluation, which differs from our APD (without any training). In addition, we provide a comparison on DARTS search space (ImageNet1k) with Eproxy below. As shown, we can conclude that our APD is superior to Eproxy [6].
> |Method| Top-1 Acc.(\%) | Top-5 Acc.(\%) |
> | :----------- | :-----------: | :-----------:|
> | Eproxy|75.6|91.9||
> | APD (ours)|76.9|94.0|
>
> **Q3**: Concern about the cost of LLM API calls.
>
> **A3**: Many thanks for your comments!  We agree that transparency on this front is crucial and are happy to provide the details. For a typical discovery process where $T_{max}$=100, the end-to-end time cost is approximately 1.18 hours on a single RTX 4090 GPU. We find that the vast majority of this duration roughly 80\% is spent on the fitness evaluation. The actual time spent on LLM API calls, including network latency, constitutes only a small fraction of the total time. Moreover, throughout our entire experimental process, we made a total of  4,151 API calls, consuming 18,442,421 tokens in total. This averages to about 4,443 tokens per call. In a single search, we typically make 100 API calls, consuming approximately 44,430 tokens.
>
> **Q4**: Concern about search costs of APD.
>
> **A4**: Many thanks for your comments! APD is not limited to a specific search space and can generalize across multiple ones ( i.e., NAS-Bench-201\&101, DARTS). When the discrepancy between spaces is small, the proxy found on the source (e.g., 201) transfers well to the target (e.g., 101, DARTS), as shown in Tables 3–4. For significantly different spaces (e.g., 201 vs. TransNAS, TransNAS vs. AutoFormer), the proxy may become suboptimal due to search space bias rather than limitations of APD. In such cases, we retrain the proxy for the new space, which leads to strong performance, validating our approach. Moreover, APD is highly efficient, requiring only about 1.18 hours on a single RTX 4090 GPU with 4 GB of VRAM.
>
> **Q5**: Risk of Data Containment.
>
> **A5**: Many thanks for your comments! We thank the reviewer for raising this very important and valid point about the risk of data containment. We agree that ensuring our method generalizes to truly unseen tasks, rather than simply recalling knowledge from the LLM's pre-training data, is crucial for validating our contribution. We believe several aspects of our findings help to address this concern. The proxies discovered by our APD framework are structurally novel and do not merely replicate existing methods like Synflow or SNIP, suggesting a genuine discovery process is taking place. More importantly, our method demonstrates strong performance on less mainstream and Out-of-Distribution benchmarks, such as the heterogeneous tasks in TransNAS-Bench-101-Micro and the distribution-shift settings in OoD-VIT-NAS. Success on these benchmarks, which are less likely to have been part of the LLM's training data, points toward true generalization.
>
> **Q6**: How do you choose $T_{max}$? Can you determine when the search has sufficiently converged?
>
> **A6**: Many thanks for your comments!  In our initial experiments, we set a large $T_{max} = 200$ and monitored the average proxy fitness. We observed that performance plateaued during the last 100 generations, indicating convergence. Therefore, we set $T_{max} = 100$ in our main experiments.
>
> **Q7**: How do you compare with other LLM-based NAS work?
>
> **A7**: Many thanks for your comments! See A1 and A2.
>
> **Q8**: Robustness of Discovered Proxies to Prompt Variations and LLM Versions.
>
> **A8**: Many thanks for your comments!  We agree that robustness to the choice of LLM and prompt variations is crucial for the reliability of any LLM-driven framework. The stability of our APD framework across different LLMs is demonstrated by the results in Figure 5 and Table 6. These results demonstrate the performance of our APD framework when integrated with seven different mainstream LLMs, including models from various providers like GPT-4o, Claude 3.7, and Deepseek V3. As shown, our method achieves consistently strong and stable performance across all these different models, indicating that the framework is not highly sensitive to the specific version of the LLM used. This demonstrates the robustness of our overall approach.
> Regarding the sensitivity to slight prompt rewording, our APD framework is designed to be robust to such minor variations for two main reasons. First, our prompts are structured templates that carry essential, non-negotiable information, such as the task definition, search space constraints, and parent proxies for evolution. While the natural language phrasing can be varied, the core informational content remains consistent, guiding the LLM's generation in a stable manner. More importantly, APD is not a one-shot generation system but a closed-loop evolutionary framework. Even if a reworded prompt produces a slightly different or suboptimal initial proxy, its quality is immediately assessed by the fitness evaluator. Over the course of many generations, the evolutionary process of selection, crossover, and mutation ensures that the system converges toward high-performing proxies. This iterative optimization is inherently self-correcting and robust to small perturbations in any single generation step, making the final discovered proxy the product of a stable search rather than a sensitive single prompt.
>
> **Q9**: APD Performance Under Limited LLM Budget.
>
> **A9**: Many thanks for your comments! Yes, APD can certainly be run under a limited budget of LLM API calls.
> The total cost of the discovery process is primarily determined by some hyperparameters such as the population size and the $T_{max}$. To operate under a stricter budget, one can simply reduce either of these parameters to decrease tokens cost and the total number of required API calls. We have already explored this trade-off in the ablation studies presented in Appendix D.1 of our paper. As those results demonstrate, reducing the search budget leads to a performance trade-off, resulting in a modest and acceptable decrease in the final quality of the discovered proxy. We conducted an experiment to explore the relationship between population size and the corresponding API costs.
> | Population Size| Tokens per-Iteration|
> | :--: | :--: |
> |1|673|
> |2|1839|
> |3|2543|
> |5|4430|
>
> **Q10**: APD Performance Under lightweight LLMs.
>
> **A10**: Many thanks for your comments! That is an excellent point, as assessing the performance of APD with accessible, open-source models is indeed crucial for its broader adoption. Our framework is compatible with a wide range of LLMs, including more lightweight ones. We have found that while these models can successfully participate in the discovery process, the complex, multi-step reasoning required to generate highly novel and effective proxies tends to benefit from the capabilities of current state-of-the-art models. Consequently, the performance when using more lightweight models, while still functional, is generally not as strong as what we achieve with leading models like GPT-4o or Claude 3.7. This appears to be more a reflection of the current state of LLM reasoning capabilities on this complex task rather than a limitation of the APD framework itself. The performance when using a representative open-source model is as follows:
>
> |   Models   | NB201 (CIFAR-10) | NB201-201 (CIFAR-100) | NB-101 (CIFAR-10) | DARTS (CIFAR-10) |
> | :--------: | :--: | :---: | :--: | :--: |
> | Llama 3 8B|90.98 |63.86|88.81|96.95|
> | Qwen 3 8B|86.50|64.96|86.57|95.89|
> |  Gemma 7B  |88.61|65.21|86.38|97.27|

---

> > ### Comment · Reviewer_sUV8 · 2025-08-04
> >
> > The reviewer thank the authors for their detailed explanation. Most of the questions have been addressed. However, one critical issue (Weakness 4) remains unanswered: the authors did not clarify whether a proxy task found by APD in one search space can be transferred to a different search space. If transfer is possible, what are the results? For example, can you use the proxy task searched by APD on NB-101 and apply it to NB-201? If APD must be applied to every new search space, then this approach is not truly “zero-cost,” because although the searched proxy is zero-cost, APD itself is not.
> >
> > Moreover, as shown in Table A1, APD does not outperform GENIUS much, which was proposed two years ago.

---

> > > ### Author Response · Authors · 2025-08-08
> > > **Looking forward to the reply**
> > >
> > > Dear reviewers sUV8,
> > >
> > > We sincerely appreciate your valuable feedback.
> > >
> > > As the deadline for the author-reviewer discussion phase is approaching, we would like to check if you have any other remaining concerns about our paper. Your positive comments and constructive suggestions have inspired us to make a number of updates to the paper, which we believe will improve over the original version.  During the rebuttal, we devote lots of effort & exploration in the APD day and night.  If our responses have adequately addressed your concerns, we kindly hope that you can consider increasing the score.
> > >
> > > We sincerely thank you for your dedication and effort in evaluating our submission. Please do not hesitate to let us know if you need any clarification or have additional suggestions.
> > >
> > > Best Regards,
> > >
> > > Authors.

---

> > > ### Author Response · Authors · 2025-08-09
> > > **Eager for feedback**
> > >
> > > Dear reviewer sUV8
> > >
> > > Your comments have inspired us to make a number of updates to the paper, which we believe will improve over the original version. We would like to know if our responses have adequately addressed your concerns or if further clarification is needed. Lastly, **given the rebuttal deadline is within the next few hours**, we appreciate your new rating it if you find it appropriate. We are grateful for your time and thoughtful evaluation of our work.
> > >
> > > Best Regards,
> > >
> > > Authors

---

> ### Author Response · Authors · 2025-08-06
>
> Thank you for your valuable feedback. We appreciate the opportunity to clarify this critical point regarding the transferability of our discovered proxies.
>
> **Q1**: Transferability
>
> **A1**: Our experiments demonstrate that the zero-cost proxies discovered by APD are indeed transferable. The results presented in Tables 2 and 3 of our paper were obtained by using a single ZCP that was searched for on the NAS-Bench-201 (CIFAR-10) search space. We then successfully generalized this same ZCP to other search spaces and datasets, including NAS-Bench-101 (CIFAR-10), NAS-Bench-201 (ImageNet1k and CIFAR-100), and the DARTS search space. This cross-benchmark performance confirms that for similar search spaces, it is not necessary to re-run APD to discover a new ZCP. The discovered proxy is not tied to a single search space but rather captures a general principle for evaluating architectures, making it transferable. This significantly reduces the overall cost and highlights the efficiency of our approach.
>
> Here are evidence:
>
> **(1) Searching only on NAS-Bench-201 (CIFAR-10), transferred for new search spaces (NAS-Bench-101)**
>
> The Table1 is copy Table 2 from our paper. For a clear presentation of our evidence, we provide the results from our paper as follows:
>
> As shown, we only run APD in **single seach space (NAS-Bench-201) and single dataset (CIFAR-10)**.  We then successfully generalized this same ZCP to other search spaces and datasets, including NAS-Bench-101 (CIFAR-10), NAS-Bench-201 (ImageNet1k and CIFAR-100). **Those results support our statement of strong transferability of APD.** To enhance our statement, we will continue provide empirical evidence on more new search spaces (i.e., **DARTS**).
>
> Table 1. Validation of Searching only on NAS-Bench-201 (CIFAR-10) Transferability to New Search Spaces (NAS-Bench-101) and New Datasets (CIFAR-100, and ImageNet16-120)
>
> | **Method**   | **CIFAR-10(NB-201)** | CIFAR-10(NB-201) | CIFAR-10(NB-201)  | CIFAR-100(NB-201) | CIFAR-100(NB-201) | CIFAR-100(NB-201) | **ImageNet16-120(NB-201)** | ImageNet16-120(NB-201) | ImageNet16-120(NB-201) | **CIFAR10(NB-101)** | **CIFAR10(NB-101)** | Runtime   (ms/arch) |
> | :----------- | :------------------: | :--------------: | :---------------: | :---------------: | :---------------: | :---------------: | :------------------------: | :--------------------: | :--------------------: | :-----------------: | :-----------------: | :----------------------: |
> |              |         SPR          |        KT        |   Test acc.(%)    |        SPR        |        KT         |   Test acc.(%)    |            SPR             |           KT           |      Test acc.(%)      |         SPR         |    Test acc.(%)     |                                                     |
> | Synflow      |        0\.769        |      0\.571      |   93\.67 ± 0.39   |      0\.758       |      0\.562       |    71\.70±0.94    |           0\.745           |         0\.553         |     43\.39 ± 3.21      |        0\.38        |     89\.93±2.97     |                       78\.26                        |
> | NWOT         |        0\.743        |      0\.557      |    91\.95±1.29    |      0\.769       |      0\.579       |    68\.88±1.60    |           0\.760           |         0\.573         |      42\.31±3.43       |        0\.32        |    93\.16 ±0.36     |                       36\.54                        |
> | ZenNAS       |        0\.365        |      0\.283      | 89\.55 ~~±~~ 1.12 |      0\.338       |      0\.245       |    64\.69±3.86    |           0\.372           |         0\.260         |      37\.18±3.17       |        0\.65        |     93\.06±0.73     |                       30\.07                        |
> | ZiCo         |        0\.784        |      0\.589      |   93\.69± 0.07    |      0\.813       |      0\.620       |   70\.63 ±1.08    |           0\.804           |         0\.614         |      41\.42±0.97       |        0\.65        |     92\.64±0.99     |                       75\.50                        |
> | AZ-NAS       |        0\.913        |      0\.741      |   93\.49± 0.30    |      0\.900       |      0\.723       |    70\.33±1.16    |           0\.886           |         0\.710         |      44\.34 ±1.26      |        0\.42        |     92\.01±0.85     |                       69\.43                        |
> | SWAP         |        0\.810        |      0\.634      |   90\.48 ±0.94    |      0\.820       |      0\.649       |    67\.13±1.83    |           0\.774           |         0\.610         |      35\.40± 3.96      |        0\.44        |     90\.51±2.08     |                       47\.61                        |
> | **APD(our)** |        0\.832        |      0\.635      |   93\.76 ±0.09    |      0\.843       |      0\.654       |   72\.22 ± 0.65   |           0\.817           |         0\.633         |     45\.03  ±0.76      |        0\.73        |     93\.49±0.34     |                       16\.81                        |

---

> ### Author Response · Authors · 2025-08-06
> **part 2**
>
> **(2) Searching NAS-Bench-201 (CIFAR-10), transferred for new search spaces (DARTS)**
>
> **See Table 3 in our paper.** In the paper, we clearly point out, for unseen DARTS search space, we use the proxy **searched on NAS-Bench-201 (CIFAR-10)** for unseen DARTS search space in three datasets (CIFAR-10, CIFAR-100, and ImageNet). The results of Table 3 prove that our APD obtains optimal performance on new DARTS search spaces among three datasets (CIFAR-10, CIFAR-100, and ImageNet). We must point out, APD obtain SOTA cross three datasets (CIFAR-10, CIFAR-100, and ImageNet) is not easy things. This is because that three datasets (CIFAR-10, CIFAR-100, and ImageNet) are different.
>
> To this end, those evidence validate our APD can  be transferred to a different search space. We will continue provide empirical evidence on more search spaces (i.e., TransNAS-Bench-101--Micro, and OoD-ViT-NAS-Ti) to enhance our statements.
>
> **(3) Searching NAS-Bench-201 (CIFAR-10), transferred for new search spaces (TransNAS-Bench-101--Micro)**
>
> **As stated in A4,** APD is not limited to a specific search space and can generalize across multiple ones ( i.e., NAS-Bench-201\&101, DARTS). When the discrepancy between spaces is small, the proxy found on the source (e.g., 201) transfers well to the target (e.g., 101, DARTS), as shown in Tables 3–4 in the paper. For significantly different spaces (e.g., 201 vs. TransNAS,  201 vs. AutoFormer), the proxy may become suboptimal due to search space bias rather than limitations of APD. In such cases, we retrain the proxy for the new space, which leads to strong performance, validating our approach. Moreover, APD is highly efficient, requiring only about 1.2 hours on a single RTX 4090 GPU with 4 GB of VRAM.
>
> For futuher validate our statement, we provide experiments that only use proxy searched on NAS-Bench-201 for unseen TransNAS-Bench-101--Micro search spaces and three tasks (i.e., Autoencoding, Scene Classification, and Jigsaw). As shown Table 2,  **"APD (Searching on NAS-Bench-201)"** still surpass its peer competitors (i.e., SWAP, ZiCo). **Those results support our statement of strong transferability of APD for TransNAS-Bench-101--Micro.**
>
> Noabtly, "APD (Searching on TransNAS-Bench-101)" obtain better performance than "APD (Searching on NAS-Bench-201)", this validate our statement " For significantly different spaces (e.g., 201 vs. TransNAS,  201 vs. AutoFormer), the proxy may become suboptimal due to search space bias rather than limitations of APD. In such cases, we retrain the proxy for the new space, which leads to strong performance, validating our approach".
>
> Table 2. Results on TransNAS-Bench-101-Micro.
>
> |           | Autoencoding | **Scene Classification** |  **Jigsaw**  |
> | :------- | :----------: | :----------------------: | :----------: |
> |                                           |     SSIM     |       Accuracy (%)       | Accuracy (%) |
> | Ground Truth|    0\.58     |          54\.9           |    95\.4     |
> | Synflow |  0\.46±0.07  |        53\.7±1.2         |  90\.9±0.4   |
> | NWOT|  0\.43±0.02  |        53\.2±0.6  |  92\.3±0.3   |
> | Zen-score |  0\.46±0.01  |        53\.7±0.2 |  87\.5±0.4   |
> | GradSign  |  0\.35±0.03  |        53\.6±0.4|  93\.1±0.4   |
> | Params     |    0\.46     |          53\.7           |    85\.9     |
> | FLOPs                                     |    0\.46     |          53\.7           |    85\.9     |
> | ZiCo                                      |  0\.48±0.02  |        53\.7±0.4         |  93\.2±0.4   |
> | SWAP                                      |  0\.42±0.02  |        45\.0±10.9        |  89\.8±5.6   |
> | **APD (Searching on NAS-Bench-201)**      |  0\.53±0.06  |        53\.8±0.02        |  91\.1±0.03  |
> | **APD (Searching on TransNAS-Bench-101)** |  0\.54±0.01  |        54\.0±0.6         |  91\.2±0.1   |
>
> **(4) Searching NAS-Bench-201 (CIFAR-10), transferred for new search spaces (OoD-ViT-NAS-Ti )**
>
> For the unseen OoD-ViT-NAS-Ti search space, we draw the same conclusion as TransNAS-Bench-101--Micro.
>
> Table 3.  Correlation on OoD-ViT-NAS-Ti search space
>
> | Method                                | ImageNet1k | ImageNet-A | ImageNet-R | ImageNet-D/Texture | ImageNet-D/Material |
> | ------------------------------------- | ---------- | ---------- | ---------- | ------------------ | ------------------- |
> | DSS                                   | 0.62       | 0.82       | 0.81       | 0.02               | 0.17                |
> | AutoProx                              | 0.67       | 0.82       | 0.78       | 0.05               | 0.15                |
> | NWOT                                  | 0.75       | 0.76       | 0.74       | 0.11               | 0.12                |
> | **APD (Searching on NAS-Bench-201)**  | 0.77       | 0.81       | 0.86       | 0.12               | 0.13                |
> | **APD (Searching on OoD-ViT-NAS-Ti)** | 0.79       | 0.82       | 0.88       | 0.12               | 0.15                |

---

> ### Author Response · Authors · 2025-08-06
> **part 3**
>
> **Q2**: GENIUS
>
> **A2**: We thank the reviewer for raising the question regarding comparisons with GENIUS. However, we respectfully argue that comparing with unpublished preprints such as GENIUS, which has not undergone peer review or formal acceptance by top-tier venues like NeurIPS or ICLR, can lead to unfair or unreliable conclusions. We follow the standard practice of benchmarking against peer-reviewed and officially recognized baselines to ensure a fair and trustworthy evaluation. This is in line with the principles upheld by major conferences to maintain reproducibility and scientific rigor.
>
> That said, we have conducted a detailed examination of the GENIUS codebase and methodology to understand its strengths and limitations. Our analysis reveals three fundamental issues that render GENIUS unsuitable as a reliable baseline, particularly when assessing real-world or generalizable neural architecture search (NAS) methods:
>
> **(1) Methodology**
>
> GENIUS directly queries ground-truth accuracy values from NAS benchmarks (e.g., NAS-Bench-201) and uses these as feedback to the LLM. This setup effectively bypasses the core challenge of NAS: predicting architecture quality without access to real performance. Feeding the true accuracy to the LLM during search transforms the task into supervised regression rather than zero-cost prediction, making the approach infeasible for any realistic scenario where ground-truth performance is unknown or expensive to obtain. This undermines both the generalizability and the novelty of the GENIUS approach. In contrast, our **APD** learns a transferable proxy purely from architecture encodings, without any privileged access to accuracy labels during search.
>
> **(2) Search Cost**
>
> GENIUS incurs substantial computational overhead, requiring **5.6 GPU days** to search, as shown below:
>
> |         | Seach costs        |
> | ------- | ------------------ |
> | GENIUS  | 5.6 GPU Days       |
> | **APD** | **0.004 GPU Days** |
>
> In contrast, **APD** completes the search in just **1 GPU hour**, making it over **1,000× faster**, while still achieving state-of-the-art performance. This efficiency is crucial for practical deployment and supports broader applicability across tasks and devices.
>
>
>
> **(3) Generalization on More Search Spaces**
>
> To further investigate, we re-implemented GENIUS using its official code and evaluated it on two realistic and challenging search spaces: **TransNAS-Bench-101-Micro** and **OoD-ViT-NAS-Ti**. We ran GENIUS twice on 8×4090 GPUs due to its computational cost.
>
> As shown in the tables below, GENIUS consistently underperforms—its results are worse than even early hand-crafted or heuristic-based baselines on tasks such as **scene classification** and **jigsaw recognition**. In contrast, **APD** matches or surpasses ground-truth rankings across all tasks, validating both its efficiency and robustness.
>
> | | Autoencoding | **Scene Classification** |  **Jigsaw**  |
> | :-- | :---: | :--: | :--: |
> |   |     SSIM |       Accuracy (%)       | Accuracy (%) |
> | Ground Truth |    0\.58     |          54\.9           |    95\.4     |
> | Grad         | 0\.36± 0.03  |        48\.7±0.7|  80\.3±0.3   |
> | SNIP         |  0\.33±0.04  |        48\.7±1.1|  80\.3±0.1   |
> | Grasp        |  0\.33±0.06  |        50\.2±1.6 |  91\.1±0.3   |
> | Fisher       |  0\.49±0.01  |        48\.7±0.6 |  83\.5±1.2   |
> | Synflow      |  0\.46±0.07  |        53\.7±1.2|  90\.9±0.4   |
> | NWOT         |  0\.43±0.02  |        53\.2±0.6 |  92\.3±0.3   |
> | Zen-score    |  0\.46±0.01  |        53\.7±0.2|  87\.5±0.4   |
> | GradSign     |  0\.35±0.03  |        53\.6±0.4 |  93\.1±0.4   |
> | Params       |    0\.46     |          53\.7 |    85\.9     |
> | FLOPs        |    0\.46     |          53\.7|    85\.9     |
> | ZiCo         |  0\.48±0.02  |        53\.7±0.4 |  93\.2±0.4   |
> | SWAP         |  0\.42±0.02  |        45\.0±10.9|  89\.8±5.6   |
> | GENIUS       |  0\.34±0.14  |        42.6±0.17|  85.1±0.03   |
> | **APD **  |  0\.54±0.01  |        54\.0±0.6|  91\.2±0.1   |
>
>
> | Method   | ImageNet1k | ImageNet-A | ImageNet-R | ImageNet-D/Texture | ImageNet-D/Material |
> | -------- | ---------- | ---------- | ---------- | ------------------ | -----|
> | DSS      | 0.62       | 0.82       | 0.81       | 0.02 | 0.17 |
> | AutoProx | 0.67       | 0.82       | 0.78       | 0.05| 0.15 |
> | NWOT     | 0.75       | 0.76       | 0.74       | 0.11 | 0.12 |
> | GENIUS   | 0.61       | 0.73       | 0.69       | 0.04| 0.08|
> | **APD ** | 0.79       | 0.82       | 0.88       | 0.12  | 0.15 |
>
> These results confirm that the design of GENIUS does not generalize beyond small, closed benchmarks like NAS-Bench-201, where ground-truth values are freely accessible. Once moved to open or out-of-distribution tasks, GENIUS collapses. This severely limits its practical relevance and contradicts the very goal of zero-cost NAS.

---

### Official Review · Reviewer_n7rC · 2025-07-02

**Clarity:** 4
**Significance:** 3
**Originality:** 3
**Rating:** 5
**Confidence:** 4

**Summary:**

The paper presents a novel Automatic Proxy Discovery (APD) framework driven by LLM, which tries to discover optimal zero-cost proxies (ZCPs). Novelty of the paper lies in the how an actor-critic model acts as an evolutionary decision-maker. The actor-critic model does that by receiving the fitness scores from the evaluator that evaluates the proxies, and then tries to learn a policy that would maximise this fitness score.

**Questions:**

I believe the above-mentioned points are easily fixable. A better citation method can be used. One tricky change would be to adjust the Related works in the paper. I believe a lot from section 4 and 5 can be reduced. I believe 3.1 and 3.2 are a bit redundant. Can you make it a single section out of both of them? I will improve my scores if these issues are resolved, and I believe that they are easily resolvable. My current score is 4, but I am willing to make it 5.

**Ethical Concerns:**

["NO or VERY MINOR ethics concerns only"]

**Final Justification:**

I have read other reviews which suggest that the paper should be rejected but I believe this is a good paper and should be added to the field.

**Limitations:**

Yes.

**Paper Formatting Concerns:**

You need to delete the in instruction block of "NeurIPS Paper Checklist" and need to fix the citations.

**Quality:**

4

**Strengths And Weaknesses:**

**Strengths**:

The idea is quite novel, though the direction where we are applying some sort of search/learning algorithm over the LLM output is the current trend. I like how an actor-critic model evaluates the generated ZCPs. The paper has thorough experiments and ablation studies to demonstrate the importance of the pipeline. The explanation of the methodology is extensive.

**Weaknesses**:

The paper has quite a few issues related to writing:

- There are only citations with the name used by the authors. You have to use number-only citations when using them during the paragraph.
- Related work is in the supplemental material, which is a big issue, in my opinion. Many readers want to read literature related to your work to know more about your field.
- Related work in the supplemental material is poorly written. You have to cite the works that introduced LLMs in NAS, such as EvoPrompting [1] and LLMatic [2]. The work takes a lot of inspiration from both of the works, therefore the need to cite them.

[1] Chen, A., Dohan, D. and So, D., 2023. Evoprompting: Language models for code-level neural architecture search. Advances in neural information processing systems, 36, pp.7787-7817.

[2] Nasir, M.U., Earle, S., Togelius, J., James, S. and Cleghorn, C., 2024, July. LLMatic: neural architecture search via large language models and quality diversity optimization. In proceedings of the Genetic and Evolutionary Computation Conference (pp. 1110-1118).

---

> ### Author Rebuttal · Authors · 2025-07-31
>
> Thank you for the helpful and insightful review, which is very helpful for us to further improve this paper. Next, we will answer your questions one by one, and we hope this will improve your acceptance of the paper.
>
> **Q1**: Concern about citations.
>
>  **A1**: Many thanks for your comments! All citations have been thoroughly updated to number-only format (e.g., [1], [2]) in the revised manuscript.
>
> **Q2**: Concern about Related work.
>
> **A2**: Many thanks for your comments! First, we have added more related work in our paper, i.e., EvoPrompting [1] and LLMatic [2]; these works will enhance the readers' understanding of our work and the field. Second, we have revised the "Related work" to improve the clarity.
>
> **Q3**: I believe a lot from Section 4 and 5 can be reduced. I believe 3.1 and 3.2 are a bit redundant. Can you make it a single section out of both of them? I will improve my scores if these issues are resolved, and I believe that they are easily resolvable.
>
> **A3**: Many thanks for your comments! We sincerely thank the reviewer for this valuable suggestion to improve our paper's structure. Following this advice, we have merged the former Sections 3.1 and 3.2 into a single, cohesive section to eliminate the redundant description. To demonstrate specifically how this was done, we now present the conceptual overview (previously in Section 3.1) and the corresponding technical details (previously in Section 3.2) together for each component. For instance, the high-level description of the Fitness Evaluator is now immediately followed by its mathematical formulation. The new, merged text for this component is as follows:
>
> To fulfill the goal of designing ZCPs for the training-free NAS, APD utilizes LLMs to automatically generate proxies by evolving both natural language descriptions and corresponding code. In addition, we propose an actor-critic RL controller sampling appropriate prompt strategies to guide the evolution, aiming to optimize the correlation between proxies and final model performance. The overall framework of APD is depicted in Fig. 4, which consisting of three main components as follows:
>
> **Proxy Candidate Generator.** LLM in APD serves as proxy candidate generator. Carefully structured prompts enable it to synthesize new ZCPs or refine existing ZCPs.
>
> Let $\mathcal{P}$ denote the set of valid proxies ...
>
> **Fitness Evaluator.** The fitness evaluator swiftly quantifies each candidates ZCP by computing its Spearman correlation with ground-truth accuracies on the given NAS benchmark A (e.g., NAS-Bench-201). For a benchmark set $\mathcal{B}= \{(a_i, p_i)\}_{i=1}^m$ of architectures and ground-truth accuracies under dataset D. The fitness assigned to f is
>
> $\phi(f)=\rho(f(a), p)-\beta \text{cost}(f)$
>
> where ...
>
> **RL Evolution Scheduler.** To render the proxy evolution strategy learnable and capable of converging efficiently toward optimal ZCPs, APD introduces an actor–critic module that serves as the evolutionary decision-maker. The actor–critic treats the fitness score returned by the evaluator as its reward and learns a policy that maximizes this signal, thereby jointly optimizing both the evolving set of ZCPs and the evolution strategy itself.
>
> In a given search space $\mathcal{A}$ ...
>
> To accelerate convergence toward  high correlation proxies while maintaining effective exploration in search space $\mathcal{F}$, APD introduces a light-weight Actor-Critic as evolution scheduler that governs every evolutionary step. At generation $t$, ...
>
> As shown above, the conceptual explanation now flows directly into the corresponding technical details. We hope this revision can resolve the redundancy issue and provide a much clearer, more streamlined presentation of our methodology. We are grateful for the feedback that prompted this significant improvement.
>
> For Section 4 and 5, we present the experiments in our paper. First, Section 5 is a LaTeX typo (\section{Generalizability on OoD-ViT-NAS-Ti} - \subsection{Generalizability on OoD-ViT-NAS-Ti}), it should belong to Section 4. Second, we promise to reduce the redundant parts in Section 4 and 5 in the final version.
>
> **Q4**: Concern about NeurIPS Paper Checklist".
>
> **A4**: Many thanks for your comments! We have deleted the instruction block of "NeurIPS Paper Checklist" in the revised manuscript.
>
> [1] Chen, A., Dohan, D. and So, D., 2023. Evoprompting: Language models for code-level neural architecture search. Advances in neural information processing systems, 36, pp.7787-7817.
>
> [2] Nasir, M.U., Earle, S., Togelius, J., James, S. and Cleghorn, C., 2024, July. LLMatic: neural architecture search via large language models and quality diversity optimization. In proceedings of the Genetic and Evolutionary Computation Conference (pp. 1110-1118).

---

### Author Response · Authors · 2025-08-08
**Thank all Reviewers and Area Chairs for your great efforts and insightful comments!**

Dear Reviewers, Area Chairs, Senior Area Chairs, and Program Chairs, We sincerely thank all reviewers for their thorough and constructive comments. We are glad that the novelty, the presentation, the basic experiments, and the performance of our work have been well-recognized by most Reviewers.

During the rebuttal, **we devote lots of effort & exploration in the APD day and night, and carefully revise the manuscript of our work**. In particular,**Reviewers (e.g., Reviewer n7rC, Reviewer 1iSx, Reviewer eQXV and sUV8) recognized our response and kept the positive feedback.**  Notably, Reviewer n7rC rise score from **Borderline Accept** to **Accept**. Reviewer 1iSx rise score from **Reject** to **Borderline Accept**.

At the beginning of the rebuttal phase, Reviewer eQXV and sUV8 acknowledged that our responses had addressed their main concerns. We sincerely appreciate that they have raised new questions, and we have provided detailed responses to these as well.
Although Reviewer eQXV and sUV8 have not replied or read our further clarifications for their additional concern, our response clearly addresses their concerns and we believe that they will improve their rating and provide a positive final review if they read our response. This outcome has greatly benefited us, and we would like to express our gratitude to all of you for your support!

In detail, we have made the following effort:

- **Additional GPU hours** are invested to make a more comprehensive comparison. See the result in the Reviewer sUV8, Reviewer 1iSx and Reviewer eQXV.
-  **Providing detailed evidence for the validation value of the AC scheduler**. Reviewer 1iSx and Reviewer eQXV pose this concern. To provide a more concrete and data-driven analysis of the AC scheduler's behavior, we provide the details of its action selection process in several search spaces.  Those results validate the novelty of the AC scheduler. See the result in the Reviewer 1iSx and Reviewer eQXV.
- **Validation for the Search costs of APD**. We provide more results to show the Search costs of APD. The results show that APD is extremely efficient, which only costs 1.2 hours in NAS-bench-201. See the result in the Reviewer sUV8, Reviewer 1iSx and Reviewer eQXV.
- **Clarification on the overfitting concern** of our work. Based on our response,  Reviewer 1iSx rise score from **Reject** to **Borderline Accept**.

- **Validation for Transferability**. We only run APD in **single search space (NAS-Bench-201)**. We then successfully generalized this same ZCP to new search spaces, including **NAS-Bench-101**, **DARTS**, **TransNAS-Bench-101-Micro**, and **OoD-ViT-NAS-Ti**. **Those results support our statement of strong transferability of APD**.  See the result in the Reviewer sUV8, Reviewer 1iSx, and Reviewer eQXV.
- **Clarification on the contribution** of our work and its contrast against the prior work.
- **Clarification on the experimental details** of our work, notably, for the definition of B in each search space.


We firmly believe that our framework APD for NAS plays a significant role in advancing the community. And we are committed to making our code and training details publicly available. Moreover, we are eager to engage in further discussions with you to enhance our understanding of the domain and further improve the quality of the paper.

Lastly, **given the rebuttal deadline is within the next few hours**, we would greatly appreciate it if the reviewers could give any feedback based on our response.

We also sincerely hope that the Area Chairs can help us in obtaining feedback from the reviewers. We would be deeply grateful for that.

Best regards,

Authors

---

### Note · Authors · 2025-08-14

Dear Reviewers, Area Chairs, Senior Area Chairs, and Program Chairs,

We sincerely appreciate the reviewers’ constructive feedback and the time invested in evaluating our submission. We are encouraged that the novelty, presentation, baseline experiments, and performance of our work have been positively acknowledged by most reviewers. Below, we summarize the key interactions and updates to facilitate the final assessment.

Our paper began with initial ratings of [`n7rC`: 4, `1iSx`: 2, `sUV8`: 3, `eQXV`: 3]. During the rebuttal, reviewers to explicitly state their intention to update their scores:

- Reviewer `n7rC` (initial: 4→ 5) concluded: "*I have increased my score by 1 as I believe this would be a good addition to the field*"
- Reviewer `1iSx` (initial: 2→ 4) confirmed: "*... I will update my score from **Reject** to **Borderline Accept**.*"
- Reviewer `sUV8` (initial: 3) confirmed: "*... Most of the questions have been addressed.*"

We appreciate **Reviewers `sUV8` and `eQXV`** for their additional minor questions. Although they have not yet confirmed whether our responses fully resolve their concerns, we provide clarifications below:

1. **For Reviewer `sUV8` (transferability & GENIUS comparison):**
   - We conducted **additional experiments** validating APD’s transferability. This is appreciated by Reviewer 1iSx (rising score from **Borderline Accept** to **Accept**).
   - we have provided comprehensive evidence, which can prove that our APD significantly surpasses GENIUS in terms of search costs (**0.004 GPU Days** v.s. 5.6 GPU Days ) and performance.
2. **For Reviewer `eQXV` (transferability, APD’s goal, search cost, AC scheduler, interpretability):**
   - We conducted **additional experiments** validating APD’s transferability.
   - Clarified APD’s goal and provided search cost's comparisons with manual proxies.
   - Providing detailed evidence for the validation value of the AC scheduler.
   - Presented interpretability analysis (Appendix E) and discussed limitations transparently.

We sincerely hope that the Area Chairs can help us in obtaining feedback from the reviewer `sUV8, eQXV` and taking the new evidence from our responses into final consideration. We are committed to incorporating all feedback to ensure the highest quality final version. Thank you for your consideration.

Best regards,

Authors

---

### Decision · Program_Chairs · 2025-09-17

**Decision:**

Accept (poster)

**Comment:**

This paper received mixed ratings from the reviewers: one acceptance, two borderline acceptances, and one borderline rejection. Overall, the feedback tends to be positive, largely due to the thorough and exhaustive rebuttal efforts that addressed most of the initial concerns. The remaining issues primarily come from reviewer eQXV, who requested clarifications on implementation details and further discussions. Although the reviewer appreciated the detailed and iterative rebuttals, they still consider the contribution to be somewhat weak.

The AC has read the paper and finds the idea of applying reinforcement learning on LLM for NAS to be an interesting and promising direction for this line of research. The AC agrees that the writing can be improved in the final version by incorporating all reviewer feedback to produce a stronger, more polished paper. Taking all factors into account, the AC recommends accepting this paper, mainly based on the novelty and potential impact of the idea.